# BRIGHT: A REALISTIC AND CHALLENGING BENCHMARK FOR REASONING-INTENSIVE RETRIEVAL

**Hongjin Su**[*h]    **Howard Yen**[*p]    **Mengzhou Xia**[*p]    **Weijia Shi**[w]    **Niklas Muennighoff**[s]

**Han-yu Wang**[h]    **Haisu Liu**[h]    **Quan Shi**[p]    **Zachary S. Siegel**[p]    **Michael Tang**[p]

**Ruoxi Sun**[g]    **Jinsung Yoon**[g]    **Sercan Ö. Arık**[g]    **Danqi Chen**[p]    **Tao Yu**[h]

[h] The University of Hong Kong    [p] Princeton University    [s] Stanford University
[w] University of Washington    [g] Google Cloud AI Research

{hjsu,tyu}@cs.hku.hk    {hyen,mengzhou,danqic}@cs.princeton.edu

## ABSTRACT

Existing retrieval benchmarks primarily consist of information-seeking queries (e.g., aggregated questions from search engines) where keyword or semantic-based retrieval is usually sufficient. However, many complex real-world queries require in-depth reasoning to identify relevant documents that go beyond surface form matching. For example, finding documentation for a coding question requires understanding the logic and syntax of the functions involved. To better benchmark retrieval on such challenging queries, we introduce BRIGHT, the first text retrieval benchmark that requires *intensive reasoning* to retrieve relevant documents. Our dataset consists of 1,384 real-world queries spanning diverse domains, such as economics, psychology, mathematics, and coding. These queries are drawn from naturally occurring and carefully curated human data. Extensive evaluation reveals that even state-of-the-art retrieval models perform poorly on BRIGHT. The leading model on the MTEB leaderboard (Muennighoff et al., 2023) SFR-Embedding-Mistral (Meng et al., 2024), which achieves a score of 59.0 nDCG@10,[1] produces a score of nDCG@10 of 18.3 on BRIGHT. We show that incorporating explicit reasoning about the query improves retrieval performance by up to 12.2 points. Moreover, incorporating retrieved documents from the top-performing retriever boosts question-answering performance. We believe that BRIGHT paves the way for future research on retrieval systems in more realistic and challenging settings.[2]

## 1 INTRODUCTION

Information retrieval is a widely employed technology that assists users in locating relevant information from extensive corpora, containing documents, web pages, and logging records (Bajaj et al., 2016; Thakur et al., 2021). Relevant information can relate to user queries in different ways—sometimes through straightforward matching patterns like shared keywords or semantic similarities, and other times through deeper, more nuanced connections such as analogous underlying principles. In many real-world scenarios, user queries can be highly complex, and finding the relevant documents requires intensive reasoning. For instance, an economist might want to find a story explained by the same economic theory as another story, or a programmer might want to use an error message to locate the corresponding syntax documentation. For these applications, relevant documents cannot be directly retrieved through lexical or semantic matching alone, but instead require additional reasoning steps to identify.

In this work, we study the problem of reasoning-intensive retrieval with BRIGHT, a new benchmark that requires intensive reasoning to retrieve relevant documents. Existing retrieval benchmarks, such as BEIR (Thakur et al., 2021) and MTEB (Muennighoff et al., 2023), primarily focus on fact-based

---

[*] Equal contribution.

[1] Retrieved from the MTEB leaderboard on 2024-05-28.

[2] Our code and data are available at https://github.com/xlang-ai/BRIGHT and https://huggingface.co/datasets/xlangai/BRIGHT.

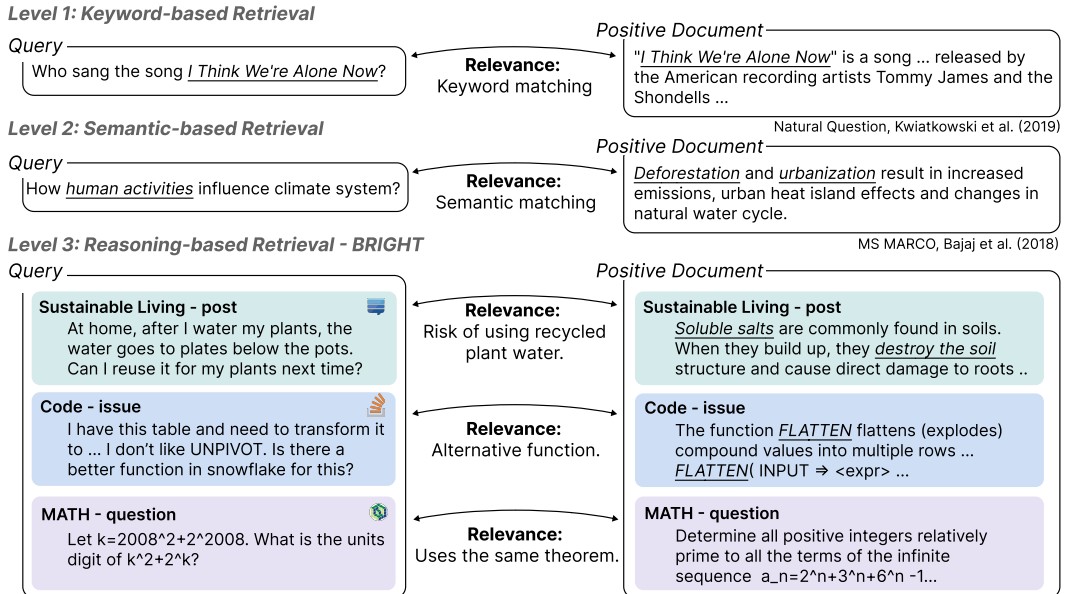

Figure 1: Existing retrieval benchmarks focus on keyword-based retrieval (level 1), or semantic-based retrieval (level 2), e.g., NQ, MS MARCO datasets (Kwiatkowski et al., 2019; Bajaj et al., 2018). BRIGHT **introduces level 3 retrieval, where the relevance between queries and documents requires intensive reasoning to determine.** Our data consists of natural user queries from diverse domains (e.g., economics, math, earth sciences, etc.). BRIGHT corpora also span across web data, such as blogs, syntax documentation, and STEM problem-solutions.

queries typically derived from search engines, where the relevance between queries and documents is often straightforward and can be detected through simple keyword or semantic matching (Lee et al., 2019; Karpukhin et al., 2020). These datasets focus on retrieving specific pieces of information (e.g., "the widest highway in North America"), which leads to the relevant documents often having high lexical or semantic overlap with the queries. In contrast, the relevance between queries and documents in BRIGHT is not easily detectable through simple keyword or semantic matching, and requires deliberate reasoning due to the complex nature of our domains and queries (Figure 1).

BRIGHT consists of 12 datasets from diverse and advanced domains, sourced from naturally occurring human data and meticulously curated sources. The benchmark comprises two main components: 1) seven datasets are constructed from StackExchange, where relevance is defined by whether a document is cited in the answer, and further validated through multiple annotators to ensure that the positive documents effectively support addressing queries. Given the inherent subjectivity of this assessment, we only include documents unanimously agreed upon by the annotators. 2) The remaining five datasets focus on coding and math problems, where the queries are inherently linked to the positive documents due to their shared underlying algorithms or theorems.

We evaluate 13 representative retrieval models of varying sizes and architectures. Comprehensive experiments highlight the challenges posed by BRIGHT, as the best-performing model, SFR-Embedding-Mistral (Meng et al., 2024), which scores 59.0 on the MTEB retrieval subset, BEIR (Thakur et al., 2021), achieves only an nDCG@10 score of 18.3 on BRIGHT. To identify promising directions for improving BRIGHT, we explore various strategies—one effective approach leverages LLMs to generate chain of thought reasoning steps (Wei et al., 2022) about the query before retrieval, resulting in average improvements of up to 12.2 points. Furthermore, augmenting the downstream model with retrieved documents from the top-performing retriever, Qwen, enhances the question-answering performance compared to the closed-book setting. However, using the oracle documents results in a larger improvement, highlighting the potential benefits of improving retriever models.

Moreover, our results demonstrate the robustness of BRIGHT against potential data leakage during large-scale pre-training, as no substantial performance gains are observed even when models are

further trained on documents from the benchmark dataset. We hope that our findings inspire research directions to advance the state of the art in reasoning-intensive retrieval.

## 2 RELATED WORK

**Benchmarking retrieval.** Existing information retrieval (IR) datasets are typically constructed for information-seeking tasks, such as question-answering (Voorhees & Tice, 2000; Craswell et al., 2020; Kwiatkowski et al., 2019; Chen et al., 2017; Maia et al., 2018), claim verification (Thorne et al., 2018; Diggelmann et al., 2020; Wadden et al., 2020), or entity retrieval (Hasibi et al., 2017; Petroni et al., 2021). Recent works expand retrieval benchmarks with more scenarios, such as instruction-following (Su et al., 2023; Weller et al., 2024; Oh et al., 2024; Li et al., 2024), multi-hop (Yang et al., 2018), and long-context retrieval (Saad-Falcon et al., 2024; Zhu et al., 2024). Comprehensive benchmarks like BEIR (Thakur et al., 2021) evaluate retrieval systems on diverse domains and tasks, with relevant documents sharing high semantic overlap with the query. Closest to our work, BIRCO (Wang et al., 2024) is designed to evaluate retrieval systems based on multifaceted objectives by leveraging existing datasets. However, it is limited to the LLM reranking setting and uses only a small candidate pool ($\sim 100$ documents) for each query. RAR-b (Xiao et al., 2024) adapts existing commonsense, math, and code datasets into a retrieval setting to test whether models can directly retrieve answers to reasoning problems. However, we focus on a more realistic scenario where the answers are unlikely to be found as a substring in documents.[3] While both of the benchmarks focus on reasoning-intensive retrieval, **BRIGHT** is the first benchmark to collect realistic user queries and align them with relevant natural documents from large corpora, requiring deep reasoning to identify the correct matches.

**Dense retrieval models and retrieval augmented generation.** State-of-the-art retrieval systems often use dense models to encode text with rich representation. These models are trained on unsupervised data (Lee et al., 2019; Izacard et al., 2022), supervised data (Su et al., 2023; Asai et al., 2023a; Muennighoff, 2022), as well as LLM-generated data (Lee et al., 2024; Wang et al., 2023a; Muennighoff et al., 2024). In this work, we benchmark a diverse set of models across different axes: sparse and dense; small and large; open-source and proprietary. Additionally, as dense generative models continue to improve, retrieval-augmented generation (RAG; Asai et al., 2020; Borgeaud et al., 2022; Asai et al., 2023b; Muennighoff et al., 2024; Asai et al., 2024; Gao et al., 2023; Shi et al., 2024), which retrieves relevant documents to help generate coherent answers, has become an important application. Although we mainly focus on retrieval in this work, we conduct initial analyses and demonstrate that using stronger retrievers improves model generation for reasoning-intensive tasks.

**Benchmarking reasoning.** Many benchmarks aim to evaluate the reasoning abilities of LLMs, especially focused on mathematics and coding. As for mathematics, for example, datasets include GSM8K (Cobbe et al., 2021) and its extensions GSM1K (Zhang et al., 2024), TheoremQA (Chen et al., 2023a), MATH (Hendrycks et al., 2021), and LeanDojo (Yang et al., 2023). As for coding, HumanEval (Chen et al., 2021), MBPP (Austin et al., 2021), and LiveCodeBench (Jain et al., 2024) are often used. These benchmarks contain question-answer pairs and are usually sourced from textbooks, online resources, competitions, or domain experts. We source queries from selected high-quality datasets and construct **BRIGHT** through additional annotations, creating a realistic reasoning-intensive retrieval benchmark.

## 3 CONSTRUCTING **BRIGHT**

We introduce **BRIGHT**, a retrieval benchmark that tests whether retrieval systems can match queries and documents whose relevance requires intensive reasoning to solve, beyond just lexical and semantic similarities. In this section, we first formulate the task of reasoning-intensive retrieval (Section 3.1). Then, we detail the data collection process for the data from StackExchange (Section 3.2), coding datasets (Section 3.3), and theorem-based questions (Section 3.4). In Table 1, we present the benchmark statistics.

---

[3]We compare to RAR-b in detail in Appendix I.

Table 1: **Data statistics of BRIGHT.** For each dataset, we show the number of queries (**Q**) and documents ($\mathcal{D}$), the average number of positive documents ($\mathcal{D}^+$) per example, the average length of queries and documents (measured by the GPT-2 tokenizer Radford et al. (2019)), and sources of queries and documents. Q&Sol refers to demonstration examples of question-solution pairs. TheoremQA-Q and TheoremQA-T refer to question retrieval and theorem retrieval based on TheoremQA respectively. Examples for each dataset can be found in Appendix C.

| Dataset | Total Number | | | Avg. Length | | Source | | Examples |
|---|---|---|---|---|---|---|---|---|
| | **Q** | $\mathcal{D}$ | $\mathcal{D}^+$ | **Q** | $\mathcal{D}$ | **Q** | $\mathcal{D}$ | |
| *StackExchange* | | | | | | | | |
| Biology | 103 | 57,359 | 3.6 | 115.2 | 83.6 | | | Tab. 20 |
| Earth Science | 116 | 121,249 | 5.3 | 109.5 | 132.6 | | Web pages: | Tab. 21 |
| Economics | 103 | 50,220 | 8.0 | 181.5 | 120.2 | StackExchange | article, | Tab. 22 |
| Psychology | 101 | 52,835 | 7.3 | 149.6 | 118.2 | post | tutorial, | Tab. 23 |
| Robotics | 101 | 61,961 | 5.5 | 818.9 | 121.0 | | news, blog, | Tab. 24 |
| Stack Overflow | 117 | 107,081 | 7.0 | 478.3 | 704.7 | | report ... | Tab. 25 |
| Sustainable Living | 108 | 60,792 | 5.6 | 148.5 | 107.9 | | | Tab. 26 |
| *Coding* | | | | | | | | |
| LeetCode | 142 | 413,932 | 1.8 | 497.5 | 482.6 | Coding question | Coding Q&Sol | Tab. 27 |
| Pony | 112 | 7,894 | 22.5 | 102.6 | 98.3 | Coding question | Syntax Doc | Tab. 28 |
| *Theorems* | | | | | | | | |
| AoPS | 111 | 188,002 | 4.7 | 117.1 | 250.5 | Math Olympiad Q | STEM Q&Sol | Tab. 29 |
| TheoremQA-Q | 194 | 188,002 | 3.2 | 93.4 | 250.5 | Theorem-based Q | STEM Q&Sol | Tab. 30 |
| TheoremQA-T | 76 | 23,839 | 2.0 | 91.7 | 354.8 | Theorem-based Q | Theorems | Tab. 31 |

## 3.1 TASK FORMULATION

Given a query $Q$ and the retrieval corpus $\mathcal{D} = \{D_1, \ldots, D_n\}$, retrievers are tasked to find relevant documents $\mathcal{D}_Q^+ = \{D_{Q,1}^+, \ldots, D_{Q,m}^+\} \subset \mathcal{D}$, where $m \ll n$ (positive). Negative documents are defined as $\mathcal{D}_Q^- = \mathcal{D} \setminus \mathcal{D}_Q^+$. In reasoning-intensive retrieval, the relevant document set $\mathcal{D}_Q^+$ is connected to the query $Q$ through specific reasoning traces or explanations (e.g., underlying principles, algorithms, or theorems) related to the query. For instance, common reasoning traces might involve identifying the query's intent, analyzing and modeling the problem, and drawing sub-conclusions based on the provided descriptions. Such reasoning traces are typically absent from the query itself, making direct retrieval using only the query very challenging.

## 3.2 STACKEXCHANGE: RETRIEVING WEB PAGES THAT HELP ANSWER QUESTIONS

> **Relevance**: A document is considered relevant to a query only if it is cited in an accepted or highly voted answer and unanimously confirmed by annotators and domain experts that it helps reason through the query with critical concepts or theories.

StackExchange[4] is a popular community-driven platform where users ask questions and receive answers from other users. Among its 170+ sites, we select 7 diverse and knowledge-intensive domains: Sustainable Living, Economics, Psychology, Robotics, Earth Science, Biology, and coding in Stack Overflow. Unlike the short questions in traditional retrieval benchmarks, questions on StackExchange often contain long and technical descriptions of the problems and end with a logically complex question, such as fixing a bug. Responses often link to external web pages that contain relevant information to address the question. We construct query-document pairs based on user posts and documents referenced in the answers (see Figure 2).

**Selecting posts.** Human annotators[5] browse posts from newest to oldest and select a post with at least one answer that (1) is accepted by the user or receives > 5 votes, and (2) contains one or more URL links. This process ensures that each dataset has a sufficient number of high-quality examples.

---

[4] https://stackexchange.com/
[5] authors and college students from corresponding fields.

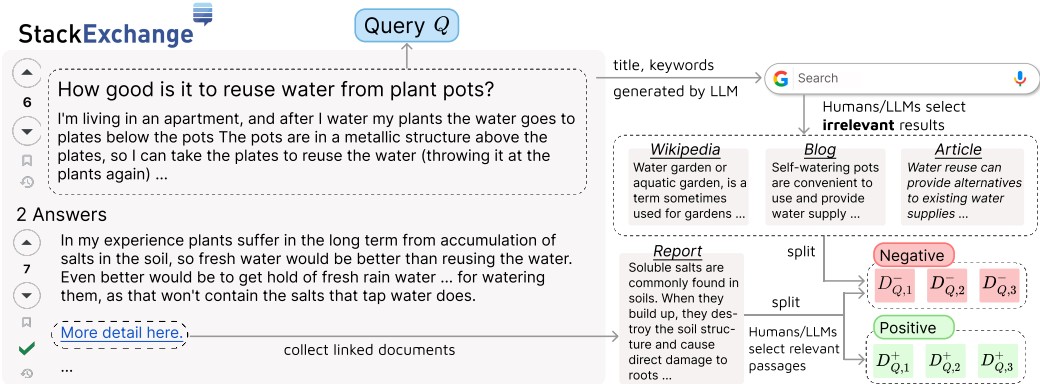

Figure 2: **An overview of the data annotation process for StackExchange data.** The post content is used as the query. Positive documents are selected passages from web pages linked in the answer, while the remaining passages, results from Google, and passages filtered by annotators are considered negatives. The web pages can include content from Wikipedia, blogs, articles, reports, and more.

**Constructing query and positive documents.** For each selected post, we construct the query and positive documents as follows:

Step 1: The annotator combines the title and content of the post to form the query $Q$.

Step 2: The annotator visits web pages linked in the answers and includes them as positive documents if they are relevant to queries or discards them otherwise.

Step 3: If no web page is considered positive, the post itself is discarded. For each collected web page, the annotator splits the content into passages
and selects positives $\mathcal{D}_{Q,i}^+$ following the relevance definition.

**Constructing negative documents.** To prevent models from relying purely on lexical or semantic similarities, we ensure that the negative documents for each query also address similar topics. Specifically, annotators gather search results from Google on the same topic and select documents that, while topically related, do not meet the specific requirements of the query (for more examples, see Appendix C). The collection procedure is as follows:

Step 1: Annotators search Google using the posts' title or LLM-summarized post keywords, and identify web pages that are semantically similar but not relevant to answering the question.

Step 2: Annotators collect up to 5 negative web pages for each query and split them into hard negative passages, which consist of $\mathcal{D}_Q^-$.

For each dataset, we compile all the collected passages into a unified retrieval corpus $\mathcal{D}$. For any given query, all passages in the corpus—excluding its positive passages—are treated as negatives, including positive and negative passages associated with other queries. In contrast to traditional retrieval tasks such as open-domain QA (Fan et al., 2019; Kwiatkowski et al., 2019), where the retrieval pool typically includes documents that directly answer the query, we simulate a realistic scenario where positive documents only provide useful information to help users derive an answer. Please find more details on the construction of the dataset in Appendix B.1.

The annotation process involves a single computer science student performing the initial annotations, which are then verified by two PhD students specializing in the corresponding fields. Only annotations that receive unanimous approval from all involved parties, the original annotator and two expert reviewers, are retained in the final dataset. This approval process applies to both the positive and negative document annotations, ensuring a high standard of accuracy and reliability in the annotated examples. See more guidelines to annotators in Appendix F.

### 3.3 CODING: RETRIEVING DOCUMENTATION OR SIMILAR SOLVED PROBLEMS

**Relevance**: The relevance between queries and positive documents is defined by whether the coding problem (i.e., query) either requires the corresponding syntax documentation or involves the same algorithm and/or data structure.

To solve a coding problem, programmers often need to refer to the documentation or find similar problems that share the same algorithmic design. However, given only a problem description, it is difficult to find relevant documentation or similar problems via simple keyword or semantic matching. We construct two retrieval datasets on coding, where the relevance between queries and documents is grounded in the syntax usage and algorithm design.

**Pony.** When working with a rare programming language, consulting the manual can be invaluable for understanding its syntax and function usage. However, in such cases, the problem description would likely have low semantic similarity and lexical overlap with the relevant documentation. This discrepancy necessitates intensive reasoning in identifying the particular syntax or function that is relevant to the problem at hand. We adopt a code generation dataset featuring Pony (Su et al., 2024), a rare programming language, and construct a retrieval dataset. We use the instructions of coding problems as queries $Q$, the annotated documentation about the required syntax as the positive documents $\mathcal{D}_Q^+$, and the complete language manual as the retrieval pool of documents $\mathcal{D}$, where each $D_i$ contains descriptions about syntax usage of Pony, such as conditionals, loops, and classes.

**LeetCode.** We also explore coding problems that deal with algorithms and data structures, where the goal is to retrieve problems and solutions that share the same algorithmic design. We source coding problems and solutions from LeetCode.[6] The problem descriptions are used as queries $Q$, and the positive documents $\mathcal{D}_Q^+$ are solved problems (with solutions) that were annotated as similar problems by LeetCode. Each document $D_i = (Q_i, A_i)$ from LeetCode contains a problem statement $Q_i$ and a Python solution $A_i$. To increase difficulty, we only keep questions that are grounded in real-world scenarios, where arriving at the key algorithm or data structure requires intensive reasoning. We construct a large corpus $\mathcal{D}$ by combining questions and solutions from LeetCode and Python code from CodeSearchNet (Husain et al., 2019). See Appendix B.6 for more details on the dataset.

### 3.4 THEOREM-BASED QUESTIONS: RETRIEVE SOLVED PROBLEMS USING THE SAME THEOREMS OR RELEVANT THEOREM STATEMENTS

> **Relevance**: A query (i.e., a solved problem) is relevant to a document if the document references the same theorem used in the query.

When tackling a new math or physics problem, users often reference previously solved problems or directly consult relevant theorem statements to guide their reasoning. Retrieving such similar problems or theorems can be challenging, as problems that share the same underlying logic may differ significantly in surface form, as shown in Table 30. In this setting, the query $Q$ is a theorem-based question, and the corpus $\mathcal{D}$ consists of solved STEM problems $D_i = (Q_i, A_i)$, where $Q_i$ is the problem statement and $A_i$ is its solution, or $D_i$ are theorem documents from formal theorem collections such as ProofWiki. We consider $D_i$ as a positive document if it shares the same theorem as the query's solution. The dataset is built from high-quality STEM sources (Cobbe et al., 2021; Yuan et al., 2023; Hendrycks et al., 2021; Ling et al., 2017; Chen et al., 2023a; Li et al., 2023a), and for details on corpus construction, refer to Appendix B.2.

**TheoremQA.** Derived from textbooks, online resources, and experts, TheoremQA (Chen et al., 2023b) contains questions that are based on specific mathematical or scientific theorems (e.g., the binomial theorems), and represent problems that students and other users might encounter in their studies. To ensure that the model does not simply rely on the surface-level wording of the questions, we use GPT-4[7] to rephrase the question into more concrete, applied scenarios while maintaining the same required theorem. The prompts used for rewriting the questions and an example are shown in Table 13. Human annotators carefully review the rewritten questions and make necessary revisions to ensure that they are valid and consistent with the original questions. A document $D_i = (Q_i, A_i)$ is positive if $A_i$ uses the same theorem as the query's solution. Additional details are in Appendix B.3.

**AoPS.** Math competition problems have been widely used to evaluate the problem-solving skills of students and LLMs (Hendrycks et al., 2021). Sourced from American and International Math Olympiads, these problems often require the application of advanced mathematical theorems and techniques, such as Fermat's Little Theorem or Ptolemy's theorem. To practice for the competitions, students often learn by solving other problems that require the same problem-solving skills. To

---

[6]https://leetcode.com/
[7]GPT-4 refers to the version `gpt-4-0125-preview` throughout this work.

Table 2: **The performance of retrieval models on BRIGHT.** We report nDCG@10 for all datasets: Biology (Bio.), Earth Science (Earth.), Economics (Econ.), Psychology (Psy.), Robotics (Rob.), Stack Overflow (Stack.), Sustainable Living (Sus.), LeetCode (Leet.), Pony, AoPS, TheoremQA with question retrieval (TheoQ.) and with theorem retrieval (TheoT.). Avg. denotes the average score across 12 datasets. The best score on each dataset is shown in bold and the second best is underlined. We show that reasoning-intensive retrieval is challenging for current retrievers, where the best model only achieves an nDCG@10 score of 24.3 on average. Model details are in Appendix A.1

| | StackExchange | | | | | | | Coding | | Theorem-based | | | Avg. |
|---|---|---|---|---|---|---|---|---|---|---|---|---|---|
| | Bio. | Earth. | Econ. | Psy. | Rob. | Stack. | Sus. | Leet. | Pony | AoPS | TheoQ. | TheoT. | |
| *Sparse model* | | | | | | | | | | | | | |
| BM25 | 18.9 | 27.2 | 14.9 | 12.5 | 13.6 | 18.4 | 15.0 | 24.4 | 7.9 | 6.2 | 10.4 | 4.9 | 14.5 |
| *Open-sourced models (<1B)* | | | | | | | | | | | | | |
| BGE | 11.7 | 24.6 | 16.6 | 17.5 | 11.7 | 10.8 | 13.3 | 26.7 | 5.7 | 6.0 | 13.0 | 6.9 | 13.7 |
| Inst-L | 15.2 | 21.2 | 14.7 | 22.3 | 11.4 | 13.3 | 13.5 | 19.5 | 1.3 | 8.1 | 20.9 | 9.1 | 14.2 |
| SBERT | 15.1 | 20.4 | 16.6 | 22.7 | 8.2 | 11.0 | 15.3 | 26.4 | 7.0 | 5.3 | 20.0 | 10.8 | 14.9 |
| *Open-sourced models (>1B)* | | | | | | | | | | | | | |
| E5 | 18.6 | 26.0 | 15.5 | 15.8 | 16.3 | 11.2 | 18.1 | 28.7 | 4.9 | 7.1 | 26.1 | 26.8 | 17.9 |
| SFR | 19.1 | 26.7 | 17.8 | 19.0 | 16.3 | 14.4 | 19.2 | 27.4 | 2.0 | 7.4 | 24.3 | 26.0 | 18.3 |
| Inst-XL | 21.6 | 34.3 | **22.4** | 27.4 | **18.2** | 21.2 | 19.1 | 27.5 | 5.0 | 8.5 | 15.6 | 5.9 | 18.9 |
| GritLM | 24.8 | 32.3 | 18.9 | 19.8 | 17.1 | 13.6 | 17.8 | 29.9 | **22.0** | 8.8 | 25.2 | 21.2 | 21.0 |
| Qwen | **30.6** | **36.4** | 17.8 | 24.6 | 13.2 | **22.2** | 14.8 | 25.5 | 9.9 | **14.4** | **27.8** | **32.9** | **22.5** |
| *Proprietary models* | | | | | | | | | | | | | |
| Cohere | 18.7 | 28.4 | 20.4 | 21.6 | 16.3 | 18.3 | 17.6 | 26.8 | 1.9 | 6.3 | 15.7 | 7.2 | 16.6 |
| OpenAI | 23.3 | 26.7 | 19.5 | 27.6 | 12.8 | 14.3 | **20.5** | 23.6 | 2.4 | 8.5 | 23.5 | 11.7 | 17.9 |
| Voyage | 23.1 | 25.4 | 19.9 | 24.9 | 10.8 | 16.8 | 15.4 | **30.6** | 1.5 | 7.5 | 27.4 | 11.6 | 17.9 |
| Google | 22.7 | 34.8 | 19.6 | **27.8** | 15.7 | 20.1 | 17.1 | 29.6 | 3.6 | 9.3 | 23.8 | 15.9 | 20.0 |

this end, we collect a new dataset of math competition problems, called AoPS, annotated with their respective problem-solving skills from AoPS Wiki [8]. The collected problem-solving skills are shown in Table 18. Similar to TheoremQA, we consider a solved math problem $D_i = (Q_i, A_i)$ positive if its solution uses the same problem-solving skill as the query's solution. From preliminary qualitative analysis, we find that competition problems are deliberately written in diverse ways such that it is challenging to identify the required techniques; thus, we do not rephrase the problem statements. Details are in Appendix B.5.

**Theorem retrieval.** Besides similar problems, retrieving relevant theorem definitions is also helpful. We use the queries from the aforementioned TheoremQA dataset with a different corpus $\mathcal{D}$, where each document $D_i$ is a theorem statement from ProofWiki[9]. We align theorems in theoremQA with ProofWiki documents using title matching, followed by GPT-4 verification to ensure a candidate theorem is used in the solution. We retain only queries with at least one relevant theorem statement. Manual annotation of relevance between problems and documents also showed substantial agreement (Cohen's $\kappa = 0.62$) between human annotators and GPT-4. Details are in Appendix B.4.

# 4 EXPERIMENTS

## 4.1 EXPERIMENTAL SETUP

We evaluate 13 representative retrieval models, ranging from traditional bag-of-words models to large dense retrieval models, including the top performers from the retrieval set of the MTEB leaderboard (Muennighoff et al., 2023), BEIR (Thakur et al., 2021). First, we employ BM25 (Robertson et al., 2009) as our primary sparse, lexical-based retrieval model, which demonstrates strong

---

[8] https://artofproblemsolving.com/wiki/

[9] proofwiki.org; a collection of over 20K formal definitions and proofs of mathematical theorems.

Table 4: **Question-answering results with different retrievers.** We use Claude-3.5-sonnet as the generation model and evaluate the answers with Claude-3.5-sonnet. We find that stronger retrieval typically results in better QA results, indicating the helpfulness of the annotated documents for addressing the posts in StackExchange.

| Retriever | Bio. | Earth. | Econ. | Psy. | Rob. | Stack. | Sus. | Average |
|---|---|---|---|---|---|---|---|---|
| None | 79.4 | 82.3 | 75.6 | 74.5 | 76.7 | 81.8 | 73.5 | 77.7 |
| BM25 | 78.2 | 82.6 | 76.3 | 78.2 | 76.3 | 83.0 | 73.6 | 78.3 |
| SBERT | 79.6 | 82.5 | 75.8 | 80.6 | 77.0 | 83.4 | **74.1** | 79.0 |
| Qwen | **80.2** | **83.5** | **77.0** | **81.1** | **77.2** | **85.8** | 72.6 | 79.6 |
| Oracle | *82.4* | *84.5* | *78.3* | *82.4* | *78.5* | *87.9* | *78.6* | *81.8* |

performance on BEIR (Thakur et al., 2021), comparable to that of larger trained dense retrieval models. We also evaluate a diverse set of open-source dense retrieval models: the small (<1B) models are SentenceBERT (109M; Reimers & Gurevych, 2019), BGE (335M; Xiao et al., 2023), and Instructor-Large (335M Su et al., 2022), and the large (>1B) models are Instructor-XL (1.5B; Su et al., 2022), E5-Mistral (7.1B; Wang et al., 2023a), SFR-Embedding-Mistral (7.1B; Meng et al., 2024), GritLM (7.1B; Muennighoff et al., 2024), and gte-Qwen1.5 (7.7B; Li et al., 2023b). Notably, all large dense models and Instructor-Large are instruction-tuned. Lastly, we include proprietary models from Cohere (Cohere), Voyage (Voyage AI), OpenAI (OpenAI), and Google(1.2B) (Lee et al., 2024). We provide details of each model in Appendix A.1. Following prior work (Thakur et al., 2021; Bajaj et al., 2018; Voorhees & Tice, 2000), we use nDCG@10 as the main metric. Please find the computing resources in Appendix A.2.

## 4.2 MAIN RESULTS

**Existing retrieval systems perform poorly on BRIGHT.** Results in Table 2 show that BRIGHT is very challenging, with the best model achieving only 24.3 nDCG@10. Although BM25 matches the < 1B models, it significantly underperforms larger models. This suggests that traditional keyword matching ("level 1 search") is insufficient for BRIGHT. Although larger models that have been trained on semantic-based retrieval datasets like MS MARCO (Figure 1), such as GritLM (Muennighoff et al., 2024), perform better than BM25, they are still unable to solve BRIGHT. Proprietary models perform similarly to large open-source ones. Overall, the low performance indicates that the existing retrieval system cannot perform reasoning-intensive retrieval, and new methods are required to solve "level 3 search".

**Querying with LLM reasoning steps improves retrieval performance.** Considering the strong reasoning abilities of LLMs, we propose that leveraging LLM-generated reasoning steps as queries may enhance retrieval performance. To validate this hypothesis, we prompt LLMs to write reasoning traces given a query with the following prompt: *"(1) Identify the essential problem in the post. (2) Think step by step to reason about what should be included in the relevant documents. (3) Draft an answer."*. We encourage LLMs to first understand the question by summarizing it, then use chain-of-thought reasoning (Wei et al., 2022) to identify relevant content, and finally write a candidate answer. We use then use these reasoning-enhanced as new queries to evaluate all retrieval models. We use GPT-4[10], GritLM (Muennighoff et al., 2024) and Llama-3-70B-Instruct[11] to generate reasoning steps. Figure 3 shows that using Llama-3-70B or GPT-4 reasoning steps as queries significantly improves performance compared to the original query (the detailed scores are in Tables 34 to 38). GritLM-generated reasoning steps improve BM25 performance but are less effective for other models likely due to having fewer parameters. Overall, BM25 improves the most, possibly because BM25 can adapt to different queries, while LLM-generated queries are out-of-distribution for trained models. With the best score still being below 30, significant room remains for improvement on BRIGHT.

**Retrieval augmentation boosts performance in question-answering.** An important application of retrieval is to ultimately improve the question-answering (QA) results. In addressing a StackExchange

---

[10]`gpt-4-0125-preview`
[11]`https://huggingface.co/meta-llama/Meta-Llama-3-70B-Instruct`

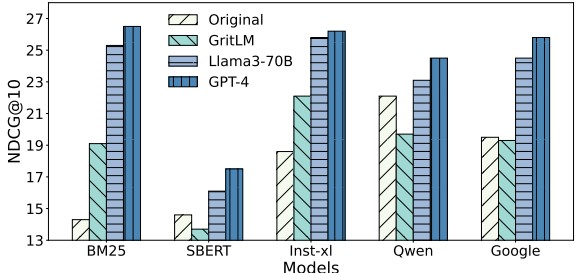

Figure 3: **Average nDCG@10 score on BRIGHT when using the original query vs. reasoning steps generated by GritLM, Llama3-70B and GPT-4 for retrieval.** Searching with LLM reasoning steps significantly improves performance. Surprisingly, BM25 achieves the best performance in the leaderboard using reasoning steps written by GPT-4 as new queries. Detailed scores are in Table 34 to 38.

| Retriever | Reranker | $k$ | nDCG@10 |
|---|---|---|---|
| BM25 | None | - | 14.3 |
| | MiniLM | 10 | 13.1 |
| | MiniLM | 100 | 8.3 |
| | Gemini | 10 | 15.7 |
| | GPT-4 | 10 | 17.4 |
| | GPT-4 | 100 | 17.0 |
| Google | None | - | 19.5 |
| | MiniLM | 10 | 16.0 |
| | MiniLM | 100 | 9.4 |
| | Gemini | 10 | 20.1 |
| | GPT-4 | 10 | 21.5 |
| | GPT-4 | 100 | 22.6 |

Table 3: **Average reranking performance on BRIGHT.** We also include the retrieval results (reranker=None) for comparison. Detailed scores can be found in Table 41 and 42.

post, one typical pipeline is to first retrieve useful documents, and then answer questions leveraging the retrieved content. While the second step could involve reasoning processes to derive an answer, we emphasize intensive reasoning in the first step to find documents. We evaluate the end-to-end QA performance of Claude-3.5-sonnet when augmented with documents retrieved by different models, and use Claude-3.5-sonnet to evaluate the answer correctness. As shown in Table 4, stronger retrievers generally result in better QA performance, with the top-performing retriever, Qwen, achieving a 1.9-point gain. However, using the oracle documents boosts performance by 4.1 points, highlighting substantial room for improvement in reasoning-intensive retrieval to enhance downstream QA results. See Appendix E for similar experiments on the TheoremQA and AoPS datasets.

Although better retrieval quality correlates with higher QA performance, the QA results may not always accurately capture retrieval performance due to the following reasons: 1) the generator model may not effectively comprehend or integrate the retrieved documents into its responses, and 2) the evaluator may fail to fully recognize the similarities and differences between two open-ended answers. Please check out Table 45 and 46 for inference and evaluation prompts used for LLMs.

## 5 ANALYSIS

### 5.1 RERANKING WITH LLMS ENHANCES RETRIEVAL PERFORMANCE

A common approach for improving retrieval results is to utilize powerful rerankers capable of performing joint computation over both the query and the documents. To this end, we investigate if performance on BRIGHT can be improved through reranking. We test this with a classical cross-encoder, MiniLM[12], and LLMs to rerank the top $k = \{10, 100\}$ retrieved documents. The cross-encoder is trained on the MS MARCO reranking task (Bajaj et al., 2016) and outputs a relevance score for each pair of query and document $(Q, D_i)$. Following Sun et al. (2023), we also rerank with LLMs by including the query and top-k documents in the prompt and asking the LLMs to order the documents based on their relevance to the query (detailed prompts can be found in Table 43). Table 3 shows that the traditional cross-encoder negatively impacts retrieval quality, with performance declining as more documents are reranked, suggesting that training rerankers on MS MARCO does not transfer well to BRIGHT. On the other hand, reranking by LLMs generally enhances performance. Stronger LLMs provide more significant improvements; for instance, based on BM25 retrieval results, Gemini-1.0 reranking increases the score by 1.4, and GPT-4 reranking enhances by 3.1 and continues to improve with higher $k$. LLMs can serve as an effective tool for reasoning-intensive retrieval, but the final results still highly depend on the underlying retrieval system.

---

[12]https://huggingface.co/cross-encoder/ms-marco-MiniLM-L-12-v2

## 5.2 ROBUSTNESS AGAINST DATA LEAKAGE FROM PRETRAINING

Many existing benchmarks suffer from inflated performance due to benchmark data contamination during large-scale pre-training (Zhou et al., 2023; Xu et al., 2024). We demonstrate that **BRIGHT** remains robust against such data leakage, even when retrieval documents are fully exposed during pre-training. To test this, we simulate a realistic scenario where language models encounter StackExchange data during internet-based training. We continue training GritLM (Muennighoff et al., 2024) on StackExchange data from our retrieval pool using both language modeling loss and contrastive learning on question-answer pairs (see Appendix A.3 for training details). This approach exposes the model to all StackExchange content in **BRIGHT**, while avoiding direct training on query-document mappings that require intensive reasoning. As shown in Table 5, the fine-tuned GritLM exhibits only a slight performance decrease, suggesting that conventional training procedures have limited impact on **BRIGHT** performance. These results demonstrate **BRIGHT**'s robustness to pre-training data leakage and highlight the need for novel approaches to advance reasoning-intensive retrieval.

Table 5: **BRIGHT is robust to massive pre-training.** By continuing training GritLM on StackExchange data without showing the mapping between queries and documents, the model does not improve the average performance after learning the in-domain knowledge, indicating the importance of reasoning capabilities in the retrieval process.

|  | Bio. | Earth. | Econ. | Psy. | Rob. | Stack. | Sus. | Avg. |
|---|---|---|---|---|---|---|---|---|
| GritLM | 25.0 | 32.8 | 19.0 | 19.9 | 17.3 | 11.6 | 18.0 | 20.5 |
| Fine-tuned GritLM | 21.1 | 25.5 | 18.8 | 30.7 | 12.7 | 12.1 | 21.9 | 20.4 |

## 5.3 LONG-CONTEXT RETRIEVAL WITH A REDUCED SEARCH SPACE IS CHALLENGING

Retrieving information from long documents is crucial for applications such as legal contracts, company financial documents, and patient notes (Saad-Falcon et al., 2024; Zhu et al., 2024). To evaluate retrieval models on reasoning-intensive tasks involving lengthy documents, we convert the StackExchange datasets to a long-context retrieval setting, where documents are complete web pages with significantly more tokens but fewer total number of documents (Table 40). With most datasets containing only a few hundred documents, nDCG@10, which evaluates the top 10 results, becomes more susceptible to randomness. Moreover, processing 10 long documents with an average length of up to 40,000 tokens is challenging for both humans and LLMs. Therefore, we decide to use recall@1 metric to provide a more reliable measure in this setting. Table 6 presents the average scores for 8 datasets from StackExchange and Pony. The highest recall achieved is 27.8, indicating that even with significantly reduced retrieval pools, the combination of long-context documents and intensive reasoning remains challenging for existing retrieval models.

Table 6: **Long-context retrieval performance where retrievers retrieve from unsplit web pages.** The results are reported as the average recall@1 score of StackExchange and Pony datasets. More detailed numbers can be found in Table 39.

| BM25 | BGE | Inst-L | SBERT | E5 | SFR | Inst-XL | GritLM | Qwen | Cohere | OpenAI | Voyage | Google |
|---|---|---|---|---|---|---|---|---|---|---|---|---|
| 11.4 | 14.8 | 18.2 | 17.4 | 25.5 | 26.0 | 17.8 | 26.0 | 27.8 | 18.4 | 21.9 | 24.6 | 22.4 |

## 6 CONCLUSION

We introduce **BRIGHT**, the first retrieval benchmark that encompasses realistic retrieval scenarios requiring intensive reasoning steps to identify relevant documents. We utilize existing online document structures and dedicate substantial human effort to curate **BRIGHT** and verify its correctness. Through extensive evaluation, we find that existing retrieval models perform extremely poorly on **BRIGHT**, with a maximum nDCG@10 score of only 24.3. Augmenting retrieval queries with reasoning steps generated by LLMs improves performance, but even the best model still achieves a score below 30. Furthermore, strong retrieval results can significantly improve downstream performance on reasoning tasks, highlighting a practical application of reasoning-intensive retrieval. We hope that **BRIGHT** will contribute to future research investigations into pushing the state-of-the-art in this direction.

## ACKNOWLEDGMENTS

We would like to acknowledge Adithya Bhaskar, Catherine Chen, Howard Chen, Tianyu Gao, Lucy He, and other members of the Princeton Language and Intelligence for their helpful feedback and discussion. We would also like to thank Qilin Liao, Yun Han, Xiaoru Teng, Cong Gao, Shengyu Wang, Xiaodong Wei, and Yan Pan for annotating and reviewing the datasets. This work is gratefully supported by an NSF CAREER award (IIS-2239290) and the Microsoft Accelerate Foundation Models Research (AFMR) grant program.

## CODE OF ETHICS AND ETHICS STATEMENT

In the process of collecting datasets for BRIGHT, we ensure that all sources come from public data, used solely for academic research and not for commercial purposes, in full compliance with the copyright rights granted by the sources. We guarantee that none of the datasets contain harmful information to society, such as racial discrimination, violence, or any private data. Our work aims to contribute to the welfare of society and humanity, and any researcher is free to use our dataset for research purposes. All the data and experiments presented in our paper adhere to the highest standards of scientific excellence, ensuring the authenticity and accuracy of the data.

## REPRODUCIBILITY

Our datasets and annotation process are introduced in Sec. 3, and the experimental settings are described in Sec. 4. Specific implementation details can be found in App. A. To facilitate the reproduction of our experiments, the code and data are provided in https://brightbenchmark. github.io/.

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

CONTENTS

## A EXPERIMENT DETAILS

### A.1 MODELS AND INSTRUCTIONS

For each model used in this paper, Table 7 provides information on the size, architecture, maximum context length of queries and documents, whether we include instructions and the specific version we use in the experiments. All parameters are set by following the official tutorial. The only exceptions are Inst-L and Inst-XL, where we empirically find that extending the maximum context length to 2048 significantly enhances the performance. In Table 8, 9, 10 and 11, We specify the instructions used for BGE, Inst-L, Inst-XL, E5, GritLM, Qwen and SFR in each dataset. For the embedding model from Google, we use the parameter "task" with the values "RETRIEVAL_QUERY" and "RETRIEVAL_DOCUMENT" to distinguish queries from documents and use the parameter "input_type" with the values "query" and "document" for the embedding model from Voyage.

Table 7: **All 13 models benchmarked in experiments.** We report the number of parameters of each model except the sparse model BM25 and proprietary models without public information. Regarding the model architecture, we distinguish between sparse and dense models and further classify dense models as encoders or decoders if known. Max $|Q|$ and Max $|D|$ denote the maximum context length we use for each model in the experiments. The instruction column indicates whether we include instructions in the retrieval. The version column denotes the specific checkpoint or implementation.

| | Size | Architecture | Max $|Q|$ | Max $|D|$ | Instruction | Version | License |
|---|---|---|---|---|---|---|---|
| Sparse model | | | | | | | |
| BM25 (Robertson et al., 2009) | N/A | Sparse | $\infty$ | $\infty$ | No | gensim[13] | LGPL-2.1-only |
| *Open-sourced models (<1B)* | | | | | | | |
| SBERT (Reimers & Gurevych, 2019) | 109M | Encoder | 512 | 512 | No | all-mpnet-base-v2 | Apache-2.0 |
| BGE (Xiao et al., 2023) | 335M | Encoder | 512 | 512 | No | bge-large-en-v1.5 | MIT |
| Inst-L Su et al. (2022) | 335M | Encoder | 2048 | 2048 | Yes | instructor-large | Apache-2.0 |
| *Open-sourced models (>1B)* | | | | | | | |
| Inst-XL (Su et al., 2022) | 1.5B | Encoder | 2048 | 2048 | Yes | instructor-xl | Apache-2.0 |
| E5 (Wang et al., 2023a) | 7.1B | Decoder | 4096 | 4096 | Yes | e5-mistral-7b-instruct | MIT |
| GritLM (Muennighoff et al., 2024) | 7.1B | Decoder | 256 | 2048 | Yes | GritLM-7B | Apache-2.0 |
| SFR (Meng et al., 2024) | 7.1B | Decoder | 4096 | 4096 | Yes | SFR-Embedding-Mistral | CC-BY-NC-4.0 |
| Qwen (Li et al., 2023b) | 7.7B | Decoder | 8192 | 8192 | Yes | gte-Qwen1.5-7B-instruct | Apache-2.0 |
| *Proprietary models* | | | | | | | |
| Cohere (Cohere) | N/A | Dense | 512 | 512 | No | Cohere-embed-english-v3.0 | Company |
| Google (Lee et al., 2024) | 1.2B | Dense | 2000 | 2000 | Yes | text-embedding-preview-0409, dimension=768 | Company |
| OpenAI (OpenAI) | N/A | Dense | 8191 | 8191 | No | text-embedding-3-large | Company |
| Voyage (Voyage AI) | N/A | Dense | 16000 | 16000 | Yes | voyage-large-2-instruct | Company |

Table 8: **Instructions used for benchmarking StackExchange datasets.** {domain} is one of Biology, Earth Science, Economics, Psychology, Robotics, Stack Overflow and Sustainable Living.

| Models | Instructions |
|---|---|
| BGE | **Query:** Represent this {domain} post for searching relevant passages: |
| Inst-L, Inst-XL | **Query:** Represent the {domain} post for retrieving relevant paragraphs: 
 **Doc:** Represent the {domain} paragraph for retrieval: |
| E5, GritLM, Qwen, SFR | **Query:** Given a {domain} post, retrieve relevant passages that help answer the post |

Table 9: **Instructions used for benchmarking the LeetCode dataset.**

| Models | Instructions |
|---|---|
| BGE | **Query:** Represent this Coding problem for searching relevant examples: |
| Inst-L, Inst-XL | **Query:** Represent the Coding problem for retrieving relevant examples: 
 **Doc:** Represent the Coding example for retrieval: |
| E5, GritLM, Qwen, SFR | **Query:** Given a Coding problem, retrieve relevant examples that help answer the problem |

Table 10: **Instructions used for benchmarking the Pony dataset.**

| Models | Instructions |
|---|---|
| BGE | **Query:** Represent this Pony question for searching relevant passages: |
| Inst-L, Inst-XL | **Query:** Represent the Pony question for retrieving relevant paragraphs: 
 **Doc:** Represent the Pony paragraph for retrieval: |
| E5, GritLM, Qwen, SFR | **Query:** Given a Pony question, retrieve relevant passages that help answer the question |

Table 11: **Instructions used for benchmarking Math datasets (AoPS and TheoremQA).**

| Models | Instructions |
|---|---|
| BGE | **Query:** Represent this Math problem for searching relevant examples: |
| Inst-L, Inst-XL | **Query:** Represent the Math problem for retrieving relevant examples: 
 **Doc:** Represent the Math example for retrieval: |
| E5, GritLM, Qwen, SFR | **Query:** Given a Math problem, retrieve relevant examples that help answer the problem |

## A.2 COMPUTING RESOURCES

We run all experiments on NVIDIA V100, A100, or H100 GPUs. The amount of time that it takes to complete one round of experiments is dependent on the model. For the sparse model, BM25, the evaluation takes less than 1 hour on CPU-only machines. For the open-sourced dense models ($< 1B$), the evaluation requires about 8 hours on one H100 GPU. For the open-sourced dense models ($> 1B$), the evaluation takes up to 36 hours on one H100 GPU. We leverage FlashAttention (Dao et al., 2022; Dao, 2024) for speedup when evaluating the dense models. For the proprietary models, the evaluation speed is dependent on the API bandwidth, but we found that one round of experiments can be completed within 2 days.

## A.3 Continual Training Setup

In Section 5.2, we introduce the continual training method GritLM on StackExchange data to evaluate whether training on in-domain data enhances the performance of **BRIGHT**. Detailed experimental settings are described in this section. Specifically, we follow GritLM to train models with two distinct objectives: a contrastive loss to maintain the model's retrieval capability and a language modeling loss to preserve the model's language generation ability. For training with the contrastive loss, we collect 3,200 (post, answer) pairs from the Biology, Earth Science, Economics, Psychology, Robotics, and Stack Overflow sections of StackExchange, and 1,538 pairs from Sustainable Living. Each post's answer is used as a positive example, with other answers serving as in-batch negatives. For training with the language modeling loss, we use both positive and negative documents from each domain within the StackExchange subsection of **BRIGHT**. These documents are split into chunks of 2048 tokens, and we sample up to 3,200 chunks for training. We use a small batch size of 64 to ensure sufficient learning steps, while following the other hyperparameters as outlined in Muennighoff et al. (2024). We continue training GritLM for 10 epochs, benchmarking the checkpoint from each epoch on the StackExchange datasets of **BRIGHT**. The detailed scores are in Table 12, where we copy the scores of GritLM to epoch=0 for easier reference. The results indicate no significant improvement across the 10 epochs, suggesting that even with intensive inclusion of StackExchange data or relevant domain knowledge in the training data of language models or retrievers, performance may not increase substantially without enhancing incorporating reasoning into the retrieval process.

Table 12: **Scores of finetuned GritLM of every epoch on StackExchange datasets of BRIGHT.** Epoch=0 indicates the performance of GritLM without further training.

| Epoch | Bio. | Earth. | Econ. | Psy. | Rob. | Stack. | Sus. | Avg. |
|---|---|---|---|---|---|---|---|---|
| 0 (GritLM) | 25.0 | 32.8 | 19.0 | 19.9 | 17.3 | 11.6 | 18.0 | 20.5 |
| 1 | 22.2 | 25.4 | 17.6 | 28.1 | 11.1 | 9.8 | 19.6 | 19.1 |
| 2 | 18.7 | 23.8 | 13.5 | 19.3 | 10.7 | 10.2 | 16.5 | 16.1 |
| 3 | 20.9 | 23.6 | 16.9 | 25.2 | 11.1 | 8.5 | 16.6 | 17.5 |
| 4 | 24.3 | 28.0 | 18.3 | 26.9 | 13.4 | 13.3 | 20.0 | 20.6 |
| 5 | 23.1 | 28.5 | 18.4 | 26.1 | 14.6 | 11.7 | 21.6 | 20.6 |
| 6 | 19.9 | 26.4 | 16.0 | 27.9 | 9.6 | 9.3 | 19.3 | 18.3 |
| 7 | 24.3 | 25.4 | 16.5 | 28.1 | 11.0 | 9.8 | 17.0 | 18.9 |
| 8 | 21.6 | 28.7 | 19.2 | 28.7 | 11.1 | 11.8 | 22.4 | 20.5 |
| 9 | 21.3 | 29.0 | 20.0 | 28.7 | 11.4 | 14.3 | 22.0 | 21.0 |
| 10 | 21.1 | 25.5 | 18.8 | 30.7 | 12.7 | 12.1 | 21.9 | 20.4 |

## B Dataset Construction

### B.1 StackExchange

**Passage split.** To split web pages or long documents into smaller pieces of passages, we employ simple heuristics with separators like two new line symbols, "#" in markdown files without additional assumptions on file structures. Although this split may not be optimal for every document, it simulates the realistic setting where long documents are automatically processed without human or expert intervention.

**False positive and false negative.** We discuss the rationale for avoiding false positives and false negatives in the data collection process. In StackExchange, the positive relevance between queries and documents is manually verified by annotators, with detailed reasoning traces to illustrate their thinking process. To avoid false negatives, the annotators only select a StackExchange post that clearly distinguishes from previously annotated examples in the same domain, e.g., different entities and semantic meanings in both posts and answers, etc. This ensures that no pair of examples share the same positive documents.

---

[13]Specifically, we use the LuceneBM25Model from gensim and the text analyzer from pyserini

## B.2   STEM QUESTION AND SOLUTION CORPUS FOR THEOREMQA AND AoPS

In this subsection, we describe the construction of the STEM question and solution corpus, which is used for both TheoremQA Questions and AoPS. We source the documents (pairs of problem statements and solutions) $D_i = (Q_i, A_i)$ from existing datasets—GSM8K (Cobbe et al., 2021), GSM8K-RFT (Yuan et al., 2023), MATH (Hendrycks et al., 2021), AQuA-RAT (Ling et al., 2017), TheoremQA (Chen et al., 2023a), and CAMEL-Math (Li et al., 2023a). To reduce the likelihood of false negatives among the STEM corpus, we leverage the metadata from the original datasets to exclude specific documents from the corpus for each test query. For example, CAMEL-Math contains problem-solution pairs labeled with the category "Calculus", which covers different questions that involve derivatives and integrals. Therefore, for queries in TheoremQA that uses "derivative chain rule" or "integral rules", we excluded CAMEL-Math pairs in the category "Calculus" to reduce possible false negatives. Thus, for each test query in TheoremQA-Q and AoPS, we manually decide which labels in the other datasets to exclude based on the metadata. We do not exclude any problem-solution pairs from GS8K, GSM8K-RFT, or AQuA-RAT due to the relative elementary difficulty (mostly basic algebra questions) in comparison to our test queries, which leverages more advanced theorems and techniques. The mapping from the test query category to the excluded problem-solution categories can be found in Table 14, 15, and 16.

An alternative approach to excluding false negatives from the corpus for each test query is to inspect every problem-solution pair and annotate if they are relevant to test query. Although this would yield harder negatives and additional positives, we opt to not to use this approach due to its expensive cost to conduct annotation between every test query and possible candidates.

## B.3   THEOREMQA: REPHRASING QUESTIONS INTO SPECIFIC SCENARIOS

TheoremQA is a dataset consisting of theorem-driven questions in mathematics, physics, finance, and computer science and electrical engineering (Chen et al., 2023a). For each question in TheoremQA, we refer it to MathInstruct dataset[14] (Yue et al., 2023), as each question in this dataset is annotated with the reasoning steps and final answers.

From preliminary analysis, we found that TheoremQA questions are often written in a way such that the theorem used to solve the problem is explicitly mentioned in the question. As a result, questions that use the same theorem can have high keyword overlap, which means retrievers can easily retrieve the correct document by matching the keywords. Thus, we rewrite the questions in TheoremQA by grounding them in real-world scenarios or applications, which makes the reasoning steps less explicit and provides more diverse questions. We leverage GPT-4 with manually written instructions and in-context demonstrations to rewrite the queries $Q$. We provide the prompt used to rewrite TheoremQA questions and an example in Table 13. After rewriting the question, the authors manually inspect each rewritten question to ensure that the question is solvable and consistent with the original question (i.e., the reasoning steps and final answer still hold). When applicable, we manually edit the rewritten question to improve the fluency and coherence of the question, and discard the query if the rewritten question is not solvable or consistent with the original question. Consequently, we obtain 206 rewritten questions in TheoremQA from the original set of 800 questions.

## B.4   THEOREMQA: ANNOTATING RELEVANT THEOREMS

For TheoremQA-Theorems, we use the test queries from TheoremQA-Questions and annotate them with useful theorem proofs and definitions. We source mathematical theorem proofs and definitions from ProofWiki, which is community-driven effort with more than 20K formal definitions and proofs of mathematical theorems. ProofWiki is preprocessed and provided by MathPile (Wang et al., 2023b). We opt to map the original theorem names to the documents in ProofWiki so that the gold documents have consistent forms with other documents in the corpus.

For each test query, we first construct a candidate set of useful documents from ProofWiki using the theorem name and definition provided by the original TheoremQA dataset. Specifically, we construct the candidate set with the following steps:

---

[14]https://huggingface.co/datasets/TIGER-Lab/MathInstruct

1. Find documents where the theorem name exists as a substring. We discard this set if there are more than 100 such documents, which typically means that the theorem name is too common.

2. Using the theorem name and definition from the original dataset as the query, we use BM25 to retrieve the top $k = 10$ documents from ProofWiki.

Then, we prompt GPT-4 (`gpt-4-0125-preview`) to check if each document's described theorem are used in the problem solutions, which labels each candidate as either a positive or negative document. The prompt for this step can be found in Table 17. The authors manually annotated 50 instances and found a substantial agreement of Cohen's $\kappa = 0.62$ with the model judgments. Finally, we keep test queries with at least one positive document.

## B.5   AoPS: Connecting AoPS problems to the MATH dataset

AoPS Wiki is a community-driven platform where users can post problems and solutions to math competition problems. These math competitions include, but are not limited to the American Mathematics Competitions (AMC), the American Invitational Mathematics Examination (AIME), and the International Mathematical Olympiad (IMO). In addition to problems, the AoPS Wiki also contains articles on various topics in mathematics, such as Fermat's Little Theorem and Ball and Urns. These articles not only describe the theorem or technique but also link to problems that can be solved by them. We browse the AoPS Wiki and collect the topics and the linked problems. The topics are listed in Table 18.

Although math competition problems are used in previous datasets, such as MATH (Hendrycks et al., 2021), they lack the necessary annotations on the problem-solving skills to construct positive documents. Thus, we opt to collect these annotations from AoPS Wiki instead.

Furthermore, since MATH examples are used in the STEM corpus, we deduplicate them by matching the collected problems with the MATH instances. Specifically, for each question $Q$ we collected from AoPS, we find the closest problem statement in MATH using n-gram overlap, and manually check if they are the same problem. If the same problems are found, we merge them into one instance, otherwise, we create a new instance and insert it into the corpus.

## B.6   LeetCode

We first obtain the publicly available LeetCode[15] questions from HuggingFace[16]. Our retrieval pool is sourced from a combination of LeetCode and CodeSearchNet, including a problem description and a solution in each example. In the following sections, we outline the process for constructing positive examples and performing additional checks to minimize the likelihood of false negatives while ensuring the use of a large retrieval pool.

**Using similar questions as positive examples.** For each question, we obtain the gold pair annotations from the "Similar Questions" field, which contains links to other LeetCode questions that are similar to the problem. While the website does not explicitly describe the guidelines behind how this field is populated, our qualitative analysis showed that these questions have a high overlap in terms of the data structure, algorithms, and/or logical reasoning used to solve the problem.

**Select problems based on real-world scenarios to avoid false negatives from CodeSearchNet.** From a preliminary qualitative analysis, we discovered that some questions in LeetCode are frequently found in CodeSearchNet due to the popularity of certain algorithms and the simplicity of their problem statements. Examples include implementing sorting algorithms or merging two linked lists, which could unexpectedly introduce false positives into this retrieval setup. Thus, we use an additional filtering step—we only keep questions that are grounded in real-world concepts that are not as commonly used in the context of coding problems. The intuition behind this is similar to the TheoremQA annotations process: the reasoning steps (i.e., the algorithms and data structures used to solve the problem) cannot be as easily deciphered as if the problem statement clearly describes the algorithm and data structure used. To this end, we first manually write instructions with six-shot in-context learning demonstrations, and use GPT-4 (`gpt-4-0125-preview`) to classify all LeetCode

---

[15] https://leetcode.com/
[16] https://huggingface.co/datasets/greengerong/leetcode

questions. Then, we validate the GPT-4 judgment with 80 annotations by the authors. Authors' and GPT-4's annotations have a Cohen's kappa of 0.73, which suggests substantial agreement. The prompt used in this step and examples can be found in Table 19. GPT-4 judged a total of 291 samples of being grounded in real-world concepts, and we randomly sample 142 questions from this set to construct our test set.

**Remove examples with high topic overlap from the pool to avoid false negatives in LeetCode.** To avoid potential false negatives that are not annotated by the LeetCode website, we leverage the "Topics" field from the website, which contains information about the algorithms and data structure used in the problem, such as "stack" and "breath first search". For each question $Q$, we collect its topics $T(Q) = \{t_q^1, \ldots, t_q^m\}$ from the LeetCode website, where $t_q$ denotes the $m \geq 1$ different topics assigned to the question by the website. Since each question may have multiple tags, we exclude other questions that have a high overlap with test questions from the corpus for that specific question. Specifically, we exclude question $Q'$ from the corpus for test query $Q$ if $\frac{|T(Q) \cap T(Q')|}{|T(Q)|} \geq 0.5$, because more than half of the topics used in $Q'$ are also used in $Q$. This means that the two questions can be highly related in reasoning steps. Overlap smaller than the 0.5 threshold means that $Q'$ is unlikely to be related, and thus a false negative, to the test question $Q$. Finally, we construct the rest of the corpus from CodeSearchNet (Husain et al., 2019)[17]. We only consider the Python functions as the solutions to the LeetCode questions are all in Python.

## C  DATA EXAMPLES

In Table 20, 21, 22, 23, 24, 25, 26, 27, 28, 29, 30, and 31, we show more examples in **BRIGHT**.

## D  REASONING CATEGORIZATIONS

To better understand the reasoning capabilities required in **BRIGHT**, we summarize 4 reasoning types with representative examples from each of the 12 **BRIGHT** datasets.

- Deductive reasoning: The document usually describes a general principle or theorem that could be applied to explain a specific scenario or solve a specific problem present in the query. For example, in Table 20, the general mechanism of meristem is applied to explain the scenario where a tree trunk could sprout and grow after being cut. (See more examples in Table 21, 25, 22, 31).

- Analogical reasoning: The document draws parallel with the query in underlying logic, which indicates that the method used in the document could be also used to solve the query problem. For example, in Table 29, although the query problem appears to be different from the one in the documents, they share similar underlying logic and can be both solved by the Chicken McNugget Theorem (See more examples in Table 27, 30).

- Causal reasoning: The document and the query present a cause-and-effect relationship, i.e., To fix the problem in the query, we need to find the cause in the document. For example, in Table 24, the query presents a problem where the debug message is missing, which is caused by the parameter settings in the source code (positive documents).

- Analytical reasoning: The document provides critical concepts or knowledge that support reasoning chains to solve problems in queries. For example, in Table 26, one will need to first analyze the problem in the query: To determine whether reusing water for plants is beneficial, we need to first understand where the water comes from and what could be contained in the water; As described in the query, the water comes from watering, which goes through plants and soil and finally arrives the plates below the pots; This indicates that the water could carry minerals, soluble salts and other materials in soil and plants. The document provides critical knowledge that soluble salts could be dissolved and accumulate in the water, which will be harmful to plants. This piece of information helps to complete the reasoning chain for deriving the final answer to the query. (See more examples in Table 23, 28).

---

[17]https://huggingface.co/datasets/code_search_net

With categorized reasoning capabilities, people who use BRIGHT can now understand better about their achievements if they observe improvements on specific datasets. We hope that this will facilitate future research on more systematically developing retrieval models with strong reasoning capabilities.

## E    DOWNSTREAM PERFORMANCE

In this work, we evaluated the effect of retrieval on downstream tasks. In this subsection, we continue the investigation into the downstream performance on the theorem-based tasks—TheoremQA Questions, TheoremQA Theorems, and AoPS—by evaluating `gpt-4o-2024-08-06` on these tasks. We run with with three types of documents in context: none (closed-book), random (randomly sampled document from the corpus), and oracle (the annotated corpus). We use a temperature of 0.2 and top-p of 0.9, and average the results across 5 different random seeds. We show the results in 47, and find that the oracle document can consistently improve the performance across all datasets. This improvement is statistically significant in TheoremQA Questions and TheoremQA Theorems, suggesting the benefits of using strong retrieval results.

## F    ANNOTATOR INSTRUCTIONS

In this section, we describe the instructions for annotators to collect data in **BRIGHT**.

### F.1    STACKEXCHANGE

1. Browse posts from the newest to the oldest.

2. Discard posts without an answer accepted by the user or obtains more than 5 votes

3. Discard answers of posts without URL links.

4. For each link in the answer, write down the answers to: (1). why are the document and the post relevant; (2). what is the reasoning required to understand the relevance between the post and the document. If these are not possible, discard the link.

5. Check whether the linked documents provide critical information to understand the post or address the questions. The relevance could include explaining critical concepts or details or providing theorems, lemmas or code pieces that would contribute to solving the problems. Refer to examples for a better understanding on the relevance (Examples in Table 20 to 26)

6. Use LLMs (e.g., ChatGPT, Claude, etc.) to generate post keywords, or use the post title to search for web pages with large keyword or semantic overlap in Google. Search for at most 5 negative web pages per query. Ensure that hard negatives are totally unhelpful in addressing the post. Common hard negatives include descriptions or documentation of similar contexts but different problems, or code pieces that share the same variable names but are unrelated to the post.

7. Check all the other negative passages, which include positive and negative documents in previously annotated examples in the same dataset. Ensure that all those documents and current problems focus on different sub-topics, and thus totally irrelevant.

8. Split every web page into small passages either by two newline symbols, "#" in markdown files or fixed-length tokens, and fine-grain positive ones following criteria in step 4 and 5.

### F.2    THEOREMQA

In TheoremQA, the main task for the annotator is to check if the GPT-4 rewritten questions are valid. The specific instructions are as follows:

1. Read the rewritten question and determine if it is solvable.

2. If it is solvable, read the original question and solution, and determine if the rewritten question is consistent with the original question. That is, the same reasoning steps and the final answer should hold.

3. If it is also consistent, mark the question as valid, and make any minor edits to the problem statement (e.g., to improve grammar or fluency) as you see fit.

4. If it is not solvable or not consistent, read the original question and solution, and correct the rewritten question if possible. If not, then discard the problem.

### F.3 AoPS

In AoPS, annotators are tasked to find questions from the AoPS Wiki and record the problems:

1. Browse through the AoPS Wiki and find topic/category pages (example 1, example 2).

2. Look through each page and find pages specific theorems or techniques that can be used to solve problems. The page should link to at least two competition problems (example 1, example 2).

3. Record the links of both the theorem/technique as well as the problem pages.

The annotators are assigned a category to look for theorems in to avoid overlaps, and the categories are {algebra, geometry, calculus, probability, number theory, other}. After all links are collected, we use a web scraper to collect the problem statement and solutions, and we manually check the quality of the scraped data.

### F.4 LeetCode

In LeetCode, annotators determine whether a question is grounded in real-world concepts. We give a similar instruction to the annotator as to GPT-4:

1. Read the problem statement carefully.

2. Categorize the question into one of three categories:
   - 0: The question is not grounded in any real-world concepts. The description only uses coding-specific terms, such as "linked list", "binary search", "palindrome", "sorting", etc..
   - 1: The question is not grounded in any real-world concepts or real-world concepts that are commonly used in the context of coding, such as needle in a haystack, strings/words, or a spiral matrix.
   - 2: The question is grounded in real-world concepts that are not commonly used in the context of coding, such as building height, planting trees, or games. It may still use some code-specific terms to specify the data structure involved.

### F.5 LLM usage

There is no specific procedure for the annotators to use LLMs. The LLMs serve a tool to help annotators understand queries and documents, i.e., whenever they fail to understand something, they ask LLMs for clarification, explanation, etc. They are also used to summarize the content of StackExchange posts for searching negative documents in Google for StackExchange. To diversify the search results, the annotator prompts LLMs with role-playing scenarios (e.g., as a biology student) to generate keywords of the posts. The responses from LLMs are not included in any of **BRIGHT** datasets. Since the responses from LLMs are not guaranteed to be correct, the annotators always search for trustworthy sources to verify the information. The LLMs used in the annotation include ChatGPT[18], GPT-4, and Claude-3.

### F.6 Sensitive information

All the data in **BRIGHT** are manually collected, carefully verified, and reviewed to remove any personally identifiable information or offensive content.

---

[18]https://chatgpt.com/

# G LIMITATIONS AND FUTURE WORK

One limitation of this work is that the judgment about relevance between queries and documents is subjective. Even if the StackExchange answers are accepted by the users or obtain high votes, it is not guaranteed that everyone will agree the referenced documents are relevant. We may not expect human retrieval results to be exactly the same as our annotation. However, we mitigate this issues by using multiple annotators and only retain the queries in which all annotators, including domain experts, agree on the relevance. Therefore, we believe that the relevance judgment in **BRIGHT** is reliable and consistent.

Aside from retrieval, other embedding tasks such as clustering may also require reasoning. We have not addressed multi-modal settings in this paper, but they represent intriguing avenues for future exploration.

# H POTENTIAL SOCIAL IMPACTS

This paper presents a new retrieval benchmark that features relevance beyond lexical and semantic similarity and requires intensive reasoning to solve. There are many potential societal consequences of our work, e.g., improving search algorithms, fostering better information access, developing more advanced retrieval models, etc. It could also promote collaboration among researchers and facilitate the development of more effective search engines, ultimately benefiting society by enhancing the way people find and access information.

# I COMPARISON TO RAR-B

While both the *Reasoning as Retrieval Benchmark* (RAR-b) and *BRIGHT* evaluate retrieval systems on their reasoning abilities, they differ significantly in their approach and objectives:

- **Dataset Construction:**
    - *RAR-b*: Adapts existing multiple-choice benchmarks into a retrieval format, where the queries are original questions and the corpus consists of unique options from all the questions.
    - *BRIGHT*: Purposefully built as a retrieval benchmark which uses queries and documents in realistic retrieval scenarios.
- **Document Characteristics:**
    - *RAR-b*: Often uses very short sequences (<10 words) derived from multiple-choice options.
    - *BRIGHT*: Focuses on substantially longer documents (>100 tokens), more closely mirroring practical retrieval scenarios.
- **Practical Relevance:**
    - *RAR-b*: It's more conceptual than practical, as real-world retrieval typically involves retrieving documents instead of answers, making this test more about abstract reasoning abilities.
    - *BRIGHT*: Designed to reflect real-world information seeking behaviors and needs.

We believe both datasets effectively assess retrievers' capabilities in handling reasoning-intensive queries, and we'll be adding clearer comparisons of RAR-b in our revision.

Table 13: **Prompt used to rewrite TheoremQA questions.** Blue text denote instance-specific inputs. Violet text denotes the rewritten question and answer outputted by GPT-4. For each question, we provide the original question and answer, as well as the theorem name and theorem definition from the original TheoremQA dataset. The instruction prompts the model to rewrite the question with a different surface form without changing the reasoning steps or the final answer. We hand-write three examples to illustrate the rewriting process. We also allow the model to skip the question.

```
Rewrite the following question such that the logical steps and final
answer are still the same, only change the surface form of the question.
Avoid using the words related to the theorem used in the question.  You
can do this by using real-world examples and applications to illustrate
the concepts, in a way that is easier to understand for a layman.  Try
to ground the question in a real-world context such as finance and
engineering problems, but be creative and feel free to use any domain
you like!  However, the question should be standalone and solvable.
Rewrite the question in a json format, with the fields "question" and
"answer".  The "theorem" field indicates the theorem used in the
question, but you should not use the words related to the theorem in the
rewritten question.  If you do not think that the question cannot be
rewritten in layman's terms in a standalone fashion, then you can write
a json object with the field "skip" set to True.

{ "question":  "In a group of 10 people, each of whom has one of 3
different eye colors, at least how many people must have the same eye
color?", "answer":  "4", "theorem":  "pigeonhole principle",
"theorem_definition":  "The Pigeonhole Principle is a fundamental
concept in combinatorics, a branch of mathematics that deals with
counting and arranging objects.  [additional text omitted...]" }
one demonstration was omitted...
{ "question":  "dy/dt = \sqrt{t}, y(1) = 1.  What is y(4)?", "answer":
"5.667", "theorem":  "integral rules", "theorem_definition":  "Integral
rules in calculus are a set of techniques [additional text omitted...]
" }
{ "question":  "You're tracking the growth of a plant from a seed.  The
rate at which the plant grows in height is equal to the square root of
the number of days since you planted it.  One day after the seed was
first planted, the plant was 1 inch tall.  How tall will the plant be
after 4 days?", "answer":  "5.667" }

{ "question":  "Given image \begin{tabular}{|llll|} \hline 7 & 1 & 6 & 0
 \\ 3 & 3 & 7 & 6 \\ 6 & 6 & 5 & 7 \\ \hline \end{tabular} , and the bit
-depth of the image is 4. Suppose you want to use the thresholding
technique to segment the image. What is the appropriate threshold value
based on the histogram of the image? Follow the following rule when you
do thresholding or grouping: pixel $(i, j)\in$ Group A pixels if $g(i, j
)\leq$ current threshold $\mathrm{T}$; pixel $(i, j)\in$ Group B pixels
otherwise, where $g(i, j)$ is the intensity value of pixel $(i, j)$.",
"answer":  "4", "theorem":  "image contrast ", "theorem_definition":
"Image contrast in signal processing refers to the difference in
intensity or color between various elements or regions within an image.
[additional text omitted...]  " }

{ "question":  "Imagine you have a collection of 12 unique stickers,
each with a number from 0 to 7 representing its rarity level.  You're
planning to divide these stickers into two groups based on their rarity
to make it easier for collectors to understand.  You decide to use a
method where stickers with a rarity level at or below a certain number
go into the 'Common' group, and those above this number go into the
'Rare' group.  Given the distribution of stickers' rarity levels as
follows:  two stickers each of levels 7, 6, and 5; three stickers of
level 3; and one sticker each of levels 1 and 0.  What is the rarity
level that should be used as the cutoff to divide the stickers into the
'Common' and 'Rare' groups, ensuring a balanced understanding of
rarity?", "answer":  "4" }
```

Table 14: **TheoremQA-Q Theorem names and the excluded topic names from CAMEL-Math.**

| TheoremQA-Q Theorem | CAMEL-Math Topics |
|---|---|
| acyclic graph | Graph theory, Combinatorics |
| binomial theorem | Combinatorics, Algebra |
| catalan-mingantu number | Combinatorics |
| cauchy's integral theorem | Complex analysis |
| cayley's formula | Graph theory, Combinatorics |
| convexity | Optimization |
| cramer rao lower bound | Statistics |
| definite matrix criteria | Algebra, Linear algebra |
| derivative chain rule | Calculus |
| double integral theorem | Calculus |
| eigenvalues and eigenvectors | Linear algebra |
| euler's method | Calculus, Numerical analysis, Differential equations |
| euler's theory | Graph theory, Combinatorics |
| expected utility | Game theory |
| fisher information | Statistics |
| fourier's theorem | Fourier analysis |
| gauss's lemma | Number theory, Algebra |
| integral rules | Calculus |
| intermediate value theorem | Calculus |
| isomorphisms | Group theory, Algebra |
| jensen's inequality | Statistics, Probability |
| l'hôpital's rule | Calculus |
| limit laws for sequences | Calculus |
| limiting theorem | Calculus |
| linear independence | Algebra, Linear algebra |
| linear subspaces | Algebra, Linear algebra |
| linear systems | Algebra, Linear algebra |
| liouville's theorem | Complex analysis |
| martingale | Statistics, Probability |
| matrix determinant formula | Algebra, Linear algebra |
| maximal planar graph | Graph theory, Combinatorics |
| maximum entropy | Statistics, Probability |
| message passing algorithm | Graph theory, Combinatorics |
| multinomial theorem | Combinatorics |
| newton-raphson method | Calculus, Numerical analysis |
| order | Group theory, Algebra |
| ordinary differential equation | Calculus, Differential equations |
| p-value | Statistics |
| pigeonhole principle | Combinatorics |
| poisson process | Statistics, Probability |
| projection theory | Algebra, Linear algebra |
| ramsey's theorem | Graph theory, Combinatorics |
| rolle's theorem | Calculus |
| series convergence | Calculus |
| shortest path | Graph theory, Optimization, Combinatorics |
| similarity | Geometry |
| squeeze theorem | Calculus |
| stirling number of the first kind | Combinatorics |
| stirling number of the second kind | Combinatorics |
| t-test | Statistics |
| taylor's approximation theorem | Calculus |
| trapezoidal rule | Calculus, Numerical analysis |
| vertex cover | Graph theory, Combinatorics |
| viterbi algorithm | Statistics |
| wave theorem | Differential equations |

Table 15: **Subfield name in the original TheoremQA dataset and their corresponding categories in BRIGHT. Each category is used to specify the excluded problem-solution pairs from MATH and AoPS.**

| Name | Category |
|------|----------|
| Algebra | algebra |
| Atomic physics | others |
| Calculus | calculus |
| Celestial mechanics | others |
| Classic mechanics | others |
| Combinatorics | number theory |
| Complex analysis | calculus |
| Computer networking | others |
| Condensed matter physics | others |
| Derivatives | calculus |
| Economics | others |
| Electromagnetism | others |
| Equity investments | others |
| Fixed income | others |
| Fluid mechanics | others |
| Functional analysis | calculus |
| Geometry | geometry |
| Graph theory | others |
| Group theory | algebra |
| Information theory | others |
| Kinetics | others |
| Machine learning | others |
| Mathematical analysis | calculus |
| Number theory | number theory |
| Numerical analysis | calculus |
| Optics | others |
| Particle | others |
| Portfolio management | others |
| Probability theory | probability |
| Quantitive methods | others |
| Quantum | others |
| Real analysis | calculus |
| Relativity | others |
| Signal processing | others |
| Statistical physics | others |
| Statistics | others |
| Stochastic process | probability |
| Thermodynamics | others |
| Wave | others |

Table 16: **AoPS theorems and techniques and the excluded topic names from CAMEL-Math and TheoremQA.**

| AoPS Theorem | CAMEL-Math Topics | TheoremQA Theorems |
|---|---|---|
| Binomial Theorem | Combinatorics, Algebra | binomial theorem, multinomial theorem |
| Vietas Formulas | Algebra | vieta's formula, birg-vieta's theorem |
| Ptolemys theorem | Geometry | properties of kites, similarity, rhombus, rectangle, quadrilateral, triangle, isosceles triangle, parallelogram |
| Recursive Series | Calculus, Algebra | |
| Power of a Point | Geometry | similarity |
| Ball and urn | Combinatorics | |
| Newtons Sums | Algebra | vieta's formula |
| Probability | Probability, Statistics | probability |
| Fibonacci sequence | Combinatorics, Number theory | |
| Chicken McNugget Theorem | Combinatorics, Number theory | |
| Central Tendency | Statistics | |
| Principle of Inclusion Exclusion | Combinatorics, Set theory | inclusion-exclusion principle |
| Factorial | Combinatorics | |
| Picks Theorem | Geometry | similarity, triangle, isosceles triangle, parallelogram, rhombus, quadrilateral, rectangle, triangle midsegment theorem |
| Shoelace Theorem | Geometry | rhombus, rectangle, quadrilateral, triangle, isosceles triangle, parallelogram |
| Geometric probability | Geometry, Probability | |
| Euclidean algorithm | Number theory, Cryptography, Algebra | euclidean algorithm |
| Mass Points | Geometry | similarity, triangle, isosceles triangle, parallelogram, rhombus, quadrilateral, rectangle, triangle midsegment theorem |
| Geometric Series | Algebra | |
| Triangle Inequality | Geometry | inequalities, triangle, isosceles triangle |
| Simons Favorite Factoring Trick | Number theory, Algebra | |
| Properties of Logarithms | Algebra | |
| Fermats Little Theorem | Number theory, Cryptography | fermat's little theorem, euler's totient theorem |

Table 17: **Prompt used to check TheoremQA-Theorem documents.** Blue text denote instance-specific inputs. For each question, we provide the problem statement, the solution, the theorem name, the theorem definition, and the document text from a ProofWiki document.

```
Instruction:  Determine if the given text can be helpful in
understanding and solving the given problem.  Use the theorem, its
definition, and the problem solution to help you make a decision.  The
text can be helpful if it uses very similar reasoning steps as the
solution and applies the theorem in a related way as the solution
applies the theorem to solve the problem.  The text should be able to
assist a student who is not familiar with the theorem in ultimately
solving the problem.

The input is given to you in a json format with the following keys:
Problem statement:  The problem statement that the student is trying to
solve.
Solution:  The solution to the problem.
Theorem:  The theorem that is used in the solution.
Theorem definition:  The definition of the theorem.
Text:  The text that you need to evaluate.
Think step by step and reason about the theorem and the text first
before finally making a decision.  Output True if the text is helpful,
and False if it is not.

{ "Problem statement":  "", "Solution":  "", "Theorem":  "", "Theorem
definition":  "", "Text":  "", }
Now, write your answer in the following format:
Reasoning:  [your reasoning here]
Answer:  [True/False]
```

Table 18: **Theorems and techniques used in the AoPS dataset, and their corresponding subfield categories.**

| Name | Category |
|---|---|
| Ball and urn | number theory |
| Binomial Theorem | algebra |
| Central Tendency | others |
| Chicken McNugget Theorem | algebra |
| Euclidean algorithm | number theory |
| Factorial | number theory |
| Fermat's Little Theorem | number theory |
| Fibonacci sequence | number theory |
| Geometric Series | algebra |
| Geometric probability | probability |
| Mass Points | geometry |
| Newtons Sums | algebra |
| Picks Theorem | geometry |
| Power of a Point | geometry |
| Principle of Inclusion Exclusion | number theory |
| Probability | probability |
| Properties of Logarithms | algebra |
| Ptolemy's theorem | geometry |
| Recursive Series | algebra |
| Shoelace Theorem | geometry |
| Simon's Favorite Factoring Trick | number theory |
| Triangle Inequality | algebra |
| Vieta's Formulas | algebra |

Table 19: **Prompt used to check LeetCode questions.** Blue text denote instance-specific inputs. Violet text denotes the output from GPT-4. For each question, we provide the title and the problem statement. We use six in-context demonstrations, and the instruction prompts the model to categorize the question based on the criteria provided. Although there are three possible labels, we only keep questions that were labeled as 2, which are grounded in real-world concepts. For sake of brevity, we only show one example, but other examples can be found on the code repo.

```
Read the coding question and categorize it using the following criteria:
0:  The question is not grounded in any real-world concepts.  The
description only uses coding-specific terms, such as "linked list",
"binary search", "palindrome", "sorting", etc..
1:  The question is not grounded in any real-world concepts or
real-world concepts that are commonly used in the context of coding,
such as needle in a haystack, strings/words, or a spiral matrix.
2:  The question is grounded in real-world concepts that are not
commonly used in the context of coding, such as building height,
planting trees, or games.  It may still uses some code-specific terms to
specify the data structure involved.

You should only consider the initial problem statement and problem
title, not the examples or constraints.
Use the following examples to help you categorize the question:

Example 1:
{{
"title":  "Merge Two Sorted Lists",
"question":  "You are given the heads of two sorted linked lists `list1`
and `list2`.
Merge the two lists in a one **sorted** list.  The list should be made
by splicing together the nodes of the first two lists.

Return _the head of the merged linked list_.

**Example 1:**
rest of the example omitted...
}}

{{
"label":  0
}}

Examples 2-6 are omitted for brevity...

Now, consider the question below and categorize it using the criteria
above.  Output your answer in a json format:
{{
"title":  "Container With Most Water",
"question":  "You are given an integer array `height` of length `n`.
There are `n` vertical lines drawn such that the two endpoints of the
`ith` line are `(i, 0)` and `(i, height[i])`.

Find two lines that together with the x-axis form a container, such that
the container contains the most water.

Return _the maximum amount of water a container can store_.
rest of the question omitted..."
}}

{ "label":  2 }
```

Table 20: **Biology example.** The positive document explains the concept of meristem, which is the mechanism by which plant cells develop into other tissues and organs. This provides critical support for why a tree chunk sprouts and grows after being cut.

---

*Query*

---

After I cut trees into logs and remove the branches in winter, they start growing. They sprout out and grow completely normal looking stems and leaves and maintain them all summer. The sprouts mostly appear around the cut branches. Sometimes they last all winter and grow for another year. How does a tree trunk sprout and grow after being cut?

---

*Chain-of-thought reasoning to find documents*

---

The essential question is to figure out the growing mechanism of trees after being cut. It is usually related to cell division, which later develops into tissues and organs. We need to find relevant information of these cells in plants and their growing capabilities after being cut.

---

*Example positive document*

---

Meristem
Tunica-corpus model of the apical meristem (growing tip). The epidermal (L1) and subepidermal (L2) layers form the outer layers called the tunica. The inner L3 layer is called the corpus. Cells in the L1 and L2 layers divide in a sideways fashion, which keeps these layers distinct, whereas the L3 layer divides in a more random fashion.
In cell biology, the meristem is a type of tissue found in plants. It consists of undifferentiated cells (meristematic cells) capable of cell division. Cells in the meristem can develop into all the other tissues and organs that occur in plants. These cells continue to divide until they become differentiated and lose the ability to divide.
There are three types of meristematic tissues: apical (at the tips), intercalary or basal (in the middle), and lateral (at the sides also known as cambium). At the meristem summit, there is a small group of slowly dividing cells, which is commonly called the central zone. Cells of this zone have a stem cell function and are essential for meristem maintenance. The proliferation and growth rates at the meristem summit usually differ considerably from those at the periphery.
...
Under appropriate conditions, each shoot meristem can develop into a complete, new plant or clone. Such new plants can be grown from shoot cuttings that contain an apical meristem.
...

---

*Example negative document*

---

Cutting firewood from felled trees safely and easily (bucking)
There are loads of articles on the web showing how (and sometimes how not to!) cut down trees, but not so much about actually cutting firewood after the tree has been felled and stripped of its branches (called snedding or limbing in arborist speak). This 'how to' shows the easiest way to produce firewood by cutting many logs at the same time. I actually use my firewood cutting operation as part of my fitness regime, much more useful than going to the gym and the same muscle burn the day after!
I start by cutting the trees into 10' (3m) logs so that I can easily drag them to my cleared working area close to the pick up point.
You can make a homemade 'cradle' that holds a whole bunch of logs from sturdy construction lumber (2 by 4). This enables you to safely cut many at once, as long as you put some of the heavy logs both at the bottom and on the top. Don't forget to make the cradle narrow enough for the size of chainsaw bar you have. Usually not more than 12" or 300mm inside measurement. Make sure that it is constructed in such a way that you cannot hit any nails etc with the chainsaw.
Place the logs onto the cradle keeping the ends off the logs flush one side ( or at least staggered in 12" steps). If you are fussy like me you can mark one of the top logs at 12" (300mm) intervals with chalk to ensure even log lengths. Then cut alternate sides (so that it doesn't topple over to one side), finishing off with a couple of cuts over the cradle itself, don't forget to stop before you saw the thing in half lol! (See upgraded version below...)
...

---

Table 21: **Earth Science example**. The positive document describes the Airy's isostasy model, which is needed in calculating the hydrostatic equilibrium.

| |
|---|
| ***Query*** |
| How to calculate hydrostatic equilibrium? |
| I'm trying to solve the following problem. The sea level in the past was 200 m higher than today. The seawater became in isostatic equilibrium with the ocean basin. what is the increase in the depth x of the ocean basins? Water density is $p_w = 1000$ kg $m^{-3}$, and mantle density is 3300 kg $m^{-3}$ |
| Using the compensation column, I reach:
$x = (p_w * 200m)/3300 = 60.60$ m
but normally I expected to find 290 m.
Can someone explain to me what's wrong? |
| ***Chain-of-thought reasoning to find documents*** |
| This problem is about the hydrostatic equilibrium. This may be related to Pascal's law, which gives insights on the pressure change in fluid's mechanics. To understand hydrostatic equilibrium or Pascal's law, it would be helpful to provide a detailed explanation of the concept and formula used in calculations. Other content on the application of the theorems, or related lemmas that also focus on the discussion of isotatic equilibrium with the ocean basin would also be useful. |
| ***Example positive document*** |
| Airy |
| The basis of the model is Pascal's law, and particularly its consequence that, within a fluid in static equilibrium, the hydrostatic pressure is the same on every point at the same elevation (surface of hydrostatic compensation):
$h_1 \cdot p_1 = h_2 \cdot p_2 = h_3 \cdot p_3 = ...h_n \cdot p_n$
For the simplified picture shown, the depth of the mountain belt roots (b1) is calculated as follows:
$(h_1 + c + b_1)p_c = (cp_c) + (b_1 p_m)$
$b_1(p_m - p_c) = h_1 p_c$
$b_1 = h_1 p_c/(p_m - p_c)$
where $p_m$ is the density of the mantle (ca. 3,300 kg $m^{-3}$) and $p_c$ is the density of the crust (ca. 2,750 kg $m^{-3}$). Thus, generally:
$b_1 = 5h_1$
... |
| ***Example negative document*** |
| Global or eustatic sea level has fluctuated significantly over Earth's history. The main factors affecting sea level are the amount and volume of available water and the shape and volume of the ocean basins. The primary influences on water volume are the temperature of the seawater, which affects density, and the amounts of water retained in other reservoirs like rivers, aquifers, lakes, glaciers, polar ice caps and sea ice. Over geological timescales, changes in the shape of the oceanic basins and in land/sea distribution affect sea level. In addition to eustatic changes, local changes in sea level are caused by the earth's crust uplift and subsidence.
Over geologic time sea level has fluctuated by more than 300 metres, possibly more than 400 metres. The main reasons for sea level fluctuations in the last 15 million years are the Antarctic ice sheet and Antarctic post-glacial rebound during warm periods.
The current sea level is about 130 metres higher than the historical minimum. Historically low levels were reached during the Last Glacial Maximum (LGM), about 20,000 years ago. The last time the sea level was higher than today was during the Eemian, about 130,000 years ago.
... |

Table 22: **Economics example**. The positive document describes the stochastic dominance, which is the underlying concept used to solve the problem.

---

*Query*

Which of these two lotteries, a consumer with Von-Neumann Morgenstern preferences will choose under exponential distribution?
Consider two lotteries each having an exponential distribution. The function of cumulative distribution of an exponential distribution is:
$F(x; \lambda) = 1 - e^{-\lambda x} \forall x \in R_+$.
The expected gain given by this distribution is $E[X] = 1/\lambda$.
Suppose that the first lottery, $F_1$ has a parameter $\lambda_1$ and the second, $F_2$ has a parameter $\lambda_2$. Suppose that $\lambda_1 < \lambda_2$. Which of these two lotteries, a consumer with Von-Neumann Morgenstern preferences will choose?
I am confused on this question.

---

*Chain-of-thought reasoning to find documents*

The expected value is different from the expected utility. The preferences under uncertainty in this case should be linked to first-order stochastic dominance. We need to look up the detailed definition to understand how the problem could be solved by using that framework.

---

*Example positive document*

Statewise dominance implies first-order stochastic dominance (FSD), which is defined as:
Random variable A has first-order stochastic dominance over random variable B if for any outcome x, A gives at least as high a probability of receiving at least x as does B, and for some x, A gives a higher probability of receiving at least x. In notation form, $P[A \geq x] \geq P[B \geq x]$ for all x, and for some x, $P[A \geq x] > P[B \geq x]$.
In terms of the cumulative distribution functions of the two random variables, A dominating B means that $F_A(x) \leq F_B(x)$ for all x, with strict inequality at some x.
In the case of non-intersecting distribution functions, the Wilcoxon rank-sum test tests for first-order stochastic dominance.
...
Extended example
Consider three gambles over a single toss of a fair six-sided die:
State (die result) 1 2 3 4 5 6
gamble A wins $ 1 1 2 2 2 2
gamble B wins $ 1 1 1 2 2 2
gamble C wins $ 3 3 3 1 1 1
Gamble A statewise dominates gamble B because A gives at least as good a yield in all possible states (outcomes of the die roll) and gives a strictly better yield in one of them (state 3). Since A statewise dominates B, it also first-order dominates B.
...

---

*Example negative document*

Which Lottery Is Easiest To Win Within the U.S.?
Winning the lottery is one of the least likely things to happen in your life. However, some lotteries are much easier to win than others.
It's common for a lot of Americans to daydream about winning the lottery. These dreams became a reality for a few lucky winners as they became millionaires overnight. The cold truth is that the overwhelming majority of lottery dreams will remain dreams.
Most people know that the odds of winning the lottery are not in your favor. If it were easy to win, everyone would play, and the prizes would only be a few cents.
You're likelier to be struck by lightning, live to be 110 years old, and be declared a saint by the Catholic Church.
The combination of high jackpots and low odds has caused many people to try and cheat in the lottery. The attempts might be successful for a short period, but they always end with prison sentences.
Fortunately, there are a few ways that you can legally improve your odds of winning. It will start by choosing games that have the best odds.
...

---

Table 23: **Psychology example**. The positive document describes the critical equation needed to explain the confusion in the post.

---

*Query*

Why do fNIRS devices commonly use two different frequencies?
One of the most common techniques used for functional neuroimaging nowadays is functional near infra-red spectroscopy (fun fact: IIRC Natalie Portman worked on a research paper involving fNIRS as the modality), which shines near infrared light into the brain from a source to a detector (both called optodes) in a "banana" shape.
It's not uncommon to read that most of these devices, be they continuous wave (CW) or one of the two kinds that involve fast modulation, frequency domain (FD) or time domain (TD), require two separate frequencies to be emitted. For instance, NIRx explains it as follows on their website:
"For neuro-imaging applications it is by far most common to illuminate with two discrete wavelength, which is the minimum requirement to assess relative variations of both oxygenation states of the hemoglobin molecule independently."
Why is that the case?
I haven't delved into the intricacies of it, but no reason immediately jumps out at me. For instance, in the case of CW, the relative difference in intensity is all that matters, so why do we need two frequencies?

---

*Chain-of-thought reasoning*

In order to understand why we need two frequencies in fNIRS, we will need to check the measurements and calculation in continuous wave systems. This may be related to oxygenated (HbO) and deoxygenated (HbR) hemoglobin in the brain. We need to refer to the dual-wavelength approach, which is fundamental to fNIRS's ability to provide meaningful information about local brain activity.

---

*Example positive document*

...
Where OD is the optical density or attenuation, $I_0$ is emitted light intensity, $I$ is measured light intensity, $\epsilon$ is the attenuation coefficient, $[X]$ is the chromophore concentration, $l$ is the distance between source and detector and $DPF$ is the differential path length factor, and $G$ is a geometric factor associated with scattering.
When the attenuation coefficients $\epsilon$ are known, constant scattering loss is assumed, and the measurements are treated differentially in time, the equation reduces to:
$\Delta[X] = \Delta OD/(\epsilon d)$
Where d is the total corrected photon path-length.
Using a dual wavelength system, measurements for HbO2 and Hb can be solved from the matrix equation:
$$\begin{pmatrix} \Delta OD_{\lambda 1} \\ \Delta OD_{\lambda 2} \end{pmatrix} = \begin{pmatrix} \epsilon_{\lambda 1}^{Hb} d & \epsilon_{\lambda 1}^{HbO_2} d \\ \epsilon_{\lambda 2}^{Hb} d & \epsilon_{\lambda 2}^{HbO_2} d \end{pmatrix} \begin{pmatrix} \Delta[X]^{Hb} \\ \Delta[X]^{HbO_2} \end{pmatrix}$$
...

---

*Example negative document*

What Is Wave Frequency?
The number of waves that pass a fixed point in a given amount of time is wave frequency. Wave frequency can be measured by counting the number of crests (high points) of waves that pass the fixed point in 1 second or some other time period. The higher the number is, the greater the frequency of the waves. The SI unit for wave frequency is the hertz (Hz), where 1 hertz equals 1 wave passing a fixed point in 1 second. The Figure below shows high-frequency and low-frequency transverse waves.
Q: The wavelength of a wave is the distance between corresponding points on adjacent waves. For example, it is the distance between two adjacent crests in the transverse waves in the diagram. Infer how wave frequency is related to wavelength.
A: Waves with a higher frequency have crests that are closer together, so higher frequency waves have shorter wavelengths.
...

---

Table 24: **Robotics example**. The positive document provides the code pieces to modify.

| *Query* |
| --- |
| Can't see debug messages using RCLCPP_DEBUG |
| I can't see messages using RCLCPP_DEBUG by terminal and rqt, but I can using other levels of verbosity( INFO, ERROR, FATAL. . . ).Selecting debug in rqt to see those messages doesn't work either. |
| I'm using rolling, working in C++ in a plugin of a controller and launching tb3_simulation_launch.py from nav2_bringup. |
| I also saw a post here where they recommended to set the environment variable: |
| RCLCPP_LOG_MIN_SEVERITY=RCLCPP_LOG_MIN_SEVERITY_DEBUG |
| but that didn't work either. It must be something silly that I'm missing. Has this ever happened to you? |
| Thank you |

| *Chain-of-thought reasoning* |
| --- |
| The problem looks like to be related to the setting of log level. If selecting debug in rqt does not work, we can check the code about potential arguments that could affect the logging messages and verbosity, |

| *Example positive document* |
| --- |
| ... |
| declare_log_level_cmd = DeclareLaunchArgument( |
| 'log_level', default_value='info', description='log level' |
| ) |
| load_nodes = GroupAction( |
| condition=IfCondition(PythonExpression([ńot ، use_composition])), |
| actions=[ |
| SetParameter('use_sim_time', use_sim_time), |
| Node( |
| package='nav2_controller', |
| executable='controller_server', |
| output='screen', |
| respawn=use_respawn, |
| respawn_delay=2.0, |
| parameters=[configured_params], |
| arguments=['–ros-args', '–log-level', log_level], |
| remappings=remappings + [('cmd_vel', 'cmd_vel_nav')], |
| ), |
| ... |

| *Example negative document* |
| --- |
| The logger methods are named after the level or severity of the events they are used to track. The standard levels and their applicability are described below (in increasing order of severity): |
| |
| DEBUG |
| Detailed information, typically of interest only when diagnosing problems. |
| INFO |
| Confirmation that things are working as expected. |
| WARNING |
| An indication that something unexpected happened, or indicative of some problem in the near future (e.g. 'disk space low'). The software is still working as expected. |
| ERROR |
| Due to a more serious problem, the software has not been able to perform some function. |
| CRITICAL |
| A serious error, indicating that the program itself may be unable to continue running. |
| ... |

Table 25: **Stack Overflow example**. The positive document provides the documentation that explains the method to solve the problem.

---

*Query*

---

Is there a Snowflake command that can do the following:

a,b,c
1,10,0.1
2,11,0.12
3,12,0.13
to a table like this:

key,value
a,1
a,2
a,3
b,10
b,11
b,13
c,0.1
c,0.12
c,0.13
?

This operation is often called melt in other tabular systems, but the basic idea is to convert the table into a list of key value pairs.

There is an UNPIVOT in SnowSQL, but as I understand it UNPIVOT requires to manually specify every single column. This doesn't seem practical for a large number of columns.

---

*Chain-of-thought reasoning steps to find documents*

---

This operation is a kind of table transformation like reshaping, serialization, or flattening. We are looking for an operation in Snowflake that can take 2-dimensional values, and transform them into an view that presents one-one correlation mapping between key and value.

---

*Example positive document*

---

FLATTEN
Flattens (explodes) compound values into multiple rows.
FLATTEN is a table function that takes a VARIANT, OBJECT, or ARRAY column and produces a lateral view (i.e. an inline view that contains correlation referring to other tables that precede it in the FROM clause).
FLATTEN can be used to convert semi-structured data to a relational representation.
Syntax
FLATTEN( INPUT => <expr> [ , PATH => <constant_expr> ]
                         [ , OUTER => TRUE | FALSE ]
                         [ , RECURSIVE => TRUE | FALSE ]
                         [ , MODE => 'OBJECT' | 'ARRAY' | 'BOTH' ] )
...

---

*Example negative document*

---

SYSTEM$EXTERNAL_TABLE_PIPE_STATUS
Retrieves a JSON representation of the current refresh status for the internal (hidden) pipe object associated with an external table.
Automatically refreshing the metadata for an external table relies internally on Snowpipe, which receives event notifications when changes occur in the monitored cloud storage. For more information, see Introduction to external tables.
Syntax
SYSTEM$EXTERNAL_TABLE_PIPE_STATUS( '<external_table_name>' )
...

---

Table 26: **Sustainable Living example**. The positive document explains the critical concept of soluble salts, which indicate that they contain dissolved minerals and could be harmful to plants.

| *Query* |
| --- |
| How good is it to reuse water from plant pots? |

I'm living in an apartment, and after I water my plants the water goes to plates below the pots. The pots are in a metallic structure above the plates, so I can take the plates to reuse the water (throwing it at the plants again).
This reuse seems beneficial, because I think I can get rid of mosquitoes that would reproduce in the stagnated water. And also some nutrients of the soil (as well as earthworms) can return to the vase.
Is there some negative points in doing that?

| *Chain-of-thought reasoning to find documents* |
| --- |

The water that accumulates in the plates as a result of watering is likely to contain minerals, soluble salts, and other materials that exist in soil or plants. To figure out whether there is any negative point in reusing that water, we thus need to understand whether the components in that water will result in any adverse effects.

| *Example positive document* |
| --- |

Soluble Salts
Soluble salts may accumulate on the top of the soil, forming a yellow or white crust. A ring of salt deposits may form around the pot at the soil line or around the drainage hole. Salts may also build up on the outside of clay pots. In house plants, signs of excess soluble salts include reduced growth, brown leaf tips, dropping of lower leaves, small new growth, dead root tips, and wilting.
Soluble salts are minerals dissolved in water. Fertilizer dissolved in water becomes a soluble salt. When water evaporates from the soil, the minerals or salts stay behind. As the salts in the soil become more and more concentrated, it becomes more difficult for plants to take up water. If salts build up to an extremely high level, water can be taken out of the root tips, causing them to die. High levels of soluble salts damage the roots directly, weakening the plant and making it more susceptible to attack from insects and diseases. One of the most common problems associated with high salt levels is root rot.
The best way to prevent soluble salt injury is to stop the salts from building up. When watering, allow some water to drain through the container and then empty the saucer. Do not allow the pot to sit in water. If the drained water is absorbed by the soil, the salts that were washed out are reabsorbed through the drainage hole or directly through a clay pot.
...

| *Example negative document* |
| --- |

Water reuse in California
Water reuse in California is the use of reclaimed water for beneficial use. As a heavily populated state in the drought-prone arid west, water reuse is developing as an integral part of water in California enabling both the economy and population to grow.
Wastewater Reclaimed water is treated wastewater that comes from homes and businesses, such as sink water, shower water, and toilet water including everything dumped into wastewater drains from laundry soap to bleach to oil to human waste. Wastewater can be divided into greywater and blackwater, with the first being defined as water that had been used for laundry, bathing, sink washing, and dishwashers. Blackwater is defined as sewage that includes feces from toilets.[1] Due to the low amounts of physical pollutants in greywater, most of its contaminants are dissolved organic matter, which can be physically filtered and cleaned through various membranes, as well as through biological treatment methods.
...

Table 27: **LeetCode example.** Both the query and the positive document uses two pointers in the solution. The negative example is retrieved by BM25.

---

*Query*

Given 'n' non-negative integers representing an elevation map where the width of each bar is '1', compute how much water it can trap after raining.
**Example 1:**
**Input:** height =$[0, 1, 0, 2, 1, 0, 1, 3, 2, 1, 2, 1]$
**Output:** 6
**Explanation:** The above elevation map (black section) is represented by array $[0, 1, 0, 2, 1, 0, 1, 3, 2, 1, 2, 1]$. In this case, 6 units of rain water (blue section) are being trapped.
**Example 2:**
**Input:** height = $[4, 2, 0, 3, 2, 5]$
**Output:** 9
**Constraints:**
* 'n == height.length'
* '1 <= n <= 2 * 104'
* '0 <= height[i] <= 105'

---

*Chain-of-thought reasoning to find documents*

This problem can be solved using a two-pointer approach, and uses ideas from dynamic programming to keep track of the maximum height to the left and right of each bar. We can find other example code that also use these techniques.

---

*Example positive document*

```
def max_area(height):
    """You are given an integer array 'height' of length 'n'. There are 'n' vertical lines drawn
```
such that the two endpoints of the 'ith' line are '(i, 0)' and '(i, height[i])'.
Find two lines that together with the x-axis form a container, such that the container contains the most water.
Return the maximum amount of water a container can store.
**Notice** that you may not slant the container.
**Example 1:**
**Input:** height = $[1, 8, 6, 2, 5, 4, 8, 3, 7]$
**Output:** 49
**Constraints:**
* 'n == height.length'
* '2 <= n <= 105'
* '0 <= height[i] <= 104'"""
```
    max_area, left, right = 0, 0, len(height) - 1
    while left < right:
        max_area = max(max_area, min(height[left], height[right]) * (right - left))
        if height[left] < height[right]:
            left += 1
        else:
            right -= 1
    return max_area
```

---

*Example negative document*

```
def get_power(x):
    """Your country has an infinite number of lakes. Initially, all the lakes are empty, but when
```
it rains over the 'nth' lake, the 'nth' lake becomes full of water. If it rains over a lake that is
**full of water**, there will be a **flood**. Your goal is to avoid floods in any lake.
Given an integer array 'rains' where: *...rest of the document omitted*

---

Table 28: **Pony example**

| |
| --- |
| ***Query*** |
| Given the lengths of a triangle's sides, write a pony program to classify it as equilateral, isosceles or scalene. |
| ***Chain-of-thought reasoning to find documents*** |
| To determine the category of the triangle, we need to compare the side lengths. A common syntax for conditions and comparison used in many other programming languages is the 'if' or 'switch' related to control structure. 
 We need to find the syntax that allows to specify such conditions. |
| ***Example positive document*** |

# Control Structures

To do real work in a program you have to be able to make decisions, iterate
through collections of items and perform actions repeatedly. For this, you need
control structures. Pony has control structures that will be familiar to programmers who have used most languages,
such as 'if', 'while' and 'for', but in Pony, they work slightly differently.

## Conditionals

The simplest control structure is the good old 'if'. It allows you to perform some action only when a condition is true.
In Pony it looks like this:

```
if condition then
control_body
end
```

Here is a simple example:

```
if a > b then
env.out.print("a is bigger")
end
```

Often the condition may be composed of many sub conditions connected by 'and' and 'or'.
...

***Example negative document***

# Classes

Just like other object-oriented languages, Pony has __classes__.
A class is declared with the keyword 'class', and it has to have a name
that starts with a capital letter, like this:

```
class Wombat
```

Do all types start with a capital letter?
Yes! And nothing else starts with a capital letter.
So when you see a name in Pony code, you will instantly know whether it's a type or not.

## What goes in a class?

A class is composed of:

1. Fields.
2. Constructors.
3. Functions.
...

Table 29: **AoPS example.** The negative example is retrieved by BM25, and it uses stars and bars to solve the problem, whereas the query and gold document use the Chicken McNugget Theorem.

---

***Query*** *(from 2015 AMC 10B Problem 15)*

The town of Hamlet has 3 people for each horse, 4 sheep for each cow, and 3 ducks for each person. Which of the following could not possibly be the total number of people, horses, sheep, cows, and ducks in Hamlet?
**(A)** 41     **(B)** 47     **(C)** 59     **(D)** 61     **(E)** 66

***Chain-of-thought reasoning to find documents***

We can use the Chicken McNugget Theorem to solve this problem. We can find other solutions that also apply this theorem.

***Example positive document***

Find the sum of all positive integers $n$ such that, given an unlimited supply of stamps of denominations $5, n$, and $n + 1$ cents, 91 cents is the greatest postage that cannot be formed.
By the Chicken McNugget theorem, the least possible value of $n$ such that 91 cents cannot be formed satisfies $5n - (5 + n) = 91 \implies n = 24$, so $n$ must be at least 24.
For a value of $n$ to work, we must not only be unable to form the value 91, but we must also be able to form the values 92 through 96, as with these five values, we can form any value greater than 96 by using additional 5 cent stamps.
Notice that we must form the value 96 without forming the value 91. If we use any 5 cent stamps when forming 96, we could simply remove one to get 91. This means that we must obtain the value 96 using only stamps of denominations $n$ and $n + 1$.
Recalling that $n \geq 24$, we can easily figure out the working $(n, n + 1)$ pairs that can used to obtain 96, as we can use at most $\frac{96}{24} = 4$ stamps without going over. The potential sets are $(24, 25), (31, 32), (32, 33), (47, 48), (48, 49), (95, 96)$, and $(96, 97)$.
The last two obviously do not work, since they are too large to form the values 92 through 94, and by a little testing, only $(24, 25)$ and $(47, 48)$ can form the necessary values, so $n \in \{24, 47\}$. $24 + 47 = \boxed{71}$.

***Example negative document***

Alice has 24 apples. In how many ways can she share them with Becky and Chris so that each of the three people has at least two apples?
**(A)** 105     **(B)** 114     **(C)** 190     **(D)** 210     **(E)** 380
Note: This solution uses the non-negative version for stars and bars. A solution using the positive version of stars is similar (first removing an apple from each person instead of 2).
This method uses the counting method of stars and bars (non-negative version). Since each person must have at least 2 apples, we can remove $2 * 3$ apples from the total that need to be sorted. With the remaining 18 apples, we can use stars and bars to determine the number of possibilities. Assume there are 18 stars in a row, and 2 bars, which will be placed to separate the stars into groups of 3. In total, there are 18 spaces for stars $+2$ spaces for bars, for a total of 20 spaces. We can now do $\binom{20}{2}$. This is because if we choose distinct 2 spots for the bars to be placed, each combo of 3 groups will be different, and all apples will add up to 18. We can also do this because the apples are indistinguishable. $\binom{20}{2}$ is 190, therefore the answer is $\boxed{\textbf{(C) } 190}$.

---

Table 30: **TheoremQA Questions example.** Both the query and the positive document uses the Pigeonhole Principle to solve the problem. The negative document is retrieved by BM25, and it only requires simple arithmetics to solve.

| |
|---|
| *Query* |
| Mary is planning to bake exactly 10 cookies, and each cookie may be one of three different shapes – triangle, circle, and square. Mary wants the cookie shapes to be a diverse as possible. What is the smallest possible count for the most common shape across the ten cookies? |
| *Chain-of-thought reasoning to find documents* |
| This problem is similar to the formulation of Pigeonhole Principle. We can find solutions to other examples that can also be solved using this principle. |
| *Example positive document* |
| Arbitrarily place 19 points in a unit square and cover as many of these points as possible with a circle of diameter $\frac{\sqrt{2}}{3}$. Question: At least how many points can be guaranteed to be covered? We can divide the unit square into 9 smaller squares, each with side length 1/3. Since there are 19 points and 9 smaller squares, by the Pigeonhole Principle, at least one of these smaller squares must contain at least 3 points. |
| Now, consider a circle with diameter $\frac{\sqrt{2}}{3}$. The diagonal of each smaller square has length $\frac{\sqrt{2}}{3}$, so the circle can be placed inside the smaller square in such a way that it covers the entire square. Therefore, the circle can cover at least 3 points in that square. |
| Therefore, the answer is 3. |
| *Example negative document* |
| Yanna baked twenty butter cookies and forty biscuits in the morning. In the afternoon, she baked ten butter cookies and twenty biscuits. How many more biscuits did she bake than butter cookies? She baked 20 + 10 = 30 butter cookies. And, she baked 40 + 20 = 60 biscuits. Therefore, she baked 60 - 30 = 30 more biscuits than butter cookies. The answer is 30 |

Table 31: **TheoremQA Theorems example.** The query uses the Pigeonhole Principle, which is the gold document from ProofWiki. The negative document is retrieved by BM25, which is not relevant to the question

---

*Query*

Mary is planning to bake exactly 10 cookies, and each cookie may be one of three different shapes – triangle, circle, and square. Mary wants the cookie shapes to be a diverse as possible. What is the smallest possible count for the most common shape across the ten cookies?

*Chain-of-thought reasoning to find documents*

This problem is similar to the formulation of Pigeonhole Principle. We may look into the principle and find hints about how this problem may be solve using that principle.

*Example positive document*

Pigeonhole Principle Tags: Pigeonhole Principle, Named Theorems, Combinatorics begin{theorem} Let $S$ be a finite set whose cardinality is $n$. Let $S_1, S_2, \ldots, S_k$ be a partition of $S$ into $k$ subsets. Then: :at least one subset $S_i$ of $S$ contains at least $\lfloor \frac{n}{k} \rfloor$ elements where $\lfloor \cdot \rfloor$ denotes the ceiling function.
end{theorem}

begin{proof} AimForCont no subset $S_i$ of $S$ has as many as $\lfloor \frac{n}{k} \rfloor$ elements. Then the maximum number of elements of any $S_i$ would be $\lfloor \frac{n}{k} \rfloor - 1$. So the total number of elements of $S$ would be no more than $k \left( \lfloor \frac{n}{k} \rfloor - 1 \right) = k \lfloor \frac{n}{k} \rfloor - k$. There are two cases: :$n$ is divisible by $k$ :$n$ is not divisible by $k$. Suppose $k \backslash n$. Then $\lfloor \frac{n}{k} \rfloor = \frac{n}{k}$ is an integer and: :$k \lfloor \frac{n}{k} \rfloor - k = n - k$ Thus: :$|S| = \sum_{i=1}^{k} |S_i| \leq n - k < n$ This contradicts the fact that $|S| = n$. Hence our assumption that no subset $S_i$ of $S$ has as many as $\lfloor \frac{n}{k} \rfloor$ elements was false. Next, suppose that $k mid n$. Then: :$|S| = k \lfloor \frac{n}{k} \rfloor - k < \frac{k(n+k)}{k} - k = n$ and again this contradicts the fact that $|S| = n$. In the same way, our assumption that no subset $S_i$ of $S$ has as many as $\lfloor \frac{n}{k} \rfloor$ elements was false. Hence, by Proof by Contradiction, there has to be at least $\lfloor \frac{n}{k} \rfloor$ elements in at least one $S_i \subseteq S$. qed
end{proof}

*Example negative document*

begin{definition}[Definition:Conditional/Semantics of Conditional]Let $p \implies q$ where $\implies$ denotes the conditional operator.$p \implies q$ can be stated thus:* ""'If" $p$ is true "then" $q$ is true.""* "'$q$ is true "if" $p$ is true.""* "'(The truth of) $p$ "implies" (the truth of) $q$.""* "'(The truth of) $q$ "is implied by" (the truth of) $p$.""* "'$q$ "follows from" $p$.""* "'$p$ is true "only if" $q$ is true.""'The latter one may need some explanation. $p$ can be either true or false, as can $q$. But if $q$ is false, and $p \implies q$, then $p$ can not be true. Therefore, $p$ can be true "only if" $q$ is also true, which leads us to our assertion.* "'$p$ is true "therefore" $q$ is true.""* "'$p$ is true "entails" that $q$ is true.""* "'$q$ is true "because" $p$ is true.""* "'$p$ "may" be true "unless" $q$ is false.""* ""'Given that" $p$ is true, $q$ is true.""* "'$q$ is true "whenever" $p$ is true.""* "'$q$ is true "provided that" $p$ is true.""* "'$q$ is true "in case" $p$ is true.""* "'$q$ is true "assuming that" $p$ is true.""* "'$q$ is true "on the condition that" $p$ is true.""'Further colloquial interpretations can often be found in natural language whose meaning can be reduced down "'$p$ only if $q$"', for example:* "'$p$ is true "as long as" $q$ is true"'::Example::::""Mummy, can I go to the pictures?""::::""""As long as"' you've done your homework. Have you done your homework? No? Then you cannot go to the pictures.""::In other words::::""You can go to the pictures "'only if"' you have done your homework.""::Using the full language of logic::::""If it is true that you are going to the pictures, it is true that you must have done your homework.""* "'$p$ is true "as soon as" $q$ is true"'::::""Are we going to this party, then?""::::""""As soon as"' I've finished putting on my makeup.""::The analysis is the same as for the above example of "'as long as"'.
end{definition}

Table 32: **Example of Gemini reasoning on Sustainable Living**

---

*Query*

---

How good is it to reuse water from plant pots?
I'm living in an apartment, and after I water my plants the water goes to plates below the pots.
The pots are in a metallic structure above the plates, so I can take the plates to reuse the water
(throwing it at the plants again).
This reuse seems beneficial, because I think I can get rid of mosquitoes that would reproduce
in the stagnated water. And also some nutrients of the soil (as well as earthworms) can return
to the vase.
Is there some negative points in doing that?

---

*LLM Reasoning steps*

---

## Plant Pot Water Reuse

**(1) Essential Problem:**

The user wants to know if it is beneficial or detrimental to reuse the water collected from plant
pot drainage plates.
They are concerned about potential mosquito breeding and the impact on plant growth.

**(2) Reasoning:**

**Reused Water Composition:**

- Potentially contains dissolved nutrients, minerals and salts from the soil.
- May contain microorganisms.
- Could harbor mosquito larvae if stagnant.

**Negative Impact on Plant Growth:**

- Stagnant water can attract mosquitoes.
- High salt concentrations from fertilizers can harm plants.
- Waterborne pathogens could potentially harm plants.

**(3) Draft Answer:**

## Reusing Water from Plant Pots: A Balancing Act

It's commendable that you're looking for ways to conserve water and potentially benefit your
plants.
Reusing water from plant pot drainage plates can be a good idea, but there are some drawbacks
to consider:

* **Mosquito Breeding:** Stagnant water in the plates can become a breeding ground for
mosquitoes,
posing a health risk. Regularly empty and clean the plates to prevent this.
* **Salt Accumulation:** Repeatedly using the same water can lead to a buildup of salts from
fertilizers, which can harm plants.
* **Pathogen Risk:** While less common, waterborne pathogens could potentially harm your
plants.
Using filtered or boiled water can minimize this risk.

---

Table 33: **Example of Gemini reasoning on Sustainable Living, continue**

| *Example positive document* |
|---|

Soluble Salts
Soluble salts may accumulate on the top of the soil, forming a yellow or white crust. A ring of salt deposits may form around the pot at the soil line or around the drainage hole. Salts may also build up on the outside of clay pots. In house plants, signs of excess soluble salts include reduced growth, brown leaf tips, dropping of lower leaves, small new growth, dead root tips, and wilting.

Soluble salts are minerals dissolved in water. Fertilizer dissolved in water becomes a soluble salt. When water evaporates from the soil, the minerals or salts stay behind. As the salts in the soil become more and more concentrated, it becomes more difficult for plants to take up water. If salts build up to an extremely high level, water can be taken out of the root tips, causing them to die. High levels of soluble salts damage the roots directly, weakening the plant and making it more susceptible to attack from insects and diseases. One of the most common problems associated with high salt levels is root rot.

The best way to prevent soluble salt injury is to stop the salts from building up. When watering, allow some water to drain through the container and then empty the saucer. Do not allow the pot to sit in water. If the drained water is absorbed by the soil, the salts that were washed out are reabsorbed through the drainage hole or directly through a clay pot.

...

Table 34: **The performance of retrieval models when using reasoning steps generated by GritLM as queries**

| | StackExchange | | | | | | | Coding | | Theorem-based | | | Avg. |
|---|---|---|---|---|---|---|---|---|---|---|---|---|---|
| | Bio. | Earth. | Econ. | Psy. | Rob. | Stack. | Sus. | Leet. | Pony | AoPS | TheoQ. | TheoT. | |
| *Sparse model* | | | | | | | | | | | | | |
| BM25 (Robertson et al., 2009) | 33.1 | 38.7 | 19.2 | 28.0 | 16.8 | 18.9 | 20.6 | 19.7 | 13.2 | 3.3 | 13.0 | 8.9 | 19.4 |
| *Open-sourced models (<1B)* | | | | | | | | | | | | | |
| BGE (Xiao et al., 2023) | 17.7 | 30.1 | 18.8 | 23.0 | 12.3 | 13.3 | 17.4 | 23.5 | 3.2 | 4.0 | 16.7 | 11.6 | 16.0 |
| Inst-L Su et al. (2022) | 21.4 | 27.0 | 19.1 | 25.8 | 13.6 | 13.6 | 18.9 | 16.2 | 0.7 | 4.6 | 20.2 | 8.6 | 15.8 |
| SBERT (Reimers & Gurevych, 2019) | 16.7 | 22.0 | 15.2 | 24.0 | 9.4 | 10.7 | 13.1 | 24.2 | 1.8 | 3.8 | 16.1 | 9.7 | 13.9 |
| *Open-sourced models (>1B)* | | | | | | | | | | | | | |
| E5 (Wang et al., 2023a) | 24.1 | 36.7 | 18.3 | 21.1 | 10.7 | 16.1 | 14.7 | 27.0 | 0.3 | 4.0 | 20.2 | 17.8 | 17.6 |
| SFR (Meng et al., 2024) | 21.8 | 31.5 | 20.0 | 23.0 | 12.1 | 16.5 | 16.6 | 26.2 | 0.6 | 5.3 | 20.2 | 15.6 | 17.4 |
| Inst-XL Su et al. (2022) | 32.9 | 44.3 | 27.1 | 36.4 | 19.3 | 25.8 | 25.2 | 24.6 | 1.6 | 5.6 | 18.1 | 7.9 | 22.4 |
| GritLM (Muennighoff et al., 2024) | 24.7 | 31.5 | 18.5 | 21.1 | 14.0 | 12.3 | 20.7 | 28.9 | 5.1 | 5.3 | 20.0 | 17.7 | 18.3 |
| Qwen (Li et al., 2023b) | 25.4 | 35.9 | 19.5 | 29.7 | 11.8 | 20.3 | 21.7 | 22.9 | 7.9 | 2.6 | 19.3 | 22.4 | 19.9 |
| *Proprietary models* | | | | | | | | | | | | | |
| Cohere (Cohere) | 21.7 | 27.1 | 22.1 | 21.6 | 15.6 | 17.8 | 16.8 | 21.4 | 2.0 | 4.7 | 15.5 | 10.5 | 16.4 |
| OpenAI (OpenAI) | 25.9 | 30.0 | 21.4 | 29.5 | 11.1 | 12.0 | 21.0 | 22.6 | 2.7 | 7.7 | 19.6 | 12.5 | 18.0 |
| Voyage (Voyage AI) | 28.1 | 29.0 | 20.0 | 28.2 | 9.5 | 19.3 | 17.2 | 29.3 | 1.7 | 7.2 | 24.6 | 10.9 | 18.7 |
| Google (Lee et al., 2024) | 25.6 | 34.7 | 22.2 | 30.0 | 14.5 | 21.0 | 16.3 | 28.9 | 1.6 | 8.4 | 18.8 | 12.9 | 19.6 |

Table 35: **The performance of retrieval models when using reasoning steps generated by Gemini-pro as queries**

| | StackExchange | | | | | | | Coding | | Theorem-based | | | Avg. |
|---|---|---|---|---|---|---|---|---|---|---|---|---|---|
| | Bio. | Earth. | Econ. | Psy. | Rob. | Stack. | Sus. | Leet. | Pony | AoPS | TheoQ. | TheoT. | |
| *Sparse model* | | | | | | | | | | | | | |
| BM25 (Robertson et al., 2009) | 54.1 | 49.4 | 20.7 | 33.2 | 19.9 | 21.8 | 23.3 | 20.9 | 12.3 | 2.7 | 18.3 | 9.7 | 23.9 |
| *Open-sourced models (<1B)* | | | | | | | | | | | | | |
| BGE (Xiao et al., 2023) | 28.2 | 35.1 | 21.8 | 25.1 | 13.1 | 15.6 | 18.7 | 24.7 | 5.4 | 3.8 | 18.5 | 14.2 | 18.7 |
| Inst-L (Su et al., 2022) | 35.6 | 41.7 | 22.6 | 33.5 | 17.5 | 19.7 | 16.4 | 17.8 | 0.9 | 6.6 | 25.0 | 12.3 | 20.8 |
| SBERT (Reimers & Gurevych, 2019) | 19.8 | 24.6 | 15.5 | 24.7 | 11.4 | 11.4 | 16.7 | 25.1 | 2.3 | 4.1 | 19.2 | 11.2 | 15.5 |
| *Open-sourced models (>1B)* | | | | | | | | | | | | | |
| E5 (Wang et al., 2023a) | 29.4 | 42.3 | 18.0 | 19.3 | 13.8 | 18.4 | 14.3 | 27.6 | 0.2 | 5.9 | 22.0 | 24.5 | 19.6 |
| SFR (Meng et al., 2024) | 28.9 | 38.4 | 20.3 | 23.8 | 16.2 | 17.2 | 17.1 | 26.8 | 0.4 | 6.9 | 22.8 | 22.5 | 20.1 |
| Inst-XL (Su et al., 2022) | 43.6 | 48.8 | 27.0 | 35.4 | 23.1 | 25.9 | 25.7 | 24.4 | 1.5 | 6.7 | 19.2 | 12.3 | 24.5 |
| GritLM (Muennighoff et al., 2024) | 29.0 | 33.7 | 20.7 | 24.4 | 17.5 | 15.1 | 18.9 | 31.8 | 4.2 | 6.7 | 23.8 | 23.3 | 20.7 |
| Qwen (Li et al., 2023b) | 34.7 | 41.5 | 20.0 | 29.1 | 15.7 | 19.8 | 21.2 | 23.6 | 4.4 | 6.0 | 27.2 | 28.4 | 22.6 |
| *Proprietary models* | | | | | | | | | | | | | |
| Cohere (Cohere) | 30.4 | 33.3 | 23.0 | 27.1 | 16.8 | 21.3 | 21.3 | 20.7 | 3.4 | 5.2 | 19.2 | 16.0 | 19.8 |
| OpenAI (OpenAI) | 34.9 | 38.4 | 24.8 | 33.4 | 15.0 | 15.4 | 17.2 | 24.4 | 4.8 | 6.8 | 23.0 | 20.0 | 21.5 |
| Voyage (Voyage AI) | 37.4 | 40.1 | 22.8 | 29.7 | 13.2 | 19.9 | 19.4 | 32.3 | 2.3 | 5.3 | 28.1 | 18.0 | 22.4 |
| Google (Lee et al., 2024) | 35.5 | 38.9 | 23.9 | 33.1 | 16.3 | 25.2 | 15.9 | 32.3 | 2.7 | 8.0 | 26.3 | 17.9 | 23.0 |

Table 36: **The performance of retrieval models when using reasoning steps generated by Llama3-70B as queries**

| | StackExchange | | | | | | | Coding | | Theorem-based | | | Avg. |
|---|---|---|---|---|---|---|---|---|---|---|---|---|---|
| | Bio. | Earth. | Econ. | Psy. | Rob. | Stack. | Sus. | Leet. | Pony | AoPS | TheoQ. | TheoT. | |
| *Sparse model* | | | | | | | | | | | | | |
| BM25 (Robertson et al., 2009) | 53.8 | 51.4 | 24.1 | 35.3 | 19.6 | 24.8 | 25.6 | 21.1 | 13.7 | 4.9 | 16.6 | 17.5 | 25.7 |
| *Open-sourced models (<1B)* | | | | | | | | | | | | | |
| BGE (Xiao et al., 2023) | 28.6 | 39.1 | 22.7 | 23.5 | 14.7 | 19.0 | 21.9 | 25.4 | 7.6 | 5.4 | 23.2 | 17.2 | 20.7 |
| Inst-L (Su et al., 2022) | 38.5 | 45.4 | 24.8 | 34.4 | 19.5 | 24.2 | 19.4 | 19.4 | 1.3 | 4.4 | 25.1 | 15.9 | 22.7 |
| SBERT (Reimers & Gurevych, 2019) | 19.9 | 25.7 | 16.9 | 24.1 | 10.0 | 13.2 | 16.6 | 24.7 | 6.7 | 3.8 | 20.3 | 14.2 | 16.3 |
| *Open-sourced models (>1B)* | | | | | | | | | | | | | |
| E5 (Wang et al., 2023a) | 30.0 | 43.8 | 18.1 | 23.4 | 13.4 | 17.9 | 15.0 | 26.9 | 0.3 | 5.9 | 20.5 | 23.9 | 19.9 |
| SFR (Meng et al., 2024) | 25.8 | 38.6 | 21.5 | 25.4 | 14.6 | 17.6 | 17.5 | 27.2 | 0.5 | 7.0 | 22.1 | 22.1 | 20.0 |
| Inst-XL (Su et al., 2022) | 44.6 | 51.5 | 32.1 | 40.2 | 20.2 | 30.7 | 27.5 | 25.1 | 1.7 | 6.3 | 21.6 | 14.0 | 26.3 |
| GritLM (Muennighoff et al., 2024) | 28.2 | 34.9 | 22.4 | 24.3 | 17.2 | 17.4 | 20.6 | 32.0 | 4.2 | 5.0 | 23.6 | 21.4 | 20.9 |
| Qwen (Li et al., 2023b) | 32.7 | 43.0 | 21.7 | 30.5 | 13.4 | 22.9 | 21.8 | 25.8 | 8.1 | 5.2 | 26.4 | 29.7 | 23.4 |
| *Proprietary models* | | | | | | | | | | | | | |
| Cohere (Cohere) | 34.7 | 35.1 | 26.3 | 28.1 | 18.2 | 24.9 | 23.7 | 24.5 | 4.2 | 6.5 | 20.3 | 20.3 | 22.2 |
| OpenAI (OpenAI) | 33.8 | 41.1 | 26.1 | 35.6 | 12.2 | 15.9 | 22.1 | 24.1 | 6.0 | 7.8 | 21.7 | 19.3 | 22.1 |
| Voyage (Voyage AI) | 36.8 | 41.2 | 23.8 | 26.9 | 13.2 | 20.7 | 23.3 | 31.9 | 1.5 | 6.9 | 26.8 | 22.0 | 22.9 |
| Google (Lee et al., 2024) | 39.5 | 44.5 | 26.6 | 37.0 | 18.6 | 27.9 | 17.3 | 31.1 | 3.2 | 8.8 | 22.9 | 22.0 | 24.9 |

Table 37: **The performance of retrieval models when using reasoning steps generated by Claude-opus as queries**

| | StackExchange | | | | | | | Coding | | Theorem-based | | | Avg. |
|---|---|---|---|---|---|---|---|---|---|---|---|---|---|
| | Bio. | Earth. | Econ. | Psy. | Rob. | Stack. | Sus. | Leet. | Pony | AoPS | TheoQ. | TheoT. | |
| *Sparse model* | | | | | | | | | | | | | |
| BM25 (Robertson et al., 2009) | 54.2 | 52.1 | 23.5 | 38.4 | 22.5 | 24.2 | 26.0 | 20.0 | 19.6 | 4.1 | 19.0 | 18.1 | 26.8 |
| *Open-sourced models (<1B)* | | | | | | | | | | | | | |
| BGE (Xiao et al., 2023) | 30.2 | 38.5 | 23.5 | 29.3 | 14.4 | 17.2 | 21.8 | 22.4 | 6.2 | 4.3 | 25.3 | 19.9 | 21.1 |
| Inst-L (Su et al., 2022) | 37.4 | 39.5 | 25.2 | 34.4 | 20.0 | 20.6 | 19.0 | 18.2 | 1.6 | 4.1 | 26.8 | 18.5 | 22.1 |
| SBERT (Reimers & Gurevych, 2019) | 18.6 | 24.8 | 18.6 | 24.9 | 11.4 | 12.9 | 14.7 | 23.0 | 5.8 | 3.1 | 20.1 | 19.0 | 16.4 |
| *Open-sourced models (>1B)* | | | | | | | | | | | | | |
| E5 (Wang et al., 2023a) | 31.6 | 43.4 | 20.7 | 25.9 | 13.6 | 19.4 | 14.2 | 29.1 | 0.3 | 6.8 | 25.1 | 27.1 | 21.4 |
| SFR (Meng et al., 2024) | 29.5 | 38.8 | 24.2 | 28.4 | 15.3 | 18.8 | 18.1 | 28.5 | 0.7 | 7.2 | 25.5 | 25.6 | 21.7 |
| Inst-XL (Su et al., 2022) | 45.3 | 50.0 | 28.8 | 43.1 | 22.7 | 27.8 | 26.8 | 24.1 | 1.9 | 6.0 | 23.9 | 16.0 | 26.4 |
| GritLM (Muennighoff et al., 2024) | 32.3 | 36.6 | 22.5 | 28.8 | 19.4 | 19.3 | 24.0 | 31.1 | 6.3 | 6.0 | 26.9 | 27.3 | 23.4 |
| Qwen (Li et al., 2023b) | 35.7 | 45.4 | 22.0 | 34.4 | 15.4 | 22.5 | 25.7 | 24.6 | 5.9 | 6.2 | 28.8 | 31.7 | 24.8 |
| *Proprietary models* | | | | | | | | | | | | | |
| Cohere (Cohere) | 33.4 | 36.8 | 25.1 | 28.5 | 18.9 | 22.9 | 20.7 | 21.7 | 4.4 | 6.8 | 22.6 | 20.7 | 21.9 |
| OpenAI (OpenAI) | 36.2 | 39.8 | 25.8 | 36.6 | 12.9 | 15.7 | 22.1 | 25.3 | 5.6 | 7.7 | 23.4 | 21.8 | 22.7 |
| Voyage (Voyage AI) | 37.1 | 40.4 | 24.8 | 31.1 | 12.4 | 20.4 | 19.9 | 31.3 | 2.1 | 6.9 | 29.6 | 21.8 | 23.1 |
| Google (Lee et al., 2024) | 37.3 | 44.1 | 27.3 | 38.8 | 21.4 | 27.0 | 17.3 | 31.4 | 3.1 | 9.2 | 27.7 | 22.5 | 25.6 |

Table 38: **The performance of retrieval models when using reasoning steps generated by GPT-4 as queries**

| | StackExchange | | | | | | | Coding | | Theorem-based | | | Avg. |
|---|---|---|---|---|---|---|---|---|---|---|---|---|---|
| | Bio. | Earth. | Econ. | Psy. | Rob. | Stack. | Sus. | Leet. | Pony | AoPS | TheoQ. | TheoT. | |
| *Sparse model* | | | | | | | | | | | | | |
| BM25 (Robertson et al., 2009) | 53.6 | 54.1 | 24.3 | 38.7 | 18.9 | 27.7 | 26.3 | 19.3 | 17.6 | 3.9 | 19.2 | 20.8 | 27.0 |
| *Open-sourced models (<1B)* | | | | | | | | | | | | | |
| BGE (Xiao et al., 2023) | 29.3 | 42.2 | 23.4 | 27.9 | 11.3 | 17.6 | 21.5 | 24.2 | 8.2 | 7.5 | 24.6 | 25.8 | 22.0 |
| Inst-L Su et al. (2022) | 39.6 | 45.1 | 24.5 | 34.5 | 17.8 | 26.1 | 19.0 | 18.5 | 1.3 | 6.7 | 28.7 | 20.5 | 23.5 |
| SBERT (Reimers & Gurevych, 2019) | 18.5 | 26.3 | 17.5 | 27.2 | 8.8 | 11.8 | 17.5 | 24.3 | 10.3 | 5.0 | 22.3 | 23.5 | 17.7 |
| *Open-sourced models (>1B)* | | | | | | | | | | | | | |
| E5 (Wang et al., 2023a) | 29.3 | 43.9 | 19.9 | 26.6 | 11.6 | 19.8 | 15.6 | 29.1 | 0.9 | 5.3 | 27.0 | 36.6 | 22.1 |
| SFR (Meng et al., 2024) | 26.0 | 39.4 | 21.4 | 28.3 | 13.1 | 19.4 | 19.3 | 28.4 | 1.5 | 7.1 | 26.7 | 34.1 | 22.0 |
| Inst-XL Su et al. (2022) | 46.7 | 51.2 | 29.9 | 40.5 | 20.8 | 30.1 | 26.9 | 25.1 | 2.1 | 8.2 | 24.2 | 17.0 | 26.9 |
| GritLM (Muennighoff et al., 2024) | 33.3 | 39.1 | 22.4 | 28.9 | 17.4 | 21.3 | 24.1 | 31.9 | 12.0 | 6.7 | 27.3 | 30.1 | 24.5 |
| Qwen (Li et al., 2023b) | 35.5 | 43.1 | 24.3 | 34.3 | 15.4 | 22.9 | 23.9 | 25.4 | 5.2 | 4.6 | 28.7 | 34.6 | 24.8 |
| *Proprietary models* | | | | | | | | | | | | | |
| Cohere (Cohere) | 31.3 | 37.0 | 25.6 | 29.2 | 16.3 | 24.9 | 22.3 | 20.7 | 7.0 | 7.0 | 23.9 | 25.8 | 22.6 |
| OpenAI (OpenAI) | 35.2 | 40.1 | 25.1 | 38.0 | 13.6 | 18.2 | 24.2 | 24.5 | 6.5 | 7.7 | 22.9 | 23.8 | 23.3 |
| Voyage (Voyage AI) | 36.7 | 42.8 | 24.6 | 34.2 | 13.7 | 24.2 | 21.7 | 31.4 | 2.2 | 6.6 | 30.3 | 28.1 | 24.7 |
| Google (Lee et al., 2024) | 36.4 | 45.6 | 25.6 | 38.2 | 18.7 | 29.5 | 17.9 | 31.1 | 3.7 | 10.0 | 27.8 | 30.4 | 26.2 |

Table 39: **Long-context retrieval performance on unsplit web pages of StackExchange data.** The scores are reported in recall@1

| | Bio. | Earth. | Econ. | Psy. | Rob. | Stack. | Sus. | Pony | Avg. |
|---|---|---|---|---|---|---|---|---|---|
| Sparse models | | | | | | | | | |
| BM25 | 10.7 | 15.4 | 10.7 | 8.4 | 7.4 | 22.2 | 10.7 | 5.4 | 11.4 |
| Open-sourced models (<1B) | | | | | | | | | |
| BGE | 16.4 | 27.7 | 20.9 | 11.6 | 10.9 | 13.3 | 16.9 | 0.4 | 14.8 |
| Inst-L | 24.6 | 29.9 | 13.1 | 20.3 | 12.9 | 15.0 | 25.4 | 3.9 | 18.1 |
| SBERT | 25.6 | 34.1 | 18.9 | 15.8 | 10.9 | 15.0 | 18.0 | 1.2 | 17.4 |
| Open-sourced models (>1B) | | | | | | | | | |
| E5 | 29.9 | 36.3 | 26.2 | 46.7 | 17.3 | 14.5 | 32.2 | 1.1 | 25.5 |
| SFR | 30.3 | 37.0 | 24.3 | 47.7 | 17.3 | 14.5 | 35.0 | 2.0 | 26.0 |
| Inst-XL | 21.5 | 31.0 | 13.1 | 20.5 | 13.9 | 15.0 | 20.1 | 6.0 | 17.6 |
| GritLM | 37.5 | 40.3 | 25.7 | 34.4 | 17.8 | 20.1 | 32.4 | 0.0 | 26.0 |
| Qwen | 39.2 | 36.1 | 25.7 | 42.3 | 21.3 | 23.5 | 33.1 | 1.3 | 27.8 |
| Proprietary models | | | | | | | | | |
| Cohere | 31.5 | 34.5 | 18.9 | 20.5 | 9.9 | 15.8 | 15.2 | 0.8 | 18.4 |
| OpenAI | 32.1 | 31.4 | 23.8 | 34.2 | 11.9 | 10.7 | 26.3 | 0.0 | 21.3 |
| Voyage | 34.4 | 35.4 | 26.7 | 41.6 | 12.9 | 12.8 | 31.1 | 1.3 | 24.5 |
| Google | 30.9 | 38.0 | 21.9 | 30.7 | 12.9 | 19.2 | 25.7 | 0.3 | 22.4 |

Table 40: **Statistics of StackExchange and Pony data before web pages and documentation are split.** For each dataset, we show the number of queries and documents, the average length of queries and documents, and the average number of gold documents.

| | # Query | # Doc | Avg Query Len | Avg Doc Len | # Avg Gold Doc |
|---|---|---|---|---|---|
| Biology | 103 | 524 | 115.2 | 9422.4 | 1.3 |
| Earth Science | 116 | 601 | 109.5 | 27312.3 | 1.6 |
| Economics | 103 | 516 | 181.5 | 11896.4 | 1.1 |
| Psychology | 101 | 512 | 149.6 | 12411.66 | 1.1 |
| Robotics | 101 | 508 | 818.9 | 14998.2 | 1.1 |
| Stack Overflow | 117 | 1858 | 478.3 | 40759.7 | 1.1 |
| Sustainable Living | 108 | 554 | 148.5 | 12077.7 | 1.2 |
| Pony | 112 | 577 | 102.6 | 1361.0 | 6.9 |

Table 41: **MiniLM (cross-encoder), Gemini and GPT-4 reranking scores based on BM25 top-10 or top-100 retrieval results.**

| Reranker | top-k | StackExchange | | | | | | | Code | | Math | | | Avg. |
|---|---|---|---|---|---|---|---|---|---|---|---|---|---|---|
| | | Bio. | Earth. | Econ. | Psy. | Rob. | Stack. | Sus. | Leet. | Pony | AoPS | TheoQ. | TheoT. | |
| None | - | 19.2 | 27.1 | 14.9 | 12.5 | 13.5 | 16.5 | 15.2 | 24.4 | 7.9 | 6.2 | 9.8 | 4.8 | 14.3 |
| MiniLM | 10 | 15.4 | 26.6 | 13.0 | 11.8 | 14.3 | 15.4 | 13.6 | 21.8 | 8.7 | 6.1 | 6.5 | 4.2 | 13.1 |
| | 100 | 8.5 | 18.9 | 6.0 | 5.4 | 7.6 | 7.9 | 8.9 | 15.0 | 11.3 | 6.1 | 3.6 | 0.5 | 8.3 |
| Gemini | 10 | 21.9 | 29.7 | 16.9 | 14.2 | 16.1 | 16.7 | 16.7 | 24.5 | 8.0 | 6.2 | 9.5 | 8.2 | 15.7 |
| GPT-4 | 10 | 23.8 | 33.7 | 18.4 | 16.4 | 18.4 | 20.3 | 17.2 | 22.6 | 10.2 | 6.5 | 11.3 | 9.6 | 17.4 |
| | 100 | 33.8 | 34.2 | 16.7 | 27.0 | 22.3 | 27.7 | 11.1 | 3.4 | 15.6 | 1.2 | 2.0 | 8.6 | 17.0 |

Table 42: **MiniLM (cross-encoder), Gemini and GPT-4 reranking scores based on Google retrieval top-10 or top-100 results.**

| Reranker | top-k | StackExchange | | | | | | | Code | | Math | | | Avg. |
|---|---|---|---|---|---|---|---|---|---|---|---|---|---|---|
| | | Bio. | Earth. | Econ. | Psy. | Rob. | Stack. | Sus. | Leet. | Pony | AoPS | TheoQ. | TheoT. | |
| None | - | 23.0 | 34.4 | 19.5 | 27.9 | 16.0 | 17.9 | 17.3 | 29.6 | 3.6 | 9.3 | 21.5 | 14.3 | 19.5 |
| MiniLM | 10 | 17.0 | 30.6 | 15.8 | 20.3 | 12.3 | 15.0 | 14.6 | 24.0 | 6.0 | 9.8 | 14.2 | 11.9 | 16.0 |
| | 100 | 7.5 | 21.7 | 6.4 | 6.2 | 7.0 | 7.1 | 8.3 | 16.0 | 17.2 | 8.1 | 4.2 | 2.9 | 9.4 |
| Gemini | 10 | 23.8 | 35.8 | 19.6 | 29.0 | 16.4 | 17.2 | 18.6 | 29.1 | 5.0 | 9.4 | 20.8 | 16.3 | 20.1 |
| GPT-4 | 10 | 26.1 | 36.5 | 20.9 | 32.6 | 16.8 | 22.6 | 20.8 | 24.5 | 5.5 | 8.9 | 22.9 | 19.8 | 21.5 |
| | 100 | 42.5 | 40.9 | 25.9 | 42.1 | 23.2 | 35.1 | 17.2 | 5.6 | 10.8 | 2.4 | 6.6 | 19.3 | 22.6 |

Table 43: **Gemini prompt to rerank documents.**

---

Gemini

---

The following passages are related to the query: {query}

[1]. {doc 1}
[2]. {doc 2}
...

First identify the essential problem in the query.
Think step by step to reason about why each document is relevant or irrelevant.
Rank these passages based on their relevance to the query.
Please output the ranking result of the most relevant {k} passages as a list,
where the first element is the id of the most relevant passage,
the second element is the id of the second most element, etc.
Please strictly follow the format to output a list of {k} ids:
'''

[...]
'''

---

Table 44: **Results of retrieval models copied from MTEB Muennighoff et al. (2023) for easier reference.** Argu. refers to ArguAna, Climate. refers to ClimateFEVER, CQA. refers to CQADup-stackRetrieval, FIQA. refers to FIQA2018, Hot. refers to HotpotQA, MS. refers to MSMARCO, NF. refers to NFCorpus, Quora refers to QuoraRetrieval, SCI. refers to SCIDOCS, Sci. refers to SciFact, Touche. refers to Touche2020, TREC. refers to TRECCOVID. Except BM25, whose results are from Thakur et al. (2021), all other results are from Muennighoff et al. (2023).

| | Argu. | Climate. | CQA. | DBPedia | FEVER | FIQA. | Hot. | MS. | NF. | NQ | Quora. | SCI. | Sci. | Touche. | TREC. | Avg. |
|---|---|---|---|---|---|---|---|---|---|---|---|---|---|---|---|---|
| *Sparse model* | | | | | | | | | | | | | | | | |
| BM25 (Robertson et al., 2009) | 31.5 | 21.3 | 29.9 | 31.3 | 75.3 | 23.6 | 60.3 | 22.8 | 32.5 | 32.9 | 78.9 | 15.8 | 66.5 | 36.7 | 65.6 | 41.6 |
| *Open-sourced models (<1B)* | | | | | | | | | | | | | | | | |
| BGE (Xiao et al., 2023) | 63.5 | 36.6 | 42.2 | 44.1 | 87.2 | 45.0 | 74.1 | 42.6 | 38.1 | 55.0 | 89.1 | 22.6 | 74.6 | 24.8 | 74.8 | 54.3 |
| Inst-L Su et al. (2022) | 57.1 | 27.7 | 43.8 | 36.7 | 72.7 | 45.5 | 55.2 | 39.7 | 34.1 | 50.1 | 88.4 | 18.6 | 64.3 | 21.6 | 58.1 | 47.6 |
| SBERT (Reimers & Gurevych, 2019) | 46.5 | 22.0 | 45.0 | 32.1 | 50.9 | 50.0 | 39.3 | 39.8 | 33.3 | 50.5 | 87.5 | 23.8 | 65.6 | 19.9 | 51.3 | 43.8 |
| *Open-sourced models (>1B)* | | | | | | | | | | | | | | | | |
| E5 (Wang et al., 2023a) | 61.9 | 38.4 | 43.0 | 48.9 | 87.8 | 56.6 | 75.7 | 43.1 | 38.6 | 63.5 | 89.6 | 16.3 | 76.4 | 26.4 | 87.3 | 56.9 |
| SFR (Meng et al., 2024) | 67.2 | 36.4 | 46.5 | 49.1 | 89.4 | 60.4 | 77.0 | 43.4 | 41.9 | 69.9 | 89.8 | 19.9 | 77.7 | 29.0 | 87.6 | 59.0 |
| Inst-XL Su et al. (2022) | 55.7 | 26.5 | 43.1 | 40.2 | 70.0 | 47.0 | 55.9 | 41.6 | 36.0 | 57.2 | 88.9 | 17.4 | 64.6 | 23.4 | 71.4 | 49.3 |
| GritLM (Muennighoff et al., 2024) | 63.2 | 30.9 | 49.4 | 46.6 | 82.7 | 60.0 | 79.4 | 42.0 | 40.9 | 70.3 | 89.5 | 24.4 | 79.2 | 27.9 | 74.8 | 57.4 |
| Qwen (Li et al., 2023b) | 62.7 | 44.0 | 40.6 | 48.0 | 93.4 | 55.3 | 72.3 | 41.7 | 38.3 | 61.8 | 89.6 | 27.7 | 75.3 | 20.3 | 72.7 | 56.2 |
| *Proprietary models* | | | | | | | | | | | | | | | | |
| Cohere (Cohere) | 61.5 | 38.4 | 41.5 | 43.4 | 89.0 | 42.2 | 70.7 | 42.9 | 38.6 | 61.6 | 88.7 | 20.3 | 71.8 | 32.4 | 81.9 | 55.0 |
| Voyage (Voyage AI) | 64.1 | 32.7 | 46.6 | 46.0 | 91.5 | 59.8 | 70.9 | 40.6 | 40.3 | 65.9 | 87.4 | 24.3 | 80.0 | 39.2 | 85.1 | 58.3 |
| OpenAI (OpenAI) | 58.1 | 30.3 | 47.5 | 44.8 | 87.9 | 55.0 | 71.6 | 40.2 | 42.1 | 61.3 | 89.1 | 23.1 | 77.8 | 23.4 | 79.6 | 55.4 |
| Google (Lee et al., 2024) | 62.2 | 33.2 | 48.9 | 47.1 | 87.0 | 59.2 | 71.3 | 32.6 | 40.3 | 61.3 | 88.2 | 20.3 | 75.4 | 25.9 | 82.6 | 55.7 |

Table 45: QA inference prompt

Problem:
{problem_description}

Documents:
{retrieved_doc}

Based on the provided documents, write an answer to the problem.

Table 46: QA evaluation prompt

———- PROBLEM START ———-
{predicted_answer}
———- PROBLEM END ———-
———- STUDENT ANSWER START ———-
{predicted_answer}
———- STUDENT ANSWER END ———-
———- REFERENCE ANSWER START ———-
{gold_answer}
———- REFERENCE ANSWER END ———-
Criteria:
0 - The student's answer is completely irrelevant or blank.
10 - The student's answer addresses about 10% of the reference content.
20 - The student's answer addresses about 20% of the reference content.
30 - The student's answer addresses about 30% of the reference content.
40 - The student's answer addresses about 40% of the reference content.
50 - The student's answer addresses about 50% of the reference content.
60 - The student's answer addresses about 60% of the reference content.
70 - The student's answer addresses about 70% of the reference content.
80 - The student's answer addresses about 80% of the reference content.
90 - The student's answer addresses about 90% of the reference content.
100 - The student's answer addresses about 100% of the reference content.
Use the following format to give a score:
REASON:
Describe why you give a specific score
SCORE:
The score you give, e.g., 60
Do not say anything after the score.

Table 47: Downstream results of `gpt-4o-2024-08-06` on TheoremQA Questions, TheoremQA Theorems, and AoPS. The results are averaged across 5 random seeds and we report the standard deviation in the subscript.

| Model | Setting | TheoQ. | TheoT. | AoPS |
|---|---|---|---|---|
| GPT-4o | None | $76.3_{1.6}$ | $82.1_{3.0}$ | $36.6_{2.6}$ |
| | Random | $76.5_{1.3}$ | $85.0_{1.8}$ | $37.4_{2.6}$ |
| | Qwen | $76.4_{2.2}$ | $82.6_{2.2}$ | $36.1_{1.9}$ |
| | Oracle | $79.3_{0.4}$ | $89.7_{1.7}$ | $37.2_{3.8}$ |

Table 48: The performance of retrieval models on **BRIGHT** measured by Precision@10.

| | StackExchange | | | | | | | Coding | | Theorem-based | | | Avg. |
| | Bio. | Earth. | Econ. | Psy. | Rob. | Stack. | Sus. | Leet. | Pony | AoPS | TheoQ. | TheoT. | |
|---|---|---|---|---|---|---|---|---|---|---|---|---|---|
| *Sparse model* | | | | | | | | | | | | | |
| BM25 | 7.6 | 12.4 | 7.1 | 6.0 | 5.6 | 8.0 | 6.1 | 6.0 | 7.9 | 3.1 | 2.2 | 1.3 | 6.1 |
| *Open-sourced models (<1B)* | | | | | | | | | | | | | |
| BGE | 5.9 | 8.6 | 8.0 | 8.2 | 5.1 | 4.9 | 5.7 | 6.3 | 5.9 | 3.2 | 2.7 | 1.9 | 5.5 |
| Inst-L | 6.8 | 8.5 | 7.6 | 9.4 | 4.5 | 5.7 | 6.3 | 5.4 | 1.2 | 4.0 | 4.4 | 2.6 | 5.5 |
| SBERT | 7.2 | 7.4 | 8.3 | 10.1 | 4.5 | 5.0 | 6.8 | 6.5 | 7.0 | 3.2 | 3.8 | 2.7 | 6.0 |
| *Open-sourced models (>1B)* | | | | | | | | | | | | | |
| E5 | 8.9 | 10.2 | 8.1 | 8.6 | 7.2 | 4.6 | 6.9 | 6.9 | 4.9 | 4.2 | 5.7 | 6.5 | 6.9 |
| SFR | 9.3 | 10.3 | 9.0 | 10.3 | 7.1 | 5.9 | 7.7 | 6.6 | 2.1 | 4.2 | 5.2 | 6.5 | 7.0 |
| Inst-XL | 10.0 | 13.8 | 10.6 | 11.0 | 6.7 | 8.8 | 9.2 | 6.3 | 4.6 | 4.7 | 3.3 | 1.9 | 7.6 |
| GritLM | 11.1 | 12.7 | 9.2 | 10.8 | 6.8 | 6.0 | 6.8 | 7.5 | 17.9 | 4.6 | 5.7 | 5.3 | 8.7 |
| Qwen | 13.5 | 14.1 | 8.2 | 11.2 | 5.8 | 10.1 | 6.1 | 6.3 | 9.7 | 7.1 | 6.2 | 7.3 | 8.8 |
| *Proprietary models* | | | | | | | | | | | | | |
| Cohere | 8.6 | 10.3 | 10.4 | 10.2 | 6.5 | 7.8 | 8.5 | 6.5 | 1.8 | 3.4 | 3.0 | 1.7 | 6.6 |
| OpenAI | 11.4 | 11.0 | 9.8 | 12.4 | 5.8 | 6.3 | 9.0 | 6.3 | 2.4 | 4.5 | 5.1 | 2.9 | 7.2 |
| Voyage | 11.0 | 9.9 | 9.6 | 11.0 | 5.6 | 7.3 | 7.5 | 7.5 | 1.1 | 5.0 | 5.6 | 3.2 | 7.0 |
| Google | 10.3 | 12.2 | 8.9 | 11.4 | 5.6 | 8.3 | 7.9 | 6.9 | 3.6 | 5.0 | 4.8 | 4.2 | 7.4 |

Table 49: The performance of retrieval models on **BRIGHT** measured by Recall@10.

| | StackExchange | | | | | | | Coding | | Theorem-based | | | Avg. |
| | Bio. | Earth. | Econ. | Psy. | Rob. | Stack. | Sus. | Leet. | Pony | AoPS | TheoQ. | TheoT. | |
|---|---|---|---|---|---|---|---|---|---|---|---|---|---|
| *Sparse model* | | | | | | | | | | | | | |
| BM25 | 21.8 | 31.4 | 16.8 | 15.5 | 19.4 | 16.8 | 21.1 | 29.5 | 3.6 | 6.0 | 11.4 | 9.0 | 16.9 |
| *Open-sourced models (<1B)* | | | | | | | | | | | | | |
| BGE | 15.1 | 27.0 | 16.2 | 18.4 | 14.4 | 12.1 | 17.0 | 31.3 | 3.0 | 7.2 | 14.6 | 11.0 | 15.6 |
| Inst-L | 18.8 | 27.8 | 17.5 | 26.8 | 15.6 | 15.1 | 17.4 | 23.6 | 0.7 | 8.2 | 22.8 | 14.8 | 17.4 |
| SBERT | 18.1 | 25.4 | 18.7 | 23.9 | 11.0 | 12.7 | 18.8 | 31.4 | 3.5 | 5.8 | 20.8 | 15.7 | 17.2 |
| *Open-sourced models (>1B)* | | | | | | | | | | | | | |
| E5 | 22.0 | 29.4 | 18.4 | 18.3 | 18.7 | 11.9 | 23.0 | 34.6 | 2.4 | 8.2 | 27.2 | 34.8 | 20.7 |
| SFR | 22.7 | 30.6 | 21.7 | 25.3 | 19.8 | 16.0 | 25.3 | 32.9 | 1.1 | 7.7 | 25.1 | 35.6 | 22.0 |
| Inst-XL | 27.3 | 38.0 | 25.4 | 35.6 | 22.0 | 21.1 | 23.9 | 31.8 | 2.5 | 8.9 | 16.6 | 9.8 | 21.9 |
| GritLM | 30.3 | 38.8 | 18.3 | 26.9 | 21.3 | 15.1 | 23.4 | 36.3 | 8.2 | 9.4 | 26.2 | 26.6 | 23.4 |
| Qwen | 38.2 | 40.6 | 18.5 | 29.5 | 14.5 | 22.4 | 17.4 | 32.1 | 4.6 | 14.8 | 30.0 | 39.4 | 25.2 |
| *Proprietary models* | | | | | | | | | | | | | |
| Cohere | 23.1 | 29.6 | 22.4 | 25.1 | 17.7 | 21.2 | 23.4 | 31.1 | 0.9 | 7.1 | 15.4 | 9.3 | 18.9 |
| OpenAI | 29.1 | 30.5 | 24.5 | 36.0 | 16.2 | 15.8 | 25.1 | 29.4 | 1.3 | 8.1 | 25.8 | 17.1 | 21.6 |
| Voyage | 29.3 | 31.2 | 21.0 | 31.0 | 15.0 | 17.5 | 20.5 | 41.5 | 0.6 | 8.7 | 28.5 | 15.4 | 21.7 |
| Google | 26.1 | 36.9 | 20.6 | 31.4 | 17.7 | 21.6 | 23.7 | 33.5 | 1.9 | 10.4 | 24.0 | 22.1 | 22.5 |

