# OpenReview forum: "BRIGHT: A Realistic and Challenging Benchmark for Reasoning-Intensive Retrieval"
_ICLR.cc/2025/Conference — ICLR 2025 Spotlight_

### Official Review · Reviewer_8SYb · 2024-10-24

**Soundness:** 4
**Presentation:** 4
**Contribution:** 4
**Rating:** 10
**Confidence:** 4

**Summary:**

## Summary
The paper presents a challenging and novel retrieval benchmark which requires reasoning to answer queries

## Contributions
1. The first benchmark to focus solely on reasoning intensive retrieval. The benchmark is constructed from diverse domains and carefully curated to remove possible noise
2. Thorough experiments are done with respect to a variety of retrieval models -- sparse, open source small models, open source large models and even proprietary models
3. The authors list the data creation and curation process in detail. This would further help in creating similar such benchmarks

**Strengths:**

1. A high quality, relevant and challenging benchmark for information retrieval tasks
2. Comprehensive evaluation on a wide variety of models
3. Love the section on how chain of thought reasoning helps improve the models -- especially BM25!
4. Downstream task evaluation is also very helpful -- confirms the documents are relevant

**Weaknesses:**

I could not think of any questions I had which were not answered in the paper. I have some observations which are mentioned in the questions

**Questions:**

1. (Lines 248-253): Why not use all three annotators independently? It seems you biased the expert annotators with annotations performed by a non-expert
2. Section 5.2 claims that continued pre-training does not help. However, you mention in Appendix A.3 that the average number went to 21.0! Why was 20.4 picked as the final number in Table 5. It is possible that continued pre-training in this fashion (only showing new documents from Stack Exchange) is making the model lose its generalization abilities. There are two options to do this ablation study -- train a model with a replay buffer, which samples data from the original training data along with stack exchange. Train a model where all data is used together (original training data and stack exchange)
3. How does an LLM (such as GPT-4o) work on the downstream task. That is give it a question, and evaluate its performance on answer quality (without giving any document)

---

> ### Author Response · Authors · 2024-11-22
> **Thank you for the review**
>
> Thanks a lot for the recognition of the work! We are encouraged that the reviewer finds the benchmark high-quality, the evaluation comprehensive. We also appreciate the reviewer's recognition of downstream task evaluation and the section on how the chain of thought reasoning helps improve the models. Below we address the questions raised by the reviewer.
>
> **Expert annotation**
> In BRIGHT annotation, all the initial annotation is completed by authors, while the reviews are finished by domain experts with compensations. Since the initial annotation is more time-consuming compared to the reviewing process, we ask authors to complete it to save costs while not sacrificing the overall dataset quality.
>
> **Additional ablation study on model training**
> In section 5.2, we demonstrate that continued pre-training using existing paradigms may not improve the performance in BRIGHT. We use heuristic parameter settings and train models for 10 epochs. In Appendix A.3, we show the results: compared to the performance at the 10th epoch, results at epoch 0,4,5,8,9 are higher, while results at epoch 1,2,3,6,7 are lower. In general, we do not observe notable improvements over the baseline results, which indicates that the existing training paradigms may not be sufficient to solve tasks in BRIGHT.
>
> Following the reviewer’s suggestion, we conduct an ablation study by mixing both StackExchange and original training data from MEDI [1]. We train the model GritLM for 5 epochs and show the results below:
>
> |       |   Biology  |  Earth science | Economic    |   Psychology   |  Robotics   |  Stack Overflow  |  Sustainable Living |  Average  |
> | ------- | -----  | -----  |  -----  | -----  |  -----  | -----  | -----  | -----  |
> | Before training                                                        |  25.0  |  32.8  |  19.0  | 19.9  |  17.3  |  11.6  |  18.0  |  20.5  |
> | Trained on StackExchange data                             |  23.1  |  28.5  |  18.4  |  26.1  |  14.6  |  11.7  |  21.6  |  20.6  |
> | Trained on both StackExchange and original data  |  19.1  | 24.4  |  18.9  |  23.3  |  10.5  | 12.3  | 17.7  | 18.0  |
>
> As shown in the results, training the model with mixing data from StackExchange and the original set fails to improve the performance. This calls for novel approaches that can effectively enhance the reasoning abilities of retrieval models.
>
> **LLM zero-shot performance on QA**
> In Table 4, we report the zero-shot performance of Claude-3.5-sonnet (first row with the retriever None). Although LLMs may have some knowledge of the evaluated benchmarks, we demonstrate that they can still benefit from the retrieved documents, where stronger retrieval models are associated with higher QA results. In particular, when provided with the gold documents, they achieve a performance gain of 17.2 on average.
>
> [1] Su, H., Shi, W., Kasai, J., Wang, Y., Hu, Y., Ostendorf, M., ... & Yu, T. (2022). One embedder, any task: Instruction-finetuned text embeddings. arXiv preprint arXiv:2212.09741.

---

### Official Review · Reviewer_2e76 · 2024-10-31

**Soundness:** 3
**Presentation:** 3
**Contribution:** 3
**Rating:** 6
**Confidence:** 4

**Summary:**

This paper argues that for retrieval tasks, many real-world queries require in-depth reasoning to identify relevant documents, which goes beyond traditional keyword or semantic-based matching focused by existing benchmarks.
To benchmark retrievers on such reasoning-intensive queries, this paper created a new dataset, Bright, which consists of 1398 queries spanning diverse domains.
The benchmarking on this new dataset demonstrated that state-of-the-art retrieval models perform poorly on this dataset.

**Strengths:**

1. This paper focuses on creating a new dataset to benchmark reasoning-intensive queries, which is a very important type of queries for RAG systems and search engines, and there does not exist such a benchmarking dataset so far.

2. The dataset covers a variety of domains, including economics, psychology, mathematics and coding, which is quite comprehensive.

3. The retrieval process in the experiments incorporated explicit reasoning about the query and achieved an up to 12.2 points improvement, which demonstrated that reasoning is indeed a bottleneck for such types of queries, which aligns very well with the motivation of the datasets.

**Weaknesses:**

1. There is no unified definition of either relevance or required reasoning across data from different domains in this dataset. Though the author wrote different "relevance" in different subsections, they are actually not definition of relevance but how the corpus is created.
It is acceptable as a dataset, however, there is a lack of deep scientific understanding about what exact (reasoning) ability is required for a retriever model to success on this dataset. Instead what we can learn from this dataset is that the retriever model may need to overfit some ad-hoc collection process.

2. Due to a lack of unified relevance definition, the dataset is more like a collection of different benchmarking applications where different relevant (can be either closely or loosely) information to the query can be helpful. Therefore, rather than using it as a retrieval benchmark to evaluate retrievers, it is more suitable to use it to evaluate the application. For these applications, the retrieval results are just intermediate results and can be difficult to judge whether they can actually help the final results.

**Questions:**

N/A

---

> ### Author Response · Authors · 2024-11-22
> **Thank you for the review**
>
> Thanks a lot for your review! We are glad to hear that the reviewer finds the BRIGHT benchmark important for RAG systems and search engines with diverse domains. We are also encouraged that the reviewer recognizes the characteristics of reasoning-intensive queries in BRIGHT. Below we address the concerns raised by the reviewer.
>
> **Definition of relevance** While we agree that a unified definition would be appealing, we argue that the diversity of relevance types in real-world scenarios actually enables a systematic study of different reasoning capabilities. Our collection process follows principled approaches that reflect authentic connections between queries and documents - for instance, documents that are linked as supporting evidence for an answer, math questions that can be solved by using the same theorem. The multi-domain nature of BRIGHT (spanning academic subjects, programming, mathematics, etc.) ensures that models cannot simply overfit to specific collection patterns, as success requires genuine reasoning capabilities that generalize across domains. While no benchmark can comprehensively evaluate all aspects of reasoning, we believe BRIGHT's diverse, principled construction across multiple domains provides reliable signals in retrieval models' reasoning capabilities.
>
> **Evaluations on applications** Apart from measuring the retrieval performance, we conduct evaluations on an important application of retrieval: question answering (QA). The results in Table 4 show that, the retrieval of external information enhances the final QA results, with a positive correlation between stronger retrievers and better QA performance. Using the oracle retrieval, we find that the final QA results are significantly improved by 17.2 on average.

---

> > ### Comment · Reviewer_2e76 · 2024-11-28
> >
> > Thanks the author for the response.
> > I wanted to clarify that I am not against a non-unified definition. However, as a benchmark that focuses on reasoning-required retrieval, the datasets in the benchmark would be better to share something in common related to reasoning so that the people who use this dataset can better understand what they achieve with regard to reasoning if they see improvement on this dataset. Otherwise, it seems you can only tell people who use this dataset that they achieve some improvement across some applications, which is quite vague.
> > However, I still believe the current effort has its values, therefore I think 6 is a reasonable score.

---

> > > ### Author Response · Authors · 2024-11-29
> > > **Thank you**
> > >
> > > Thanks a lot for the reviewer’s comments! We are glad to learn that the reviewer agrees with us on non-unified definitions of relevance. We also agree with the reviewer that the paper could be better if benchmark datasets share common properties regarding reasoning capabilities. Motivated by this, we summarize the types of reasoning required in BRIGHT:
> > >
> > > **Deductive reasoning**: The document usually describes a general principle or theorem that could be applied to solve a specific problem present in the query. For example, in Table 31, the general Pigeonhole Principle is applied to solve the specific cookie shape problem. (More examples in Table 21, 22, 23, 25, 28, 31). \
> > > **Analogical reasoning**: The document draws parallel with the query in underlying logic, which indicates that the method used in the document could be also used to solve the query problem. For example, in Table 29, although the query problem appears to be different from the one in the document, they share similar underlying logic and can be both solved by the Chicken McNugget Theorem (More examples in Table 27, 30). \
> > > **Causal reasoning**: The document and the query present a cause-and-effect relationship, i.e., To fix the problem in the query, we need to find the cause in the document. For example, in Table 24, the query presents a problem where the debug message is missing, which is caused by the parameter settings in the source code (positive documents). \
> > > **Analytical reasoning**: The document provides critical concepts or knowledge that helps to explain the confusion in the query. For example, in Table 26, one will need to first analyze the scenario: To determine whether reusing water for plants is beneficial, we need to first understand where the water comes from and what could be contained in the water; As described in the query, the water comes from watering, which goes through the soil and arrives at the plates below the pots; This indicates that the water could carry minerals, soluble salts, and other materials from the soil. The document provides critical knowledge that soluble salts could be dissolved and accumulate in the water, which will be harmful to plants. This piece of information helps to explain why reusing water for plants may not be a good strategy. (More examples in Table 20)
> > >
> > > With categorized reasoning capabilities, people who use BRIGHT can now understand better about their achievements if they observe improvements on specific datasets. In the future revision, we will add these illustrations to provide more clarity on the paper!
> > >
> > > We hope this addresses the reviewer’s questions and further clarifies our idea on reasoning-intensive retrieval. We sincerely thank the reviewer’s insightful comments and valuable feedback and would be more than happy to incorporate any further suggestions!

---

### Official Review · Reviewer_v2W5 · 2024-11-06

**Soundness:** 3
**Presentation:** 4
**Contribution:** 3
**Rating:** 6
**Confidence:** 3

**Summary:**

The paper presents BRIGHT, a challenging retrieval benchmark designed to require deep, beyond-surface-form matching between queries and documents. Human annotators constructed a dataset of 1,398 instances spanning diverse domains. Current retrieval models perform poorly on this benchmark, highlighting its difficulty. The authors also propose several techniques aimed at improving retrieval performance.

**Strengths:**

- The benchmark is challenging and has the potential to drive future research toward developing retrievers that handle difficult queries more effectively.
- The dataset is human-collected, ensuring authenticity rather than relying on artificially generated data.
- I liked the comprehensive appendix, which provided valuable insights into the annotation process and the dataset's structure.

**Weaknesses:**

- Detailed analysis of results is lacking and some (RAG, reranking) are not surprising.
- In line 413, how can LLM-reasoning queries enhance the reasoning capabilities of retrievers? If the primary effect is an increase in lexical similarity (BM25's strong performance), should models be specifically trained to leverage this feature to perform well on BRIGHT? Additionally, the results for Coding and Theorem-based datasets (Table 38) appear inconsistent.
- In line 428, regarding retrieval-augmented generation, the QA performance seems to be already quite high when paired with current retrievers that are reported to perform poorly.

**Questions:**

- Have you tried finetuning retrievers on this benchmark?
- In figure 1, level 2 queries (NQ, MSMARCO) are outdated. It would be better to compare BRIGHT’s “intensive” reasoning to recent retrieval benchmarks, such as RAR-b or subset of BeIR and MTEB that goes beyond semantic matching.

---

> ### Author Response · Authors · 2024-11-22
> **Thank you for the review**
>
> ## Response (1/3)
> We thank the reviewer for the detailed feedback. We are encouraged that the reviewer finds BRIGHT to be a challenging and diverse benchmark with the potential to drive future research on retrieval. We also appreciate the reviewer's recognition on our realistic settings and comprehensive appendix. Below we address the concerns raised by the reviewer.
>
> **Results analysis** We performed analysis, such as querying with LLM reasoning steps, robustness against large-scale pre-training, and long-context retrieval settings, in Sections 4 and 5.
> Since the paper submission, we have also performed additional qualitative analysis into the results.
>
> For example, we observe that even the strongest retriever tends to retrieve semantically-similar texts as opposed to performing additional reasoning on top of the query.
> In the table below, we show some examples of the retrieved results.
> The gold document either explains the theory closely related to the question (e.g., the disposable soma theory of ageing) or provides an example for applying a theorem (e.g., the Vieta’s Formula). However, the retrieved documents only contain semantic overlap with the query but is actually unhelpful for the answering query upon closer examination.
>
> | Query | Gold Document | Qwen Retrieved Document |
> |-----|------|-------|
> | Why does evolution not make our life longer? Why does evolution not make life longer for humans or any other species? Wouldn't evolution favour a long life?                                                                                                                                                                                                                                                                                  | The disposable soma theory of aging states that organisms age due to an evolutionary trade-off between growth, reproduction, and DNA repair maintenance.  Formulated by Thomas Kirkwood, the disposable soma theory explains that an organism only has a limited amount of resources that it can allocate to its various cellular processes. Therefore, a greater investment in growth and reproduction would result in reduced investment in DNA repair maintenance, leading to increased cellular damage, shortened telomeres, accumulation of mutations, compromised stem cells... | More than 300 different theories have been posited to explain the nature (mechanisms) and causes (reasons for natural emergence or factors) of aging. Good theories would both explain past observations and predict the results of future experiments. Some of the theories may complement each other, overlap, contradict, or may not preclude various other theories... |
> | Let $s_k$ denote the sum of the $\\textit{k}$th powers of the roots of the polynomial $x^3-5x^2+8x-13$. In particular, $s_0=3$, $s_1=5$, and $s_2=9$. Let $a$, $b$, and $c$ be real numbers such that $s_{k+1} = a \\, s_k + b \\, s_{k-1} + c \\, s_{k-2}$ for $k = 2$, $3$, $....$ What is $a+b+c$?$\\textbf{(A)} \\; -6 \\qquad \\textbf{(B)} \\; 0 \\qquad \\textbf{(C)} \\; 6 \\qquad \\textbf{(D)} \\; 10 \\qquad \\textbf{(E)} \\; 26$ | Let $r$, $s$, and $t$ be the three roots of the equation $$8x^3 + 1001x + 2008 = 0.$$Find $$(r + s)^3 + (s + t)^3 + (t + r)^3.$$ Solution: By Vieta's formulas, the sum of the three roots is $r+s+t=0$. Thus, we can write \\[(r+s)^3 + (s+t)^3 + (t+r)^3 = (-t)^3 + (-r)^3 + (-s)^3 = -(r^3+s^3+t^3).\\]Since each root satisfies the given equation, we have \\[8r^3 + 1001r + 2008 = 0,\\]so $r^3 = -\\frac{1001}{8}r - 251$. Similar equations hold for $s$ and $t$. Thus, \\[-(r^3+s^3+t^3) = \\frac{1001}{8}(r+s+t) + 3 \\cdot 251.\\]Since $r+s+t=0,$ the answer is $3 \\cdot 251 = \\boxed{753}$.                                                                                                                                                                                                                                                                                                                                                                                 | For a positive integer $n$ and nonzero digits $a$, $b$, and $c$, let $A_n$ be the $n$-digit integer each of whose digits is equal to $a$; let $B_n$ be the $n$-digit integer each of whose digits is equal to $b$, and let $C_n$ be the $2n$-digit (not $n$-digit) integer each of whose digits is equal to $c$. What is the greatest possible value of $a + b + c$ for which there are at least two values of $n$ such that $C_n - B_n = A_n^2$?$\\textbf{(A)} \\text{ 12} \\qquad \\textbf{(B)} \\text{ 14} \\qquad \\textbf{(C)} \\text{ 16} \\qquad \\textbf{(D)} \\text{ 18} \\qquad \\textbf{(E)} \\text{ 20}$ Solution: ...|
>
> We are happy to provide additional analysis if the reviewer has specific questions.

---

> ### Author Response · Authors · 2024-11-22
> **Thank you for the review**
>
> ## Response (2/3)
>
> **How can LLM-reasoning queries enhance the reasoning capabilities of retrievers?**
> Using LLM-generated reasoning steps may help with the retrieval results because it augments the query with additional reasoning steps that can be used to retrieve the gold documents.
> Consider the following example to the previously presented query “Why does evolution not make our life longer? Why does evolution not make life longer for humans or any other species? Wouldn't evolution favour a long life?”, GPT-4 generates the following reasoning steps:
> “Understanding the essential problem here involves recognizing that the question is rooted in how evolution impacts lifespan across different species, including humans, and why it doesn't necessarily favor extending life …  Life involves trade-offs, especially in how organisms allocate their energy. Energy spent on one function, such as growth or reproduction, cannot be spent on another, like repairing cells to prolong life…”
>
> Using this text as the query helps the existing retrieval systems in finding the gold document because it mentions key concepts, such as the trade-offs between prolonging life and expending energy. Since existing retrievers, such as BM25, mainly use these lexical clues to find related documents, it follows that the reasoning-traces should help when the retriever itself cannot perform these critical reasoning steps.
>
> However, this does not necessarily mean that models have to be specifically trained to leverage the lexical similarities, as this is only one method that works well with existing retrieval systems.
> We believe that there are many possible explorations for training retrievers capable of reasoning. For example, one might consider data augmentation techniques that let retrievers internalize the reasoning steps or different loss objectives to encourage more reasoning. We hope that our construction of the BRIGHT benchmark will inspire more research in these directions!
>
> **Results for Coding and Theorem-based datasets**
> In the coding and theorem-based datasets, we find that the trend is largely consistent with the other datasets--the retrieval performance tend to improve with the reasoning steps. Across 13 evaluated retrieval models, the average performance of 5 datasets in coding and theorem-based datasets in Table 38 decreases in only 2 models (GritLM and Qwen) compared to numbers in Table 2. This indicates that the LLM generated reasoning steps are in general helpful for retrieval models to solve tasks in BRIGHT.
>
> **QA results**
>
> We appreciate this insightful observation. We would like to clarify two key points:
>
> * As we described in line 455-456, the LLM as evaluator may be imperfect in measuring the answer quality. To thoroughly investigate this issue, we conducted an ablation study and showed that the LLM deviates from human judgment in more than 40% of cases. Please refer to our response to reviewer XNdW on the ratio of evaluator failure for more details on the experimental settings.
> * In Table 4, we demonstrate that there remains substantial room for improvement: using gold passages improves QA scores by 17.2 points, compared to only a maximum of 6.6 points with current retrievers. We believe these results strongly indicate that existing retrieval models need to be largely improved to solve reasoning-intensive tasks in BRIGHT.

---

> ### Author Response · Authors · 2024-11-22
> **Thank you for the review**
>
> ## Response (3/3)
>
> **Retriever finetuning**
> In BRIGHT, we collect 1398 instances across 12 settings for evaluation and do not prepare a training set. However, in section 5.2, we demonstrate that, finetuning retrievers in benchmark corpus (which enhances their knowledge in relevant domains) and quation-answer pairs in StackExchange (which maintains model’s retrieval capability) does not improve the retrieval results. This indicates that current data formats and training procedures may not be sufficient to fine-tune retrievers for solving tasks in BRIGHT, which calls for novel approaches to improve reasoning-intensive retrieval.
>
> **Compare BRIGHT to datasets beyond semantic matching**
> One dataset that also explores reasoning problems with retrieval models is RAR-b. However, it significantly differs from BRIGHT in their approaches and objectives:
>
> Dataset Construction:
> * RAR-b: Adapts existing multiple-choice benchmarks into a retrieval format, where the queries are original questions and the corpus consists of unique options from all the questions.
> * BRIGHT: Purposefully built as a retrieval benchmark which uses queries and documents in realistic retrieval scenarios.
>
> Document Characteristics:
> * RAR-b: Often uses very short sequences (<10 words) derived from multiple-choice options.
> * BRIGHT: Focuses on substantially longer documents (>100 tokens), more closely mirroring practical retrieval scenarios.
>
> Practical Relevance:
> * RAR-b: It's more conceptual than practical, as real-world retrieval typically involves retrieving documents instead of answers, making this test more about abstract reasoning abilities.
> * BRIGHT: Designed to reflect real-world information seeking behaviors and needs.
>
> We believe that both datasets effectively assess retrievers' capabilities in handling reasoning-intensive queries, and we include clear comparisons between RAR-b and BRIGHT in line 117-121.

---

> > ### Comment · Reviewer_v2W5 · 2024-11-25
> > **Reply**
> >
> > Thank you for the reply. I now recognize the challenges of BRIGHT and the distinction compared to previous benchmarks like RAR-b. While I'm still not entirely surprised by the experimental results, I understand that addressing this is the responsibility of future work. I have raised my score accordingly.

---

> > > ### Author Response · Authors · 2024-11-26
> > > **Thank you!**
> > >
> > > Dear Reviewer,
> > >
> > > Thank you very much for raising the score. Your recognition of BRIGHT is highly appreciated. We would like to express our sincere gratitude for your insightful comments and constructive feedback on the paper!
> > >
> > > Regards, \
> > > Authors of submission 11583

---

### Official Review · Reviewer_XNdW · 2024-11-09

**Soundness:** 3
**Presentation:** 3
**Contribution:** 3
**Rating:** 8
**Confidence:** 4

**Summary:**

The paper presents **BRIGHT**, a benchmark for evaluating retrievers on reasoning intensive tasks. **BRIGHT** contains 1,398 realistic tasks over 12 domains, including Biology, Psychology, Coding, and Math. By experimenting with 13 retrievers, the paper shows that **BRIGHT** is challenging for current systems - the best model reaches an nDCG@10 score of 22.1. Moreover, enhancing queries with reasoning traces improves retrieval performance, retrieval-augmentation with current retrievers improves QA accuracy, re-ranking with LLMs can increase retrieval accuracy, and pre-training on documents in **BRIGHT** does not further increase performance.

**Strengths:**

- The data collection pipeline is thorough and includes verification by two domain-expert PhD students.
- Lots has been done to ensure diversity by focusing on several different StackExchange domains, two coding tasks, and an additional effort for including theorem-based questions.
- The experiments are extensive and cover 13 different retrievers, in addition to two re-rankers.
- The paper is well-written and easy to follow.

**Weaknesses:**

- It could have been helpful to add a quantitative analysis in the analysis section. For example, an analysis that examines when models err could be useful for future research (see the Questions section for further discussion and some suggestions).

- There are a few details in the appendix that are not referenced from the main paper (e.g., the Limitations section), and I found the appendix a bit hard to follow. Consider verifying all main sections are referenced, or alternatively adding a small Table of Contents in the beginning of the Appendix.

**Questions:**

### Questions
Perhaps I missed something, but do you think any of the following analysis will be helpful for future works -
- A quantitative analysis that shows the rank of the gold document for the best retriever (QWEN).
- A qualitative analysis that examines why strong LLMs (e.g., GPT-4 in Table 3) fail to correctly rank relevant documents. The gold document can be added as an oracle to when it was not retrieved. If not, do you think that a different analysis regarding model errors can be helpful?
- The ratio of evaluator failures described in lines 454-458 (although this is minor).

Again, maybe I am missing something, but did you also experiment with the StackExchange answers as positive passages?

### Suggestions
- The abstract mentions that “The leading model on the MTEB leaderboard which achieves a score of 59.0 nDCG@10,1 produces a score of nDCG@10 of 18.0 on **BRIGHT**”. Consider adding which model this is, because the MTEB leaderboard is changing constantly.

- The example for “Level 1: Keyword-based Retrieval” only states that the part of highway 401 is *one of the widest*, while the question asks for the widest highway. I understand this is from the NQ dataset, but a positive passage that directly answers the query might be simpler for the reader.

- The appendix is sometimes referenced as Appendix (e.g., Appendix C) and sometimes as § (e.g., §B.3). Consider using one for consistency (this is of course very minor).

- The subsets of **BRIGHT** are relatively small, with the smallest including only 78 examples. Adding error bars or variance for the results, either in the paper or in a leaderboard, could help future researchers focusing on a specific subset.

---

> ### Author Response · Authors · 2024-11-22
> **Thank you for the review**
>
> Thanks a lot for the insightful reviews! We are glad to hear that the reviewer finds the benchmark diverse, the experiments extensive, data collection pipeline thorough and the paper well-written. Below we address the concerns raised by the reviewer.
>
> **Additional analysis for the rank of gold documents** Following the reviewer’s suggestion, we present the quantitative results on the rank of gold documents with QWEN below. The mean reciprocal rank (MRR) is one of the most common metrics to evaluate the rank of gold documents, while the percentage of examples with gold documents retrieved in the top 100 (Top-100 success rate) is more intuitive in understanding model errors. The results show that in a lot of cases, the gold documents are not ranked in the top 100, which indicates that there is no chance for GPT-4 to achieve high re-ranking performance if it is only provided with the top 100 results.
>
> |  Metrics  |   Biology  |  Earth science | Economic |   Psychology |  Robotics |  Stack Overflow  |  Sustainable Living | LeetCode | Pony | AoPS | Theoremqa_questions | Theoremqa_theorems |
> | ------ | -------  | -------  | ------ | -------  | -------  | ------ | -------  | -------  | ------ | -------  | -------  | -------  |
> | MRR   |  0.383 | 0.466 | 0.236 | 0.298 | 0.200 | 0.265 | 0.201 | 0.331 | 0.240  | 0.263 | 0.329 | 0.347
> | Top-100 success rate  | 46.6% | 47.4% | 34.0% | 44.6% | 24.8% | 59.0% | 25.9% | 45.1% | 10.9% | 13.5% | 39.7% | 60.5% |
>
> **Additional error analysis to add gold documents in reranking** As shown in the results above, one reason for the low performance in reranking can be attributed to the fact that the ground truth documents are not included in the top 100 candidates. Following the reviewer’s suggestion, we include the gold documents for reranking in case they are not retrieved with QWEN. In three evaluated datasets (one from StackExchange, one from coding and one from math), we show that including gold documents significantly enhances the performance, which confirms that one error source is the lack of gold documents in reranking.
> |   |   Biology  |  LeetCode | AoPS |
> | ------ | -------  | -------  | ------ |
> | w/o reranking |   30.9  |   25.5      |     27.8     |
> | w/o including gold documents |   43.0  |    11.2       |      14.9        |
> | w/ including gold documents |  50.9  | 36.6 | 34.4
>
> **The ratio of evaluator failure** To investigate the ratio of evaluator failures in the QA evaluation, we randomly sample 50 examples from Biology, LeetCode and AoPS and ask 3 people to give ratings to each LLM-generated answer independently. We take the average of the 3 scores as the human score and consider the LLM evaluation to be wrong if the LLM score deviates from the human score by more than 10 (where the full score is 100). Below we show the percentage of examples where the evaluator fails:
>
> |   Biology  | Psychology | StackOverflow |
> | -------  | -------  | ------ |
> |   42%  |   46%      |     48%     |
>
> As shown in the results, there is a significant portion of examples where the LLM evaluation deviates from human judgment. This calls for future research to develop more reliable metrics to evaluate open-ended question-answering tasks.
>
> **StackExchange answers as positive passages** Following the reviewer’s suggestion, we conducted experiments by using StackExchange answers as positive passages with QWEN, and the results in NDCG@10 are shown below. In the BRIGHT benchmark, we do not use them as positive passages, because we would like to simulate a realistic scenario where an exact answer to the query does not exist, and the retrieval models can only retrieve positive documents to help human users or LLMs to derive an answer (line 244-247).
>
> | Biology  |  Earth science | Economic |   Psychology |  Robotics |  Stack Overflow  |  Sustainable Living |
> | -------  | -------  | ------ | -------  | -------  | ------ | -------  |
> |  51.9  |  52.1 |   43.8   |  43.7  | 34.6 | 47.5 | 31.7 |
>
> **Organizations and references in Appendix** Following the reviewer's suggestion, we add a table of contents before the appendix and use consistent references (e.g., Appendix C) throughout the paper.
>
> **MTEB leaderboard clarification**
> Thanks a lot for the suggestions! We specify the leading model on 2024.05.28 as SFR-Embedding-Mistral [1] in the revision.
>
> **NQ example**
> We updated the NQ example in Figure 1 in the revision.
>
> **Small dataset**
> In the evaluation of retrieval models in BRIGHT, we do not observe result variances across different runs. For more detailed discussions, please refer to our response 2/2 to the reviewer aDWB on the small dataset and comparison with other benchmarks.
>
> [1]. MENG, R., LIU, Y., JOTY, S. R., XIONG, C., ZHOU, Y., & YAVUZ, S. SFR-Embedding-Mistral: Enhance Text Retrieval with Transfer Learning [Salesforce AI Research Blog]. 2024.

---

> > ### Comment · Reviewer_XNdW · 2024-11-24
> > **Response to authors**
> >
> > Thank you for your response, I do not have any follow-up questions at this time.

---

> ### Author Response · Authors · 2024-11-26
> **Thank you!**
>
> Dear Reviewer,
>
> We are glad to hear that our response has addressed your questions! Thank you very much for your detailed review and invaluable suggestions on the paper!
>
> Regards, \
> Authors of submission 11583

---

### Official Review · Reviewer_aDWB · 2024-11-11

**Soundness:** 4
**Presentation:** 3
**Contribution:** 3
**Rating:** 6
**Confidence:** 4

**Summary:**

The authors propose Bright, an information retrieval benchmark that focuses on questions that require 'reasoning-intensive' retrieval to provide relevant content. Whereas most existing benchmarks focus on factoid questions where there is significant semantic overlap between the query and the retrieved content (or some 'translation', but not 'reasoning'). Specifically, Bright is actually 12 datasets including 7 datasets from StackExchange covering {Biology, Earth Science, Economics, Psychology, Robotics, Stack Overflow, Sustainable Living}, 2 coding data settings from {LeetCode (python), Pony}, and 3 math reasoning datasets {TheoremQA-Question Retrieval, TheoremQA-Theorem Retrieval, Art of Problem Solving}. Details are provided regarding the procedures to collect each dataset. In short, for StackExchange, human annotators browse recent posts and select a post with at least 5 upvotes and contain at least one URL link -- which are human validated to produce questions, answers, and relevant documents. Negative documents are collecting via identifying semantically similar but irrelevant documents (i.e., negatives) via a Google search powered method. These are human validated for unanimity. Pony coding adapts a code generation dataset to retrieving pages from manuals to cover syntax and LeetCode via a fairly straightforward crawling procedure. Math reasoning adapts TheoremQA to retrieve math queries that either use the same theorem as the query's solution (question retrieval), theorems from ProofWiki (theorem retrieval), or Math Olympiad problems (Art of Problem Solving) matching other problems that use the same theorems. Experiments are conducted based on 13 different retrieval engines including sparse retrieval, open-source dense retrieval, and proprietary models. The important findings is that nDCG@10 has variance amongst different systems (i.e., improvements can be made) while being relatively low as compared to other benchmarks (i.e., it is difficult). Additional experiments show that querying with LLM reasoning (i.e., chain-of-thought) improves performance (i.e., reasoning is needed for retrieval, irrespective of underlying retrieval method), retrieval improves RAG-based results (i.e., retrieval is an important problem). They also demonstrate that reranking with increasingly powerful LLMs improves retrieval performance, Bright appears robust with respect to data leakage in pre-training (i.e., pre-training doesn't cover reasoning requirements as much as most tasks), and that long-content retrieval (e.g., legal, medical) is more difficult.

**Strengths:**

A strong benchmark paper should satisfy some of the following dimensions (along with some commentary):
- the task is useful
Difficult document retrieval is a long- and widely-studied problem. It is both more important in the era of LLMs due to increased reasoning capabilities, but potentially less important as more information is encoded in the parameters (modulo time-sensitive, etc.). Thus, additional motivation could help justify the significance of the work.

- the dataset is large enough and non-trivial to construct
This is also mixed; the dataset isn't particularly large and is ostensibly of varying quality between Stack Exchange, Coding, and Math. That being said, it is clearly more complex than many existing benchmarks for at least a subset of the questions. For some questions, the quality is seemingly higher (i.e., more human validation) than existing datasets.

- there are sufficient details regarding the construction of the benchmark
Including the appendices, there are a lot of details -- to the point where I am confident I could replicate most of the results. However, the amount and clarity of the procedure for different data sets (Stack Exchange, Coding, and Math) isn't as detailed for all cases. Also, it isn't clear in general what the human annotation guidelines were, how annotators were recruited, and how they were compensated (unless it is just the authors and volunteers). However, the details are solid overall.

- the tools provided reduce friction for new people to work on this
Code is provided and was used to run several experiments. I didn't dig through the code and thus do not know how easily it is to conduct experiments. However, I am reasonable confident it is sufficient.

- the baseline models tested on the benchmark are non-trivial
The authors conduct several experiments over several different retrieval engines including state-of-the-art systems on related datasets.

- the benchmark answers new questions or enables new solutions
The authors did conduct experiments beyond just IR performance and were able to address some of these questions using this dataset. The discussion in these sections could be strengthened, but it is solid in this regard overall.

Evaluating the paper with respect to the stated dimensions,
Originality: There are multiple 'hard QA/IR' datasets, but the emphasis here is on IR for reasoning-heavy scenarios -- which is timely and a useful contribution. Many have likely considered such datasets, but the execution here is better than a first attempt.
Quality: Overall, the work is well-motivated, well-executed, and sufficiently rigorous. My primary concern in this regard is variance in quality between different benchmark types (QA, Math, Coding) and that this is a relatively small dataset.
Clarity: Overall, the paper is easy to understand and has sufficient details, especially when considering the appendices. The figures are helpful. My two suggestions in this regard are a Table comparing Bright to the most related datasets and more discussion regarding the empirical results including specific references to cells in the tables (i.e., I didn't always know which cells I was looking at when validating quantification claims).
Significance: I am fairly certain that at least part of this benchmark will be used, but not sure if all parts will be used. Additionally, it would have more potential impact if it was a larger dataset (or there was clear evidence that it covers some expected 'production' distribution)

**Weaknesses:**

On the other hand, below are some of my concerns regarding this paper (some previously mentioned):
- I would like to see a statistics-level comparison between Bright and competing datasets in a table ("Benchmarking retrieval" (line 104) and "Benchmarking reasoning" (line131) in Related Work section)
- I would like clarification regarding the annotation guidelines, recruiting experts, compensation for lines 248-253.
- To get the details for coding and math, one pretty much has to read the appendices and from what I can tell, the significance and quality of the Stack Exchange questions is the strongest aspects of the paper.
- The appendices are more detailed (to the point where they actually seem different from the text). However, still no details regarding annotation guidelines, recruiting experts, and compensation (unless the authors did all of this)
- The dataset seems relatively small; if I am incorrect, I would recommend a table contrasting this with other datasets (along with other aspects).
- For reasoning steps, a bit more from the appendix (e.g., StackExchange vs. coding vs. math stratification) would be helpful in the main text with discussion.
- In general, it isn't clear that ordering matters for RAG settings, so NDCG-based results may not be that useful as in IR settings. I also would recommend rank-biased precision and recall (i.e., evaluations similar to 'needle-in-a-haystack' settings.
- As implied in other areas, there are a lot of results, so more specific interpretation would be helpful (but I am aware of the page limit).
- While the authors claim that there are not licensing issues, I wasn't able to verify this. Obviously, if there are licensing issues (for academic research within commercial organizations?).

**Questions:**

The only two clarification issues I had are:
- Details regarding human annotator guidelines, annotator recruitment, and compensation. This would help in better understanding the reliability of the human annotations.
- More specifics regarding licensing (i.e., a table with licensing terms for each dataset being used).

---

> ### Author Response · Authors · 2024-11-21
> **Thank you for the review**
>
> ## Response (1/2)
>
> We sincerely thank the reviewer for their positive feedback, acknowledging that our work is solid, non-trivial, original, and a timely contribution. Below we address the concerns raised by the reviewer.
>
> **Motivation to justify a retrieval benchmark in the era of LLMs** We agree with the reviewer that stronger LLMs encode more knowledge and facts. However, relying on large models to memorize and retrieve facts is far less efficient than leveraging dedicated retrieval models in terms of both latency and cost. This inefficiency is one of the key reasons why retrieval and embedding models remain highly popular in practical applications. Furthermore, even though existing LLMs have been extensively pre-trained on web corpora, incorporating documents retrieved from the Internet still significantly enhances their QA performance, as demonstrated in Table 4. Therefore, we believe that strong retrieval models provide complementary advantages in the era of powerful LLMs, and it is important to build a challenging and realistic benchmark to facilitate research in this area.
>
> **Human annotator guidelines** In Appendix E.1, we provide guidelines to annotate datasets in StackExchange. For TheoremQA, we also use manual annotation to ensure the quality of the rewritten questions. The instructions are specified in Appendix B.2, B.3, and E.2. For AoPS, the manual efforts involve browsing the AoPS wiki and finding problem-solving topics that are associated with multiple questions. We also carefully check the data collected in this process, as detailed in Appendix B.5 and E.3. For LeetCode, we deliberately choose questions that are grounded in real-world scenarios where the oracle algorithms are not immediately obvious. The details are in Appendix B.6 and E.4. We hope these illustrations help address the reviewer's questions regarding the annotation procedure!
>
> **Human annotator recruitment and compensation** All initial annotations were completed by the authors. We recruit domain experts from PhD students in corresponding fields to review the initial annotation and give a compensation of 300HKD per 100 examples.
>
> **NDCG-based results may not be that useful as in IR settings** The focus on BRIGHT is to primarily serve as an information retrieval benchmark, where using NDCG@k is a standard practice in previous works (e.g., BEIR [1], MS-MSMARCO [2]). In RAG evaluation settings, NDCG@k is also used for evaluating the retrieval quality [3]. Thus, we choose to use NDCG@10 as the main metric on BRIGHT.
> In practice, we find that retrievers with higher NDCG@10 scores also yield better performance on the downstream evaluation (Table 4).
>
> In the Appendix, we further include the performance measured by Precision@10 and Recall@10 in Table 49 and 50. The results show similar trends as those measured by NDCG@10 (Table 2), where Qwen remains the strongest retrieval model on average, and in general, the large open-sourced models and the proprietary models perform better than sparse and small open-sourced models.
>
> [1]. Thakur, N., Reimers, N., Rücklé, A., Srivastava, A., & Gurevych, I. (2021). Beir: A heterogenous benchmark for zero-shot evaluation of information retrieval models. arXiv preprint arXiv:2104.08663. \
> [2]. Bajaj, P., Campos, D., Craswell, N., Deng, L., Gao, J., Liu, X., ... & Wang, T. (2016). Ms marco: A human generated machine reading comprehension dataset. arXiv preprint arXiv:1611.09268. \
> [3]. Wang, Z. Z., Asai, A., Yu, X. V., Xu, F. F., Xie, Y., Neubig, G., & Fried, D. (2024). CodeRAG-Bench: Can Retrieval Augment Code Generation?. arXiv preprint arXiv:2406.14497.

---

> ### Author Response · Authors · 2024-11-21
> **Thank you for the review**
>
> ## Response (2/2)
>
> **Small dataset and comparison with other benchmarks** In the table below, we compare against related information retrieval benchmarks with complex queries.
>
> | Benchmark                      | Number of Datasets | Number of Test queries | Task              | Construction        |
> |--------------------------------|------------|----------------|-------------------------|---------------|
> | RAR-b [1]      |         22 |         45,745 | Directly retrieve answer to queries                                    | Re-processed from existing datasets.                                                                       |
> | BIRCO [2]     |          5 |            410 | Re-rank a pool of up to 100 documents according to a complex objective | Re-processed from existing datasets.                                                                       |
> | FollowIR [3] |          3 |            349 | Re-rank documents according to an instruction                          | Additional annotation on existing datasets.                                                                |
> | InstructIR [4]   |          1 |           1267 | Retrieve documents according to an instruction                         | GPT-4 generates and validates documents for queries from existing datasets.                                 |
> | BRIGHT (Ours)                  |         12 |           1398 | Retrieve documents that help answer reasoning-intensive questions      | Human-collected and expert-validated query-document pairs, and additional annotation on existing datasets.  |
>
> As shown in the table, BRIGHT comprises 1,398 test instances spanning 12 domains. While this may appear small by NLP standards, in the information retrieval (IR) community, 30–50 queries are often considered both valid and effective, as each query requires meticulous annotation across numerous documents [3][5].
> The only benchmark with significantly more queries than BRIGHT is RAR-b. However, this dataset is constructed by directly converting questions and answers from multiple-choice datasets into queries and gold passages. We believe this does not reflect a natural IR setting, where the answer is not readily available in the corpus, making it less realistic.
>
> **Data license** Thanks a lot for the reviewer’s questions on data license. The largest source of BRIGHT, StackExchange uses [CC BY-SA](https://stackoverflow.com/help/licensing), which allows to share or adapt materials for any purpose; TheoremQA uses [MIT](https://huggingface.co/datasets/choosealicense/licenses/blob/main/markdown/mit.md), which also allows to use or modify the dataset content. We follow previous works [6][7] in only using only publicly visible portions of LeetCode and AoPS, avoiding high-cost data collection. According to the [section 107 of copyright act](https://www.copyright.gov/title17/92chap1.html#107), “the fair use of a copyrighted work, including such use by ... scholarship, or research, is not an infringement of copyright”, where fair use is determined by “the purpose and character of the use, including whether such use is of a commercial nature or is for nonprofit educational purposes”, “the amount and substantiality of the portion used in relation to the copyrighted work as a whole”, and “the effect of the use upon the potential market for or value of the copyrighted work”. Therefore, we believe that our data collection should not imply any licensing issue.
>
> [1]. Xiao, C., Hudson, G. T., & Moubayed, N. A. (2024). RAR-b: Reasoning as Retrieval Benchmark. arXiv preprint arXiv:2404.06347. \
> [2]. Wang, X., Wang, J., Cao, W., Wang, K., Paturi, R., & Bergen, L. (2024). BIRCO: A Benchmark of Information Retrieval Tasks with Complex Objectives. arXiv preprint arXiv:2402.14151. \
> [3]. Weller, O., Chang, B., MacAvaney, S., Lo, K., Cohan, A., Van Durme, B., ... & Soldaini, L. (2024). FollowIR: Evaluating and Teaching Information Retrieval Models to Follow Instructions. arXiv preprint arXiv:2403.15246. \
> [4]. Oh, H., Lee, H., Ye, S., Shin, H., Jang, H., Jun, C., & Seo, M. (2024). INSTRUCTIR: A Benchmark for Instruction Following of Information Retrieval Models. arXiv preprint arXiv:2402.14334. \
> [5]. Webber, W., Moffat, A., & Zobel, J. (2008, October). Statistical power in retrieval experimentation. In Proceedings of the 17th ACM conference on Information and knowledge management (pp. 571-580). \
> [6]. Jain, N., Han, K., Gu, A., Li, W. D., Yan, F., Zhang, T., ... & Stoica, I. (2024). Livecodebench: Holistic and contamination free evaluation of large language models for code. arXiv preprint arXiv:2403.07974. \
> [7]. Hendrycks, D., Burns, C., Kadavath, S., Arora, A., Basart, S., Tang, E., ... & Steinhardt, J. (2021). Measuring mathematical problem solving with the math dataset. arXiv preprint arXiv:2103.03874.

---

> ### Author Response · Authors · 2024-11-27
> **Reminder for Paper Discussion**
>
> Dear reviewer,
>
> As we approach the end of the discussion period, we would greatly appreciate your input on the paper. We hope our responses and additional results address your concerns and welcome any further questions or suggestions. Thank you so much for all your time and efforts in reviewing the paper!
>
> Regards, \
> Authors of submission 11583

---

> ### Author Response · Authors · 2024-12-02
> **Reminder for Paper Discussion**
>
> Dear reviewer,
>
> As the deadline for the discussion period approaches, we hope that we have addressed your concerns about the comparisons between BRIGHT and other related benchmarks, licensing, and additional results. We would greatly appreciate it if you could respond to our rebuttal. Please let us know if you have any other questions or concerns!
>
> Regards, \
> Authors of submission 11583

---

### Public Comment · ~Dayu_Yang1 · 2024-11-21

This is definitely an interesting paper.
Intuitively, when considering different queries, the definition of a relevant document may vary.

However, it seems that this paper:
- Does not sufficiently motivate why the reasoning ability of large language models can effectively address this problem.
- It appears that the primary application of reasoning is in Section 4.2, where large language models are used for query expansion by following a fixed prompt format. I think this approach lacks significant innovation compared to traditional query expansion methods.

---

> ### Author Response · Authors · 2024-11-22
> **Thanks for the comment**
>
> Thanks a lot for your recognition of the paper. Below we address the questions raised in the comment.
>
> **Explanations on why the reasoning ability of large language models can effectively address the BRIGHT problems**
> Please refer to our response 2/3 on "How can LLM-reasoning queries enhance the reasoning capabilities of retrievers?" to reviewer v2W5.
>
> **Comparison to query expansion** We agree that the approach to first generating reasoning steps and using them for retrieval shares similarities with the existing research in query expansion. However, we would like to clarify that: (1). This approach is not the main contribution to the paper. Instead, we consider it as a baseline implementation for the BRIGHT benchmark; (2). The significant improvement achieved by using LLM-generated reasoning steps as queries confirms that reasoning is a critical capability required to solve the problems in BRIGHT.

---

### Author Response · Authors · 2024-11-22
**Summary response to all reviewers and the new revision**

We thank all the reviewers for their feedback and constructive comments. We are glad to hear that: The paper focuses on an important problem with the potential to drive the future research (Reviewer aDWB, v2W5, 2e76, 8SYb), The BRIGHT benchmark is challenging and diverse (Reviewer XNdW, v2W5, 2e76, 8SYb), The data collection pipeline is thorough and the benchmark is of high-quality (Reviewer XNdW, v2W5, 8SYb), The experiments are comprehensive (Reviewer XNdW, 8SYb), The appendix is comprehensive with sufficient details and valuable insights (Reviewer aDWB, v2W5), The paper is easy to follow (Reviewer aDWB, XNdW).

In this work, we aim to propose a challenging retrieval benchmark that focuses on realistic scenarios. Different from existing benchmarks where relevant documents can be retrieved through lexical or semantic matching, BRIGHT requires in-depth reasoning during the retrieval process, which mirrors realistic scenarios in coding, math, robotics and more. Experimental results show that existing retrieval models are not capable of solving tasks in BRIGHT, while LLM reasoning and reranking demonstrate to be effective approaches to enhance the performance.

In the revision, we updated the draft based on the reviewer’s comments:
* We specify the leading model as SFR-Embedding-Mistral in the abstract.
* We update the example in Figure 1 for easier understanding.
* We replace all the Appendix references § with “Appendix”
* We reorganize the appendix and add a table of contents.

---

### Meta-Review · Area_Chair_C6vr · 2024-12-21

**Metareview:**

This paper presents a new dataset for retrieval problems that require some kind of reasoning to solve. It describes the construction of tasks in several domains, including coding and math (where reasoning is required to retrieve problems using using similar solution components) and StackExchange QA (where reasoning is required to retrieve answers as opposed to merely related content).

The core idea of this paper is very timely and it advances the state of existing datasets in an interesting way.  The paper has several features of a strong benchmark paper: interesting task, high-quality construction process, good evaluation of baselines, etc. It is well-written and clear.

There are some questions about the size of the dataset, its relation to prior work, details of the annotation, etc.  There are also various suggested analyses. However, none of these is critical.  The most major conceptual issue is what 2e76 brings up: the definition of the task (specifically relevance) feels stretched.  aDWB wonders if all parts will be used. I think part of the answer to this is a conversation that can take place in the literature building on this paper.

**Additional Comments On Reviewer Discussion:**

Reviewer aDWB provides a good overview of what is needed for the benchmark, which captures most of the critiques and comments about the paper:

1. The task is useful: The authors justify this well in the response.

2. The dataset is large enough and varied enough: This is one of the most significant issues with the work, in that it focuses on particular types of reasoning. However, this is also a somewhat vacuous complaint, as any dataset can always be broader and higher quality.

3. The dataset has enough details: The reviewers generally found the paper clear and remaining points have been clarified.

4. Good experiments: There was some discussion of these points but generally the experimental evaluation is thorough and most concerns are resolved.

5. The benchmark answers new questions or enables new solutions: The work is generally strong along this axis.

One of the other most significant points in my view is 2e76's critique of the unified definition of relevance. I think this is fundamentally not resolved, but also nobody in the field really has good definitions for "reasoning" that I have seen, so I think this is not necessary to resolve in this paper.

---

### Decision · Program_Chairs · 2025-01-22

Accept (Spotlight)